# Injectable hydrogel bioelectrostimulator for wireless deep brain neuromodulation

Ming Yang[1,11], Wenliang Liu[1,11], Ping Chen[1,11], Zhuang Liu[2,11], Renyuan Sun[1,11], Baochun Xu[3], Qiong Wang[4], Bingqing Xue[5], Chuan Gao[1], Jiahui She[1], Chong Ma[1], Dingke Zhang[1], Zhikun Li[1], Nanxi Yi[1], Donghui Zhang [5], Jiexiong Feng [4], Cunjiang Yu [3,6,7,8] ✉, Jie Wang[2,9] ✉ & Zhiqiang Luo [1,4,10] ✉

Deep brain stimulation (DBS) is effective for treating neurological and psychiatric disorders. However, its tethered configuration, invasiveness, and limited tissue compatibility motivate wireless, minimally invasive alternatives. Here, we develop an in situ-gelled injectable conductive hydrogel (ICH), enabling wireless neuromodulation via electric-field localization under volume conduction. The ICH forms in vivo through bio-catalyzed polymerization and electrostatic self-assembly, yielding a stable, highly conductive, tissue-soft, and biocompatible network. Under high-frequency capacitive coupling, impedance difference between the ICH and surrounding brain tissue induces interfacial polarization and charge accumulation, locally concentrating the electric field to activate nearby neurons. This mechanism is supported by enhanced calcium signaling, increased c-Fos expression, and electrophysiological evidence of balanced basal ganglia-cortical activity. In a Parkinson's disease rat model, ICH-mediated stimulation improved locomotor behavior, preserved dopaminergic neurons, and restored functional connectivity and structural integrity as revealed by fMRI. This injectable hydrogel bioelectronics provides a platform for minimally invasive, wireless neuromodulation therapies.

Deep brain stimulation (DBS), a neuromodulation technique involving the implantation of electrodes into targeted brain regions, has emerged as a powerful means of intervention for neurological and psychiatric disorders, including Parkinson's disease (PD), Alzheimer's disease, and epilepsy[1,2]. However, current DBS devices rely on rigid electrodes and bulky implanted batteries that are mechanically incompatible with soft dynamic tissue, increasing the risk of device failure, foreign body response, and tissue damage[3]. Limited battery

[1]National Engineering Research Center for Nanomedicine, College of Life Science and Technology, Huazhong University of Science and Technology, Wuhan, China. [2]Department of Neurology, Songjiang Hospital Affiliated to Shanghai Jiao Tong University School of Medicine, Shanghai, China. [3]Department of Electrical and Computer Engineering, University of Illinois, Urbana-Champaign, Urbana, IL, USA. [4]Department of Pediatric Surgery, Tongji Hospital, Tongji Medical College, Huazhong University of Science and Technology, Wuhan, China. [5]Stem cells and Tissue Engineering Manufacture Center, School of Life Science, Hubei University, Wuhan, China. [6]Department of Materials Science and Engineering, University of Illinois, Urbana-Champaign, Urbana, IL, USA. [7]Department of Mechanical Science and Engineering, University of Illinois, Urbana-Champaign, Urbana, IL, USA. [8]Department of Bioengineering, Materials Research Laboratory, Beckman Institute for Advanced Science and Technology, Nick Holonyak Micro and Nanotechnology Laboratory, University of Illinois, Urbana-Champaign, Urbana, IL, USA. [9]Shanghai Key Laboratory of Emotions and Affective Disorders, Songjiang Research Institute, Songjiang Hospital Affiliated to Shanghai Jiao Tong University School of Medicine, Shanghai, China. [10]Research Center for Intelligent Fiber Devices and Equipment, State Key Laboratory of New Textile Materials and Advanced Processing, Huazhong University of Science and Technology, Wuhan, China. [11]These authors contributed equally: Ming Yang, Wenliang Liu, Ping Chen, Zhuang Liu, Renyuan Sun. ✉e-mail: cunjiang@illinois.edu; jie.wang@shsmu.edu.cn; zhiqiangluo@hust.edu.cn

capacity necessitates periodic recharging or surgical replacement, while transcutaneous leads for power delivery pose risks of infection and mechanical failure[4-6]. Although flexible polymer-based device substrates have been introduced to improve biocompatibility, their mechanical mismatch with neural tissue remains unresolved[7]. Moreover, neuroimaging techniques such as magnetic resonance imaging (MRI) are essential for elucidating therapeutic mechanisms, but conventional metal electrodes cause substantial heating and image artifacts, limiting their integration with functional imaging[8]. These limitations underscore the need for next-generation DBS technology that integrates minimally invasive, MRI-compatible electrodes with wireless-powered, miniaturized stimulators to improve biocompatibility and functionality.

Injectable nanomaterial-enabled neuromodulation provides a less invasive alternative to conventional electrode-based DBS by eliminating rigid components including electrodes, leads, batteries, and control circuits[9,10]. Functional nanomaterials can be injected into deep brain regions, where they transduce external fields into localized electrical or thermal stimuli for neuromodulation[11,12]. For example, magnetothermal stimulation with $Fe_3O_4$ nanoparticles (under 500 kHz, 15 kA m$^{-1}$ alternating magnetic field) raised tissue temperature above 43 °C to activate transient receptor potential cation channel subfamily V member 1 (TRPV1)-positive neurons[13]. Similarly, near-infrared-II (NIR-II) photothermal stimulation with conjugated polymer nanoparticles (under 1 Wcm$^{-2}$ NIR-II 1064 nm laser illumination) activates TRPV1$^+$ neurons at 39 °C[14]. Moreover, $Fe_3O_4$-$CoFe_2O_4$-$BaTiO_3$ nanoparticles under a tailored magnetic field (220 mT static and 10 mT alternating at 100–150 Hz), generated localized electric potentials for transgene-free neuromodulation[15]. More recently, ultrasound-driven piezoelectric nanoparticles such as $BaTiO_3$, when excited by focused ultrasound (~1 MHz, 1 W cm$^{-2}$), generate local electric fields sufficient to depolarize neuronal membranes[16], and hybrid upconversion-photovoltaic nanoparticles convert near-infrared light (808–980 nm) into photocurrents that directly trigger neuronal depolarization without genetic modification[17]. Despite their distinct working principles, these field-responsive nanomaterials share common challenges including high-intensity field requirements, limited energy conversion efficiency, and gradual metabolic clearance that undermine long-term stability. Moreover, effective neuromodulation requires intimate particle-cell coupling, which is difficult to maintain in heterogeneous tissue environments, and the gradual degradation or metabolic clearance of these materials further limits long-term stability and spatial precision[18].

Conductive hydrogels, owing to their excellent mechanical compliance and biocompatibility, have emerged as promising neural interface materials that enable seamless electrode-tissue integration and stable electrical communication[19,20]. In peripheral nerve interfaces, conductive hydrogels can serve as either coating materials or standalone electrodes, helping to establish robust neural interfaces for long-term applications[21,22]. However, when used as thin-film or fiber electrodes in brain-implanted bioelectronic devices, conductive hydrogels still depend on rigid substrates, which compromises their adaptability, triggers immune responses, and reduces interfacial stability in DBS applications[23,24]. Although one might embed wireless nanomaterial transducers within hydrogels to achieve self-powered stimulation, these materials cannot maintain the particle-cell coupling required for effective energy transduction once encapsulated, and their intrinsic instability further limits long-term performance, making durable and localized stimulation difficult to achieve. These limitations motivate an injectable conductive hydrogel that simultaneously serves as the neural interface and the transduction medium, integrating electrode and wireless power within a single soft implant and thereby simplifying device architecture. To realize this concept, two challenges must be addressed: (1) developing synthesis strategies to overcome the limitations of current injectable conductive hydrogels, including conductivity-biocompatibility trade-offs, mechanical stability, and cytotoxicity[25-27]; and (2) establishing an effective wireless electrical stimulation mechanism that leverages the hydrogel's ability to mediate energy transmission via external fields, which remains largely unexplored.

Herein, we demonstrate a minimally invasive, wireless DBS strategy enabled by in vivo-gelled injectable conductive hydrogels (ICHs), which can induce localized electric field concentration around the implanted ICH under in vivo volume conduction scenario (Fig. 1). The ICH forms a conductive network through bio-catalyzed polymerization and electrostatic assembly, showing high conductivity, tissue-like mechanical properties, and biocompatibility for stable electrical transmission at the electrode-tissue interface. Upon applying a high-frequency volume conduction into brain tissue via capacitive wireless power transfer, the impedance difference between the ICH and surrounding tissue induces interfacial polarization accompanied by charge accumulation, resulting in localized electric field concentration that enables effective neural stimulation. Using a wearable capacitive-coupling setup, ICH-mediated wireless DBS was applied for neuromodulation therapy in a rat model of PD. The MRI compatibility of ICH enabled whole-brain functional MRI (fMRI) analysis of neural connectivity during DBS therapy. This ICH-based wireless DBS approach addresses the limitations of conventional DBS and advances next-generation neuromodulation strategies.

## Results
### Design and characterization of ICHs
For the preparation of in vivo injectable conductive hydrogels, in situ polymerization of conjugated monomers and crosslinking of the resulting polymers in living tissues have been shown to be feasible[28]. However, improving biocompatibility remains challenging due to the potential cytotoxicity of high monomer and crosslinker concentrations. To address this limitation, we explored injectable conductive polymer hydrogels via the electrostatic assembly of negatively charged poly(3,4-ethylenedioxythiophene) polystyrene sulfonate (PEDOT:PSS) and positively charged polymerized pyrrole (PPy) directly in brain tissue without toxic crosslinkers. Stable hydrogel networks formed under physiological conditions using low concentrations of Py (0.22 wt%) and PEDOT:PSS (0.67 wt%). Py polymerization was catalyzed by a glucose oxidase-horseradish peroxidase (GOx-HRP) cascade utilizing endogenous glucose (Fig. 1a). In the presence of $O_2$, GOx catalyzes the oxidation of glucose to produce $H_2O_2$, which is subsequently reduced to $H_2O$ by HRP. During this process, Py monomers are oxidized into radicals that polymerize into conjugated PPy. Unlike conventional approaches that physically mix PEDOT:PSS into hydrogel matrices, where low loading yields insufficient conductivity and high loading increases stiffness or brittleness[25], the biocatalyzed polymerization of pyrrole in the presence of PEDOT:PSS forms a uniform interpenetrating conductive network. This process enhances electrical connectivity while maintaining softness and injectability, and does not require any solvent or acid post-treatment[26,27]. The biocatalytic polymerization of Py was visually confirmed by a rapid color change from yellow to black (Supplementary Fig. 1). Ultraviolet-visible (UV-Vis) absorption spectroscopy further confirmed the transition, showing a redshift from 300–350 nm (Py monomer) to 400–450 nm (PPy) (Supplementary Fig. 2).

Electrostatic crosslinking between PPy and PEDOT:PSS led to the formation of highly conductive polymer networks (Fig. 1b). To directly verify this electrostatic interaction, zeta potential measurements were performed under neutral conditions. The separately prepared PEDOT:PSS and PPy dispersions showed opposite surface charges (−40 mV and +18 mV, respectively), confirming the charge asymmetry required for electrostatic complexation (Supplementary Fig. 3). Optical microscopy confirmed the formation of characteristic microgel structures (Supplementary Fig. 4). Structural changes induced by biocatalytic

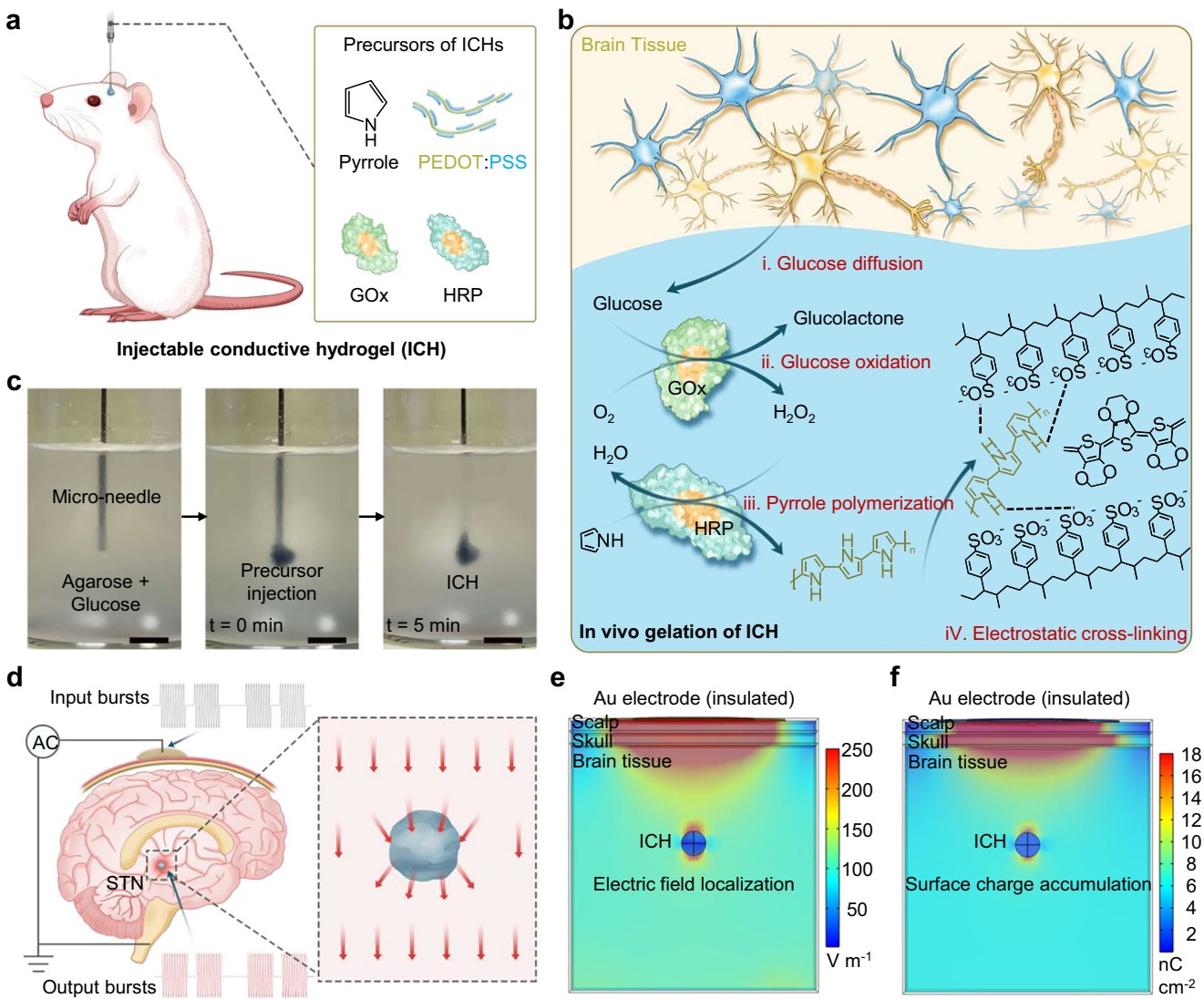

**Fig. 1 | Wireless DBS with injectable hydrogel bioelectronics. a** Preparation of in vivo injectable conductive hydrogel (ICH) with glucose-initiated gelation within brain tissue. **b** Bio-catalytic polymerization and electrostatic assembly mechanism of ICHs. GOx, glucose oxidase; HRP, horseradish peroxidase; PEDOT:PSS, poly(3,4-ethylenedioxythiophene): polystyrene sulfonate. **c** Representative images show in situ gelation of ICHs in tissue-mimicking agarose gel over time. Scale bar, 5 mm. **d** A wearable capacitive-coupling system induces weak volume conduction in brain tissue via capacitive wireless power transfer, and the impedance mismatch between ICH and native tissue drives interfacial charge redistribution at the hydrogel-tissue interface. STN, subthalamic nucleus. **e** Finite element simulation of electric field intensity distribution across the scalp-skull-brain multilayered structure, incorporating tissue-specific dielectric properties, demonstrates electrical field localization at the ICH-tissue interface due to interfacial polarization. **f** Finite element simulation of surface charge distribution in the multilayer head model, demonstrates charge accumulation at the ICH-tissue interface due to interfacial polarization.

polymerization were characterized by Fourier transform infrared (FTIR) and Raman spectroscopy. In the FTIR spectra, the 1560 cm$^{-1}$ band, attributed to the C=C stretching of the conjugated PEDOT/PPy backbone, becomes stronger and broader, suggesting enhanced π-π stacking and charge delocalization[29]. The 1160 cm$^{-1}$ peak, assigned to the symmetric C–O–C stretching in PEDOT, also intensifies, reflecting stronger interchain coupling. Meanwhile, the C–O–C/C–H vibration shifts from ~1220 to 1260 cm$^{-1}$, implying modified bonding environments and conformational reorganization within the hybrid network[30]. Raman spectra corroborated successful glucose-induced polymerization of Py, with characteristic peaks at 1058 cm$^{-1}$ and 1575 cm$^{-1}$ in ICH (Supplementary Fig. 5b). These peaks correspond to C=C/C=N stretching and C-H deformation/C-N stretching modes of PPy, confirming its incorporation into the PEDOT:PSS network[31].

To evaluate injectability, ICH precursors were injected into agarose gels (0.5% agarose) containing glucose, which serves as a tissue-mimicking model, providing both mechanical support and a suitable environment for polymerization (Fig. 1c). A physiologically relevant glucose concentration (5 mM) was used to mimic the neural tissue microenvironment. Upon injection, the precursor gelled in situ within 5 min, forming ICHs embedded in agarose. The morphology of the resulting hydrogels was governed by local diffusion and reaction kinetics. Rheological measurements confirmed that glucose is essential for initiating and sustaining Py polymerization in ICHs (Supplementary Fig. 6a). At 5 mM glucose, the storage modulus (G′) increased rapidly within 1 min, indicating a liquid-to-solid transition; in contrast, no significant change was observed without glucose. The formed ICHs showed a shear elasticity with minimal frequency dependence (Supplementary Fig. 6b), indicating a soft but stable polymer network. Increasing glucose concentration from 3 to 5 mM accelerated gelation, shortening the transition time from ~2.5 to ~1 min (Supplementary Fig. 7a, b). The corresponding Young's modulus increased from 360.04 to 495.07 Pa, indicating enhanced network formation (Supplementary Fig. 7c). These values fall within the physiological range of

brain tissue (0.1–10 kPa), confirming that the ICH possesses tissue-like softness and mechanical compliance comparable to native brain tissue.

The electrical and electrochemical properties of the ICHs were also modulated by glucose concentration. To identify the source of electrical conductivity, non-polymerized precursor mixtures without pyrrole monomers were tested as controls, including PEDOT:PSS alone, PEDOT:PSS with GOx/HRP, PEDOT:PSS with glucose, and PEDOT:PSS with both GOx/HRP and glucose, all showing low conductivity ($<6\,S\,m^{-1}$) (Supplementary Fig. 8a). In contrast, the conductivity of ICHs increased markedly with glucose concentration, reaching ~$30\,S\,m^{-1}$ at 5 mM (Supplementary Fig. 8b), confirming that the enhanced conductivity arises from the enzymatic oxidation of pyrrole to polypyrrole. Electrochemical impedance spectroscopy revealed reduced impedance across all frequencies with increasing glucose concentration, particularly at low frequencies (Supplementary Fig. 9a). The phase-angle spectra showed a decrease at low frequencies, indicating that the capacitive behavior of the hydrogel became less dominant (Supplementary Fig. 9b). This reduced capacitive dominance reflects a transition from interfacial polarization-dominated charge storage to more efficient bulk charge transport through the conductive hydrogel network[32]. Impedance values at representative frequencies (Supplementary Fig. 9c) further confirmed this trend, with ICHs formed at 5 mM glucose exhibiting the lowest impedance.

Glucose-induced polymerization also enhanced the charge storage capacity (CSC) of the ICHs. As shown in cyclic voltammetry (Supplementary Fig. 10a), ICHs formed at 5 mM glucose exhibited a larger and more rectangular voltammogram loop, indicating enhanced current response and improved electrochemical performance. The CSC increased to over $2.03\,mC\,cm^{-2}$ in ICHs formed at 5 mM glucose, compared to ~$0.83\,mC\,cm^{-2}$ at 0 mM and minimal values on bare Au electrodes (Supplementary Fig. 10b). Under biphasic voltage pulses ($\pm0.5\,V$), ICHs formed with glucose showed higher peak current densities (Supplementary Fig. 11a), indicating improved charge injection. The charge injection capacity (CIC) reached ~$182.9\,\mu C\,cm^{-2}$ at 5 mM glucose, compared to ~$106.2\,\mu C\,cm^{-2}$ at 0 mM and ~$54.65\,\mu C\,cm^{-2}$ for Au electrodes (Supplementary Fig. 11b). These results indicate that glucose-induced PPy formation enhances both charge storage capability and injection capability of the ICHs.

**Wireless electrical stimulation mediated by ICH**

To enable wireless electrical stimulation (WES) via ICH, we utilized interfacial polarization arising from impedance mismatch between the ICH and surrounding brain tissue, which induces charge accumulation and localized electric field concentration at the hydrogel-tissue interface. In a capacitive wireless power transfer setup (Fig. 1d), a soft insulated metal plate placed on the scalp acts as the power transmitter, while the grounded brain serves as the power receiver. High-frequency volume conduction can be generated in brain tissue through electrostatic induction. At the ICH-tissue interface, mismatched dielectric relaxation times ($\tau = \varepsilon/\sigma$), arising from conductivity and permittivity differences, hinder uniform field response and induce interfacial polarization[33]. This polarization effect was quantitatively analyzed by calculating the interfacial charge accumulation. Finite element modeling was then performed to simulate interfacial polarization and field localization in a scalp-skull-brain model with/without ICH (2.5 μL, ~1.7 mm diameter) (Supplementary Fig. 12). At 5 MHz, the introduction of the ICH markedly enhanced local electric field intensity around the implantation site (Fig. 1e and Supplementary Fig. 13a). The degree of field localization increased with the conductivity of the ICH, indicating stronger polarization and field confinement near highly conductive hydrogels. Correspondingly, the surface charge density at the ICH-tissue interface also increased with conductivity (Fig. 1f and Supplementary Fig. 13b), consistent with enhanced interfacial polarization caused by permittivity and conductivity mismatch between the two

media. The resulting charge accumulation intensified local current density (Supplementary Fig. 13c), confirming that higher-conductivity ICHs enhance interfacial polarization and lead to stronger charge focusing at the hydrogel-tissue interface. To assess biosafety of such a wireless power transfer process, we calculated the specific absorption rate (SAR) using a frequency-adaptive circuit model (Supplementary Fig. 14a) that incorporates capacitive transmitter dynamics, multilayer tissue impedance, and interfacial polarization elements. This equivalent circuit was adapted from established capacitive wireless power transfer frameworks that model tissue as a frequency-dependent lossy dielectric medium with multilayer impedance characteristics[34–36]. The model integrates frequency-adaptive permittivity and conductivity values for scalp, skull, and brain tissues to reproduce realistic coupling behavior and interfacial charge accumulation under high-frequency electric fields. Across all tested frequencies ($\leq5\,MHz$), the calculated SAR remained below $2\,W\,kg^{-1}$, well within internationally accepted safety limits for local tissue exposure[37], indicating safe operation under the capacitive coupling-based volume conduction scenario (Supplementary Fig. 14b).

To investigate frequency-dependent electrical output of ICH under capacitive volume conduction, we performed ex vivo electrical measurements after injecting 2.5 μL of ICHs into the subthalamic nucleus (STN) of the rat brain. As shown in Fig. 2a, $U_1$ and $U_2$ denote the voltages at the ICH-brain interface and adjacent tissue, respectively. $I_1$ is the current through the ICH, and $I_2$ is the total current across the brain. Frequency-dependent voltage and current outputs were recorded across 100 kHz to 7 MHz, revealing a peak electrical response at 5 MHz (Supplementary Fig. 15a, b). Therefore, subsequent experiments were conducted at 5 MHz to maximize electrical performance while staying within biosafety limits. Under 5 MHz capacitive coupling, voltage and current responses were recorded across increasing input voltages (0.5–5 V). As shown in Fig. 2b, $U_1$ scaled linearly from $81.84 \pm 17.48\,mV$ to $734.8 \pm 45.3\,mV$, and $U_2$ from $47.4 \pm 6.02\,mV$ to $390.6 \pm 34.28\,mV$ with increasing input voltage. $I_1$ and $I_2$ increased proportionally with current densities $J_1$ and $J_2$ rising from $0.53 \pm 0.08\,mA\cdot cm^{-2}$ to $4.73 \pm 0.51\,mA\cdot cm^{-2}$ and from $0.107 \pm 0.01\,mA\cdot cm^{-2}$ to $0.9 \pm 0.06\,mA\cdot cm^{-2}$, respectively (Fig. 2c).

To further study the influence of ICH on the current path, we compared $J_1$ with additional configurations including $J_3$ (from the center of the ICH to adjacent tissue) and $J_4$ (tissue-tissue with PU insulation) (Supplementary Fig. 16a, b). We also evaluated whether cortical regions, which are closest to the external transmitter electrode, might be indirectly stimulated under volume conduction by measuring cortical voltage ($U_3$) and current density ($J_5$) in the absence of ICH, using the same relative electrode spacing as in the deep-brain measurements ($U_2$ and $J_4$) (Supplementary Fig. 16c). Both $J_1$ and $J_3$ involve ICH in the current path, and they exhibited similarly higher current density with increasing input voltages (Supplementary Fig. 16d). Whereas $J_4$, which lacked ICH, showed minimal current density. Furthermore, cortical ($U_3/J_5$) and deep-brain ($U_2/J_4$) measurements without ICH showed similar electrical responses (Supplementary Fig. 16e, f), indicating that the electrical field distributes broadly across brain layers in the absence of ICH, resulting in diffuse and insufficient stimulation for neuronal activation. These findings confirm that ICH concentrates current density at the ICH-brain interface, which would enable effective, tether-free neural excitation. In addition, this ICH-mediated stimulation enables programmable electrical bursts via modulation of input power, pulse width, and stimulation frequency (Fig. 1d).

We next evaluated the efficacy of the ICH in modulating neuronal activity through in vitro wireless electrical stimulation. To this end, intracellular $Ca^{2+}$ dynamics were monitored in SH-SY5Y cells, chosen for their strong adhesion and consistent calcium signaling[38]. A custom cell culture chamber integrated with patterned indium tin oxide (ITO) electrodes was used to deliver electrical stimulation (Supplementary

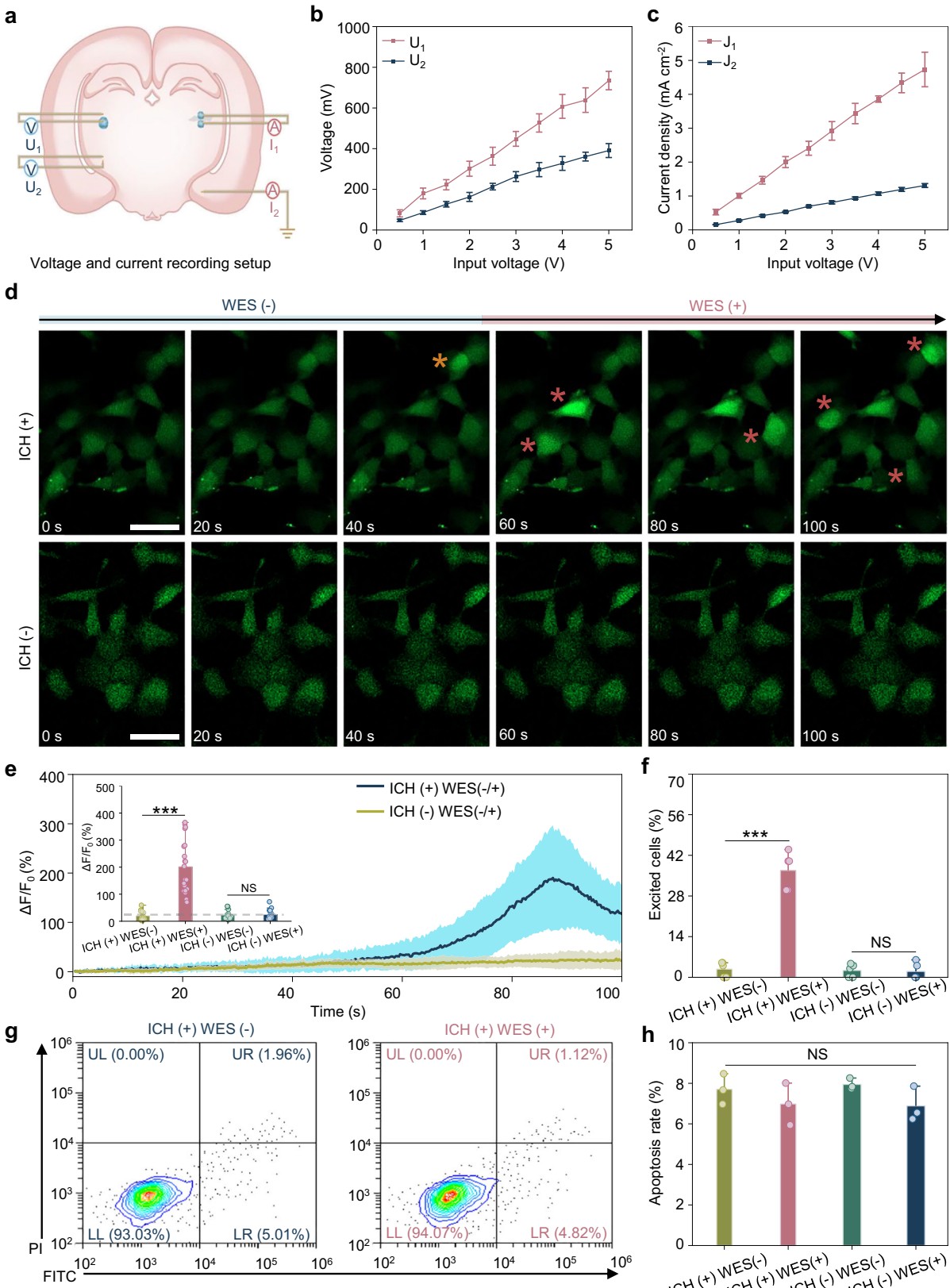

Fig. 17). Electrical signals (wave frequency of 5 MHz, pulse width of 60 μs, pulse repetition rate of 130 Hz, and amplitude of ±5 V) generated by ICH in ex vivo brain tissue were transmitted via copper interconnects to ITO electrodes to deliver 50 s of stimulation. To clarify the respective roles of the ICH and the applied field, calcium imaging was performed under four experimental conditions: ICH(+) WES(-), ICH(+) WES(+), ICH(−) WES(−), and ICH(−) WES(+). Significant fluorescence transients were detected exclusively in the ICH(+) WES(+) group, exhibiting a stimulation-dependent intensity increase (Fig. 2d) with $\Delta F/F_0$ peaking at ~200% and declining thereafter (Fig. 2e). Quantitatively, the ICH(+) WES(+) group exhibited an eightfold increase in calcium response compared to controls, with ~45% of SH-SY5Y cells

**Fig. 2 | ICH-mediated wireless neural stimulation in vitro. a** Schematic of localized electric potential and current density measurements. **b** Output voltages ($U_1$, $U_2$) scale linearly with input voltage ($n = 5$ independent experiments). Electrode depth relative to the external electrode was matched. The difference between conditions was the presence or absence of ICH. **c** Current density ($J_1$, $J_2$) increases proportionally with input voltage, showing enhanced current focusing at $J_1$ ($n = 5$ independent experiments). **d** Time-lapse calcium imaging of SH-SY5Y cells under different conditions. Red asterisks indicate activated cells. Scale bar, 50 μm. **e** Representative normalized fluorescence changes relative to baseline ($\Delta F/F_0$) show stimulation-dependent fluorescence changes in the ICH (+) wireless electrical stimulation (WES) (+) group. The inset highlights $\Delta F/F_0$ across groups ($n = 20$ cells from 5 independent experiments). **f** Quantitative comparison of excited cells across all conditions, showing that cellular activation occurs exclusively in the ICH (+) WES (+) group ($n = 5$ independent experiments). **g** Flow cytometry analysis of PC12 cell viability with/without WES. PI, propidium iodide; FITC, fluorescein isothiocyanate. **h** Apoptosis rate of PC12 cells across all groups ($n = 3$ independent experiments). Data are presented as the mean ± standard deviation in (**b**, **c**, **e**, **f**, **h**) and were analyzed by one-way ANOVA first, followed by the Tukey's post hoc test in (**e**, **f**, **h**) (two-sided). ***$P \le 0.001$. NS, not significant. Groups are denoted as ICH/WES (±/±). $p_1$ (+/+ vs +/−), $p_2$ (−/− vs +/−), $p_3$ (−/− vs +/+), $p_4$ (−/+ vs +/−), $p_5$ (−/+ vs +/+), $p_6$ (−/+ vs −/−). **e** $p_1 = 0.0001$, $p_2 = 0.9991$, $p_3 = 0.0001$, $p_4 = 0.9914$, $p_5 = 0.0001$, $p_6 = 0.9986$. **f** $p_1 = 0.0001$, $p_2 = 0.9976$, $p_3 = 0.0001$, $p_4 = 0.9878$, $p_5 = 0.0001$, $p_6 = 0.999$. **h** $p_1 = 0.6760$, $p_2 = 0.9807$, $p_3 = 0.4710$, $p_4 = 0.5972$, $p_5 = 0.9989$, $p_6 = 0.4021$.

activated versus <5% in non-stimulated groups (Fig. 2f). These results confirm that effective neuromodulation arises only from the combined action of the ICH and the applied electrical field.

To evaluate the biosafety of ICH-mediated WES, viability assays, proliferation analysis, and flow cytometry were conducted using PC12 cells cultured on ICH thin films. To examine current-dependent effects on viability, stimulation currents of 0, 200, 400, and 600 μA were applied via capacitive coupling with input voltages of 0, ±2, ±4, and ±6 V, respectively. Confocal imaging (Supplementary Fig. 18a) showed strong Calcein-AM (live) and minimal PI (dead) staining, validating the biosafety of WES at currents up to 600 μA. Quantitative analysis (Supplementary Fig. 18b) further confirmed that the density of live cells remained statistically unchanged across different stimulation currents, indicating negligible cytotoxicity. Cell proliferation under 30 min daily stimulation (130 Hz, 60 μs pulse, ±2.5 V) for 3 days was evaluated using the CCK-8 assay. No significant differences in optical density (OD) at 450 nm were observed between stimulated and control groups (Supplementary Fig. 19), indicating unaffected proliferation. Flow cytometry was then performed to analyze apoptosis and necrosis after 3-day stimulation. When ICH was not present, the WES (−) and WES (+) groups exhibited comparable distributions of live, early apoptotic, and late apoptotic or necrotic cells, indicating that WES alone does not induce cellular stress or cytotoxicity (Supplementary Fig. 20). When ICH was present, apoptosis rates in the ICH (+) WES (−) and ICH (+) WES (+) groups remained comparable, demonstrating that the stimulation delivered through ICH does not compromise cell viability (Fig. 2g, h). These results demonstrate that the stimulation protocol is safe and has minimal impact on cell activity and viability.

## ICH-mediated wireless neuromodulation in vivo

To assess in vivo neural modulation by ICH during WES, 2.5 μL of ICH precursor was injected into the left STN, a glutamatergic region involved in motor function and PD pathology (Fig. 3a). Electrophysiological signals were recorded from the primary motor cortex (M1) and globus pallidus (GPi) using multichannel microfiber electrodes, targeting the STN-GPi-M1 pathway within the basal ganglia-cortical circuit regulating motor function[39]. All recordings were conducted under the same four-group configuration as used in the calcium imaging to validate that the stimulation mechanism observed at the cellular level also operates in vivo. One week after surgery, WES via capacitive coupling was first applied to evaluate thermal safety. Finite element thermal simulations were also conducted under the same stimulation parameters to assess heat distribution within the scalp-skull-brain system (Supplementary Fig. 21a). The results revealed that temperature rise was mainly confined to the external surface, with a maximum increase of ~2 °C at the scalp, while the ICH and surrounding deep brain tissue exhibited negligible heating (<0.3 °C). Infrared thermography further confirmed this trend, showing surface temperature changes consistent with simulation results (Supplementary Fig. 21b, c). These findings indicate that deep brain WES is thermally buffered by surrounding tissues and does not induce detectable internal temperature elevation under the applied stimulation

conditions. Only ICH (+) WES (+) induced neuronal activation in the STN, as evidenced by elevated c-Fos expression relative to pre-stimulation (Fig. 3b, c). In the GPi, WES led to elevated firing rates and increased action potential amplitudes, indicating enhanced neuronal excitability (Fig. 3d). Spectral and power spectral density (PSD) analyses revealed increased power in the 10–50 Hz band (Fig. 3e and Supplementary Fig. 22). These findings suggest that ICH (+) WES (+) activates glutamatergic terminals in the STN, enhancing excitatory signaling to downstream GPi[40,41].

In the M1 region, to assess neuronal activation, we first quantified c-Fos expression in M1 (Supplementary Fig. 23). The results showed no significant difference in c-Fos expression across experimental groups, indicating that the cortical region did not receive a strong direct stimulation in a volume conduction scenario. However, ICH (+) WES (+) induced neurophysiological changes characterized by reduced neuronal excitability. Microelectrode recordings showed post-stimulation decreases in mean spike amplitude and firing rate, indicating transient cortical suppression (Fig. 3f and Supplementary Fig. 24). Temporal firing analysis revealed a reduction in spike counts (Fig. 3g, h), consistent with reduced neuronal synchrony. Spectrogram analysis (Supplementary Fig. 25) demonstrated a broadband decrease in spectral power during WES, reflecting overall suppression of neuronal activity in M1. ISI mapping (Supplementary Fig. 26) further confirmed reduced spiking frequency and longer inter-spike intervals, indicative of decreased excitability and desynchronized cortical firing. These findings show that ICH (+) WES (+) increases signal frequency in the GPi while suppressing spiking activity in M1. By contrast, the other control groups [ICH (+) WES (−), ICH (−) WES (−), and ICH (−) WES (+)] showed no detectable electrophysiological modulation, confirming that effective stimulation required both the ICH interface and external WES. This dual effect mirrors key therapeutic mechanisms of conventional DBS, supporting ICH-mediated WES as a minimally invasive strategy for motor modulation[40,41].

To evaluate chronic biocompatibility, neuroinflammatory responses and glial scarring at the ICH-tissue interface were assessed 4 weeks after implantation. Tungsten wires (500 μm diameter), a rigid neural interface with known chronic inflammatory effects, served as the positive control[42]. Tissue sections were labeled for ionized calcium-binding adapter molecule 1 (Iba-1) (microglial activation marker), and glial fibrillary acidic protein (GFAP) (astrocytic scarring indicator) using immunofluorescence. Mechanical mismatch between rigid tungsten probes and brain tissue likely causes shear-induced micro-trauma and exacerbated glial activation. In contrast, modulus-matched ICHs were expected to reduce interfacial strain and attenuate mechanotransduction-related inflammatory signaling[43] (Fig. 3i). Immunofluorescence imaging (Fig. 3j) showed reduced Iba-1⁺ microglial activation and GFAP⁺ astrocytic encapsulation around ICH compared to tungsten wires. Quantification (Fig. 3k) confirmed significantly lower fluorescence intensities of Iba-1 ($33.6 \pm 6.3\%$ of tungsten control, $p < 0.001$) and GFAP ($21.2 \pm 7.6\%$ of tungsten control, $p < 0.001$) fluorescence intensity for ICH. Additional un-cropped confocal images used for quantification show consistent fluorescence

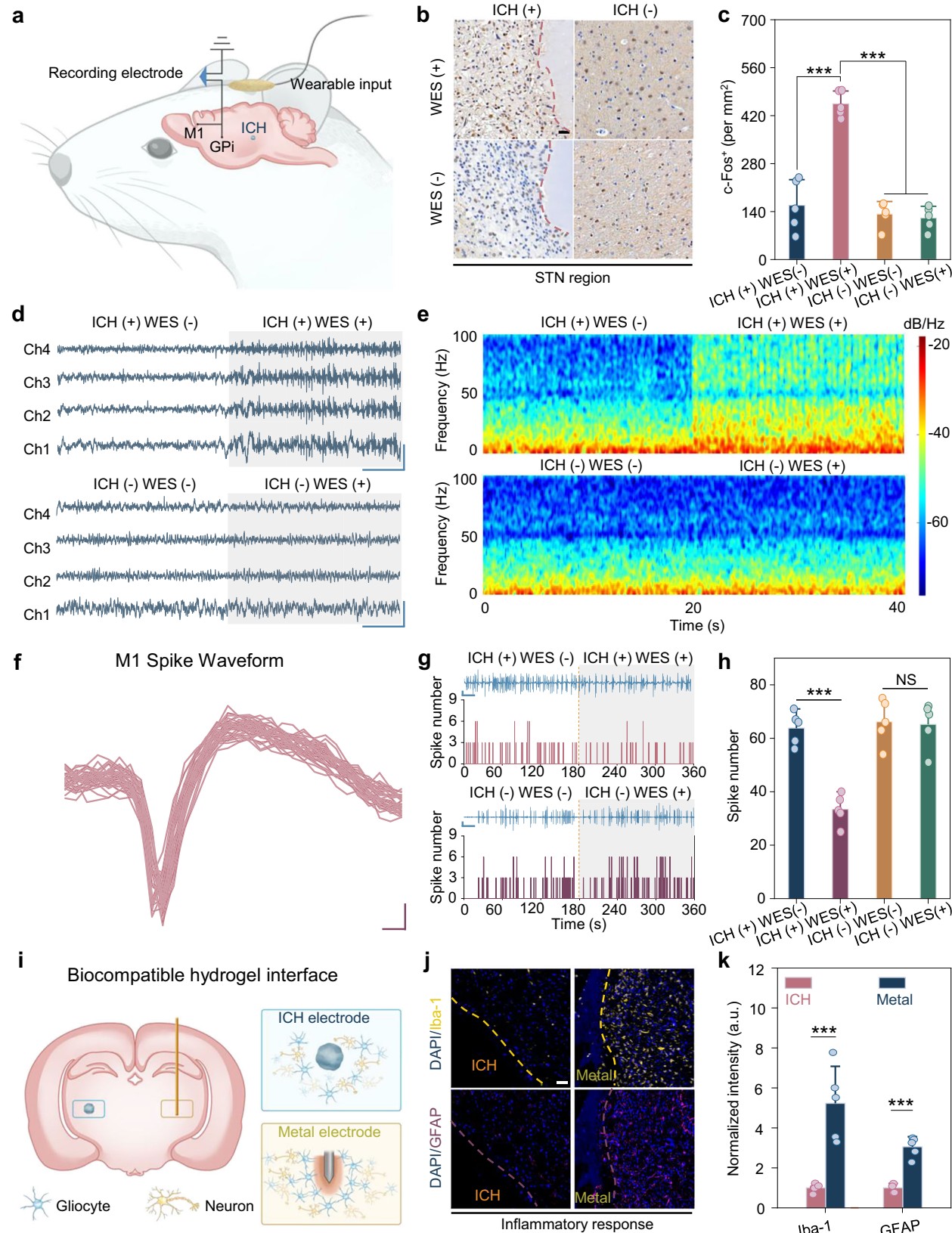

under identical imaging conditions and confirm the reliability of the normalized intensity analysis (Supplementary Fig. 27).

## ICH-mediated wireless DBS treatment for PD rats

To investigate the therapeutic potential of ICH-mediated wireless DBS, we applied stimulation to the STN in a rat model of PD. PD was induced using 6-hydroxydopamine (6-OHDA) injected into the medial forebrain bundle. 6-OHDA is a dopaminergic neurotoxin that induces localized neuronal loss and asymmetric motor deficits following stereotaxic injection[44]. Animals were divided into 4 groups: ICH (−) WES (−) (PD model with no treatment), ICH (−) WES (+) group (WES treatment without ICH), ICH (+) WES (−) (implanted with ICH but without

**Fig. 3 | ICH-mediated wireless DBS of STN in rat brain. a** Experimental setup for wireless DBS, with ICH applied to the subthalamic nucleus (STN) region, while monitoring neural activity in the primary motor cortex (M1) and globus pallidus internus (GPi). **b** Representative images of cellular Fos (c-Fos) expression in the STN region of rats. Scale bar, 100 μm. **c** Quantification of c-Fos intensity in the STN region (*n* = 5 independent animals). **d** Electrophysiological recordings from the GPi region during wireless DBS. **e** Power spectrum analysis of neural activity in the GPi region. **f** Representative electrophysiological waveform from the M1 region. **g** Electrophysiological recordings from the M1 region. **h** Spike number analysis during wireless DBS (*n* = 5 independent experiments). **i** Schematic of the ICH forming a biocompatible and seamless interface with brain tissue. **j** Representative immunostaining images of glial fibrillary acidic protein (GFAP) and ionized calcium-binding adapter molecule 1 (Iba-1) in brain sections with implanted ICH and metal electrodes (tungsten wires with diameters of 500 μm). Scale bar, 50 μm. **k** Normalized fluorescence intensity of GFAP and Iba-1 in the brain tissues around implants (*n* = 5 independent animals). Data are presented as the mean ± standard deviation in **c**, **h**, **k** and were analyzed by one-way ANOVA first, followed by the Tukey's post hoc test (two-sided). $^{***}P \leq 0.001$. NS, not significant. Groups are denoted as ICH/WES (±/±). $p_1$ (+/+ vs +/−), $p_2$ (−/− vs +/−), $p_3$ (−/− vs +/+), $p_4$ (−/+ vs +/−), $p_5$ (−/+vs +/+), $p_6$ (−/+ vs −/−). **c** $p_1 = 0.0001$, $p_2 = 0.6370$, $p_3 = 0.0001$, $p_4 = 0.8432$, $p_5 = 0.0001$, $p_6 = 0.9813$. **h** $p_1 = 0.0001$, $p_2 = 0.9539$, $p_3 = 0.0001$, $p_4 = 0.9901$, $p_5 = 0.0001$, $p_6 = 0.9963$. **k** Iba-1: p = $3.3646 \times 10^{-5}$ (Metal vs ICH); GFAP: $p = 9.6183 \times 10^{-4}$ (Metal vs ICH).

WES stimulation), and ICH (+) WES (+) group (ICH-mediated DBS treatment). Rats received 30 min of WES per day, and locomotor activity was assessed weekly using open field testing (Fig. 4a). MRI scans were conducted at weeks 0 and 4 to evaluate brain structure and function post-treatment. Histological analysis was performed after 4 weeks to assess neuronal activation and potential neuroprotection.

Behavioral assessments at week 0 showed no significant differences among groups, confirming comparable pre-operative functional status (Fig. 4b and Supplementary Fig. 28). After stimulation, ICH-mediated DBS improved locomotor activity, as indicated by increased movement distance, maximum speed, and active time (defined as the cumulative duration during which instantaneous velocity exceeded 5 cm s$^{-1}$). In contrast, the ICH (+) WES (−) and ICH (−) WES (+) groups showed no significant improvement in locomotor activity (Fig. 4c–e and Supplementary Movies 1–5). As the essential role of ICH in mediating WES has already been validated at both cellular and acute animal levels, we focused on evaluating the therapeutic outcomes of ICH-mediated wireless DBS in the PD model rather than repeating additional control experiments. Normal rats (control) were used as healthy controls for comparison. We next evaluated neuroprotective effects of ICH-mediated DBS on dopaminergic neurons in the substantia nigra[45]. Immunofluorescence of the substantia nigra pars compacta (SNc) revealed clear differences between ICH (+) WES (+) and ICH (−) WES (+) groups, with normal rats as controls (Fig. 4f, and Supplementary Figs. 29 and 30). NeuN$^+$ neurons were more abundant and uniformly distributed in the ICH (+) WES (+) group, whereas the ICH (−) WES (+) group showed reduced neuronal density (Supplementary Fig. 29). Iba-1$^+$ microglia were sparse across all groups, with no significant difference, likely to reflect the transient nature of microglial activation during acute neuroinflammation (Supplementary Fig. 30).

GFAP$^+$ astrocytes were more prominent in the ICH (+) WES (+) group (Supplementary Fig. 31). Quantification revealed a strong positive correlation between GFAP$^+$ area and NeuN$^+$ neuronal density in the ICH (+) WES (+) group, indicating an association between astrocyte activation and neuronal survival (Fig. 4g). In contrast, no significant correlation was observed in the ICH (−) WES (+) group. Immunohistochemistry revealed reduced brain-derived neurotrophic factor (BDNF) expression in the ICH (−) WES (+) group, indicating diminished neurotrophic support (Supplementary Fig. 32a). By comparison, BDNF levels were elevated in the ICH (+) WES (+) group following DBS, as evidenced by more intense staining relative to the ICH (−) WES (+) group (Supplementary Fig. 32b). No correlation was observed between GFAP$^+$ area and BDNF in the ICH (−) WES (+) group, whereas a strong positive correlation was found in the ICH (+) WES (+) group (Fig. 4h). These findings suggest that GFAP$^+$ astrocytes may contribute to sustained neurotrophic support through BDNF, reflecting a potential shift toward a neuroprotective phenotype[46,47].

Dopaminergic integrity in the substantia nigra pars compacta (SNc) was assessed by tyrosine hydroxylase (TH) immunolabeling (Fig. 4i). As expected, TH$^+$ neurons were significantly reduced in the SNc of the ICH (−) WES (+) group, reflecting PD-associated neurodegeneration. In contrast, ICH-mediated DBS preserved TH$^+$ neurons in the SNc, likely through STN activation (Fig. 4j). This neuroprotective effect suggests that ICH-mediated DBS not only modulated STN activity but also enhanced dopaminergic release and transmission in the SNc. By mitigating neuronal loss and supporting dopaminergic function, this approach may help restore dysfunctional neuronal circuits within the basal ganglia-cortical loop, offering therapeutic potential for PD.

## MRI compatibility of ICH bioelectrostimulator

MRI is a powerful tool for tracking PD progression and assessing therapeutic effects[48–50]. Conventional metal electrodes are MRI-incompatible due to safety risks, including RF-induced heating and magnetic displacement, as well as imaging artifacts that obscure brain anatomy[51]. In contrast, the hydrogel-based composition of the ICH offers tissue-like magnetic susceptibility, minimizing artifacts and enabling safe, high-resolution MRI during DBS. Therefore, MRI analysis was conducted primarily as a proof-of-concept demonstration to verify the compatibility of ICH bioelectrostimulator. High-resolution 7 T imaging was used to acquire pre- and post-treatment scans from the same animals, enabling assessment of ICH-mediated DBS-induced functional changes. Coronal slices between the AP coordinates of −4.8 mm and −2.16 mm, including the two injection sites, were first analyzed (Supplementary Fig. 33a). Neither PD modeling nor ICH injection induced visible artifacts in Echo Planar Imaging (EPI) or T2-weighted scans (Supplementary Fig. 33b), confirming structural integrity and enabling subsequent functional analysis. Whole-brain functional connectivity analysis (59 × 59 matrix) revealed widespread changes after ICH-mediated DBS, with warm colors indicating strengthened and cool colors indicating weakened connections (Supplementary Fig. 34a). These changes reflect dynamic reorganization of brain networks during DBS therapy. Notably, specific interregional connections were significantly modulated (Supplementary Fig. 34b), demonstrating that ICH-mediated DBS reshapes pathological connectivity patterns in the PD brain.

Multilevel analysis was performed to identify region-specific responses in key motor-related areas, including the striatum (Str), motor cortex (Mtr), cingulate cortex (Cg), prefrontal cortex (PL), thalamus (THL), and midbrain (Raphe)[52,53]. Following DBS, some connections (e.g., Str1 and THL1) showed decreased activity, while most others (e.g., PL2 and Raphe2) exhibited increased activity (Fig. 5a). Red and blue squares in the functional connectivity difference matrix represent regions where modulation patterns have changed significantly, highlighting differential rather than uniform effects and illustrating the proportion of regions with weakened (red) or enhanced (blue) activity which were roughly balanced. These results suggest that ICH-mediated DBS helps restore disrupted activity equilibrium in PD (Fig. 5b).

To further characterize ICH-mediated connectivity changes, enhanced links were extracted from the functional matrix and visualized in a 3D brain network model (Fig. 5c). Connectivity coefficients were quantified, revealing individual-level changes before and after DBS and highlighting the modulatory effect of the intervention. The midbrain, particularly Raphe1, showed robust changes and was

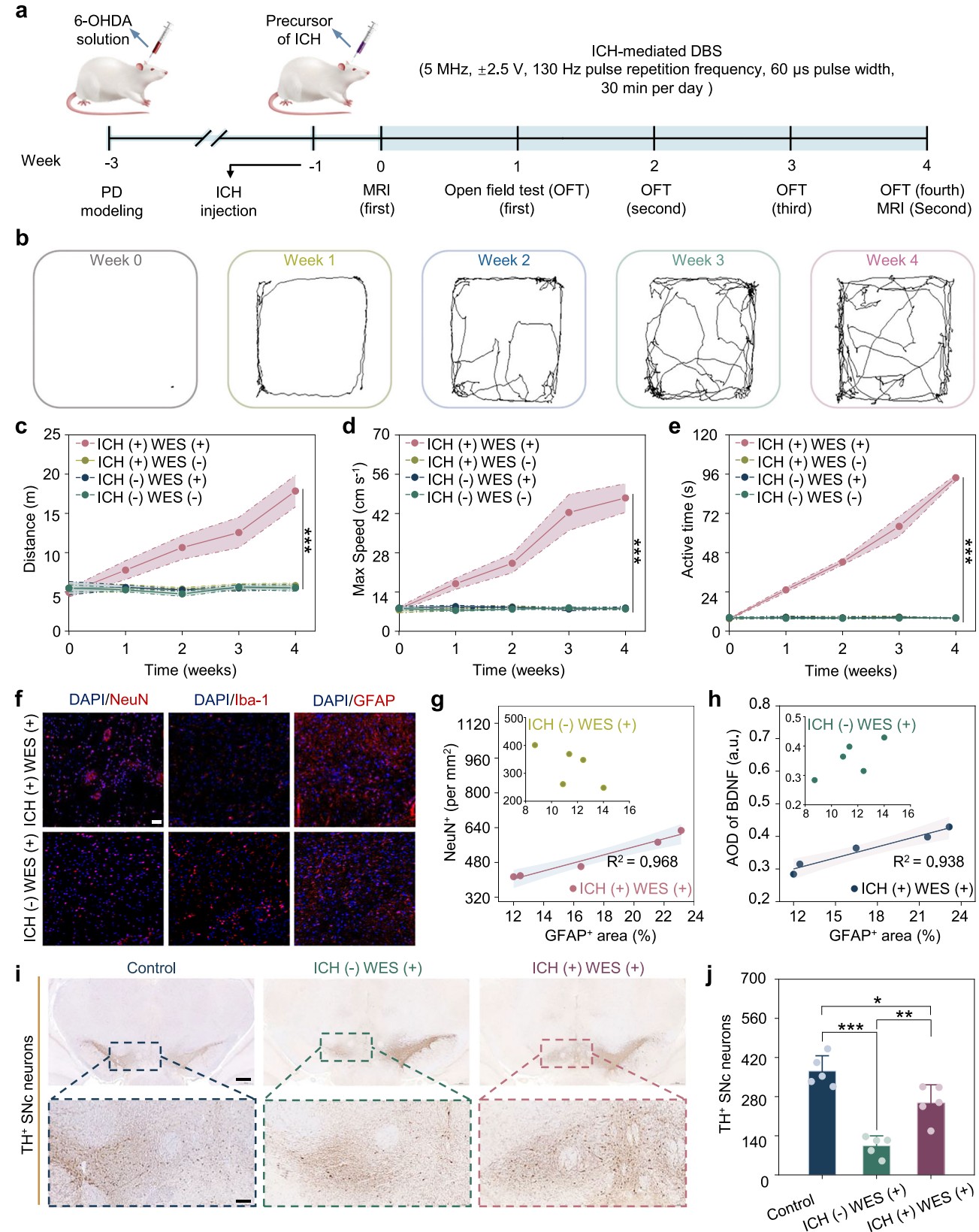

selected as the seed for whole-brain seed-based connectivity analysis. This analysis revealed significantly enhanced connectivity between the midbrain and PD-related regions, including the corpus callosum (CC), thalamus (TH), striatum (Str), and hippocampus (HP) (Fig. 5d). These strengthened connections suggest improved motor pathway integration, consistent with the broader functional reorganization observed in treated animals[54,55].

Building on observed functional connectivity changes, we next assessed gray and white matter alterations using voxel-based morphometry (VBM). High-resolution MRI provided anatomical references for comparisons across treatment groups (Supplementary Fig. 35a). Representative VBM images confirmed preserved structural integrity in both gray and white matter tracts (Supplementary Fig. 35b, c). Significant increases in gray matter volume were observed following ICH-

**Fig. 4 | ICH-mediated wireless DBS alleviates parkinsonian symptoms in PD rats. a** Timeline of in vivo Parkinson's disease (PD) rat experiments, including PD modeling, ICH injection, DBS, MRI, and open-field testing (OFT). **b** OFT trajectories ($n = 5$ independent animals). **c** Total travelled distance ($n = 5$ independent animals). **d** Maximum speed ($n = 5$ independent animals). **e** Active time ($n = 5$ independent animals). **f** Representative immunostaining of neuronal nuclei (NeuN), ionized calcium-binding adapter molecule 1 (Iba-1), and glial fibrillary acidic protein (GFAP) in the substantia nigra pars compacta (SNc). Scale bar, 50 µm. **g** Correlation analysis between GFAP⁺ area and NeuN⁺ cell density in ICH (+) WES (+). Inset: ICH (−) WES (+) ($n = 5$ independent animals). **h** Correlation between GFAP⁺ area and brain-derived neurotrophic factor (BDNF) in ICH (+) WES (+) ($n = 5$ independent animals). Inset: ICH (−) WES (+). Lines in (**g**, **h**) indicate linear regression, with shaded areas representing the 95% confidence interval. **i** Representative immunohistochemical images of tyrosine hydroxylase-positive (TH⁺) neurons. Scale bar, 500 µm (top) and 100 µm (bottom). **j** Quantification of TH⁺ neurons (n = 5 independent animals). Data are presented as the mean ± standard deviation in (**c**–**e**, **j**) and were analyzed by one-way ANOVA, followed by the Tukey's post hoc test (two-sided). $^*P \leq 0.05$, $^{**}P \leq 0.01$, $^{***}P \leq 0.001$. Groups are denoted as ICH/WES (±/±). $p_1$ (−/+ vs −/−), $p_2$ (+/+ vs −/−), $p_3$ (+/+ vs −/+), $p_4$ (+/− vs −/−), $p_5$ (+/− vs −/+), $p_6$ (+/− vs +/+). **c** $p_1 = 1$, $p_2 = 0.0001$, $p_3 = 0.0001$, $p_4 = 0.9692$, $p_5 = 0.9721$, $p_6 = 0.0001$. **d** $p_1 = 0.3493$, $p_2 = 0.0001$, $p_3 = 0.0001$, $p_4 = 0.4557$, $p_5 = 0.9968$, $p_6 = 0.0001$. **e** $p_1 = 0.9917$, $p_2 = 0.0001$, $p_3 = 0.0001$, $p_4 = 0.9734$, $p_5 = 0.9990$, $p_6 = 0.0001$. **j** $p$ (ICH (−) WES (+) vs Control) = 0.0001, $p$ (ICH (+) WES (+) vs Control) = 0.01403, $p$ (ICH (+) WES (+) vs ICH (−) WES (+)) = 0.0017.

mediated DBS (Fig. 6a). Increases were localized to the cerebellum-brainstem (CB-BS) junction and the striatum (Str), regions associated with motor coordination and execution[56]. White matter analysis also revealed widespread volume increases following treatment. Expansion was prominent in the striatum (Str), hippocampus (HP), cingulate cortex (Cg), and prelimbic cortex (PL), which are involved in motor, cognitive, and emotional regulation, suggesting enhanced structural integrity and neuroplasticity[57,58] (Fig. 6b). These findings highlight the neuroprotective potential of ICH-mediated DBS, with coordinated improvements in structural and functional networks in the PD brain.

## Discussion

We developed an injectable conductive hydrogel capable of wireless neurostimulation and demonstrated its application in minimally invasive DBS in a PD rat model. Upon in vivo injection into brain tissue, the non-cytotoxic precursors of ICHs formed a stable conductive network through electrostatic crosslinking between PEDOT:PSS and PPy, with in situ polymerization of pyrrole catalyzed by a GOx-HRP enzymatic cascade using endogenous glucose. The resulting ICHs exhibited a Young's modulus comparable to brain tissue, ensuring biocompatibility, mechanical compliance, low immune response, and seamless neural integration. When a high-frequency volume conduction was introduced into brain tissue via capacitive coupling, impedance mismatch between the ICHs and surrounding tissue induced interfacial polarization. This polarization effect led to charge accumulation at the ICH-brain tissue interface, resulting in localized electric fields and concentrated current density at the implantation site, and thereby enabling effective neural stimulation. With a simple wearable capacitive-coupling setup, the ICH generated programmable stimulation bursts by tuning input power, pulse width, and stimulation frequency, enabling a minimally invasive, fully wireless neuromodulation approach. The efficacy of ICH-mediated stimulation was validated in vitro by calcium imaging and in vivo by c-Fos expression. Electrophysiological recordings during STN stimulation confirmed modulation of basal ganglia circuits, with enhanced glutamatergic output from the STN to the GPi and suppressed cortical activity in M1 via reduced neuronal synchrony. In PD rats, ICH-mediated wireless neuromodulation promoted motor recovery, preserved dopaminergic neurons, and improved functional connectivity across motor circuits, as confirmed by structural and functional MRI. Compared with conventional implanted DBS electrodes, the hydrogel is mechanically soft and shape adaptive, which holds great promise for minimally invasive neural interfacing, and would be applied to neuromodulation therapies for various neurological disorders, including epilepsy, stroke, and neurodegeneration among others.

## Methods
### Fabrication of ICHs
The ICHs were prepared via electrostatic cross-linking of PEDOT:PSS and PPy, and the PPy was synthesized by enzyme-catalyzed in situ polymerization. Specifically, 75 µL of PEDOT:PSS solution (1.0 wt% solid content, Heraeus) was mixed with 0.3 µL of Py monomer (Aladdin) under gentle stirring to ensure homogeneity. To initiate the enzyme-catalyzed reaction, 16 µL of glucose oxidase (GOx, 10,000 µ mL⁻¹, Aladdin), 16 µL of horseradish peroxidase (HRP, 20,000 µ mL⁻¹, Aladdin), and glucose (final concentration of 5 mM) were added to the mixture. The components were thoroughly mixed to achieve a uniform dispersion, and the reaction mixture was incubated at 37 °C until complete gelation occurred.

### Structural characterization of ICHs
The polymerization of PPy through GOx-HRP enzyme cascade in the presence of glucose was validated by UV-vis-NIR absorption spectroscopy (SolidSpec-3700, Shimadzu). FTIR spectra were recorded in ATR mode using a Nicolet iS50R spectrometer (Thermo Scientific) over 400–4000 cm⁻¹. Raman spectra were obtained using a LabRAM HR800 (Horiba Jobin Yvon) equipped with a 532 nm excitation laser at room temperature. Commercial PEDOT:PSS (lyophilized) without any added pyrrole, enzymes or glucose was used as the control sample. The experimental sample was the fully formed ICH obtained by enzymatic polymerization of pyrrole in the presence of PEDOT:PSS. Before FTIR and Raman analysis, ICH samples were thoroughly dialyzed with deionized water and lyophilized to remove residual glucose and other small molecules.

### Zeta potential analysis
To verify the electrostatic interaction between PPy and PEDOT:PSS, zeta potential measurements were performed under neutral aqueous conditions. PPy and PEDOT:PSS dispersions were prepared following the same procedures as ICHs, except that only one conductive component (either pyrrole or PEDOT:PSS) was included in each preparation. The resulting suspensions were diluted with deionized water to achieve a count rate of 200–400 kcps before measurement. Zeta potential values were determined using a Zetasizer Nano ZS90 (Malvern) at 25 °C.

### Rheological properties and injectability of ICHs
The rheological properties of the ICHs were examined to evaluate their injectability and mechanical performance. Measurements were conducted at 37.5 °C using a rheometer (MCR102, Anton Paar). Time sweeps tests at a constant frequency of 1 Hz and 5% strain tracked the evolution of the storage modulus ($G'$) and loss modulus ($G''$). And the Young's modulus ($E$) was calculated using Eq. (1):

$$E = 2\sqrt{G'^2 + G''^2} \cdot (1 + \upsilon) \tag{1}$$

where $G'$ and $G''$ represent the storage and loss moduli at 1 Hz, respectively, and $\upsilon$, representing Poisson's ratio, is assumed to be 0.5.

The gelation point was defined as the crossover of these two values. After gelation, frequency sweeps ranging from 0.1 to 10 rad s⁻¹ were used to assess the hydrogel's viscoelasticity and

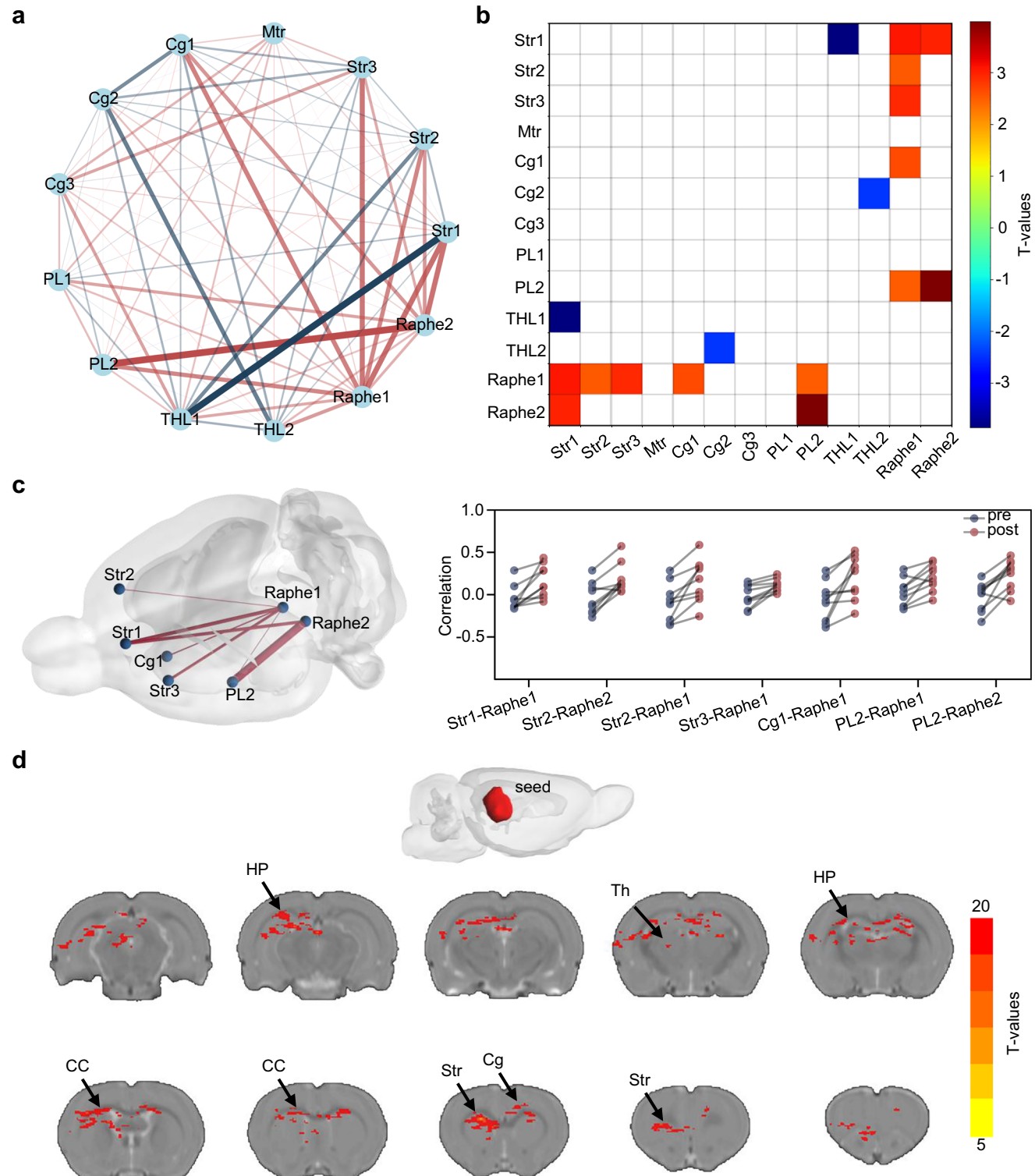

**Fig. 5 | ICH-mediated wireless DBS modulates brain functional connectivity.** **a** Connectivity network analysis showing the relationships between different brain regions, with significant positive and negative correlations indicated by red and blue edges, respectively. Str, striatum; Mtr, motor cortex; Cg, cingulate cortex; PL, prelimbic cortex; THL, thalamus; Raphe, raphe nuclei. **b** Heatmap of T-values representing the strength of connectivity between brain regions, with color intensity reflecting the T-value. **c** A 3D brain model showing significantly changed connections before and after treatment. **d** Seed-based connectivity analysis showing significant connectivity to the Raphe (red areas), with the color gradient indicating increasing T-values. Th thalamus, CC corpus callosum, HP hippocampus.

## Electrical characterization of ICHs

Electrical conductivity of ICHs was measured using a four-point probe instrument (KDY-1, Kunde Semiconductor Co., Ltd). To evaluate the contribution of individual components, precursor mixtures without pyrrole monomers were also tested, including PEDOT:PSS alone,

structural stability. Injectability was tested by evaluating whether the ICH precursors could smoothly pass through a 25 G microsyringe and disperse within a brain tissue model. To simulate the mechanical environment of neural tissue, a 0.5% (w/v) agarose gel (Macklin) was employed.

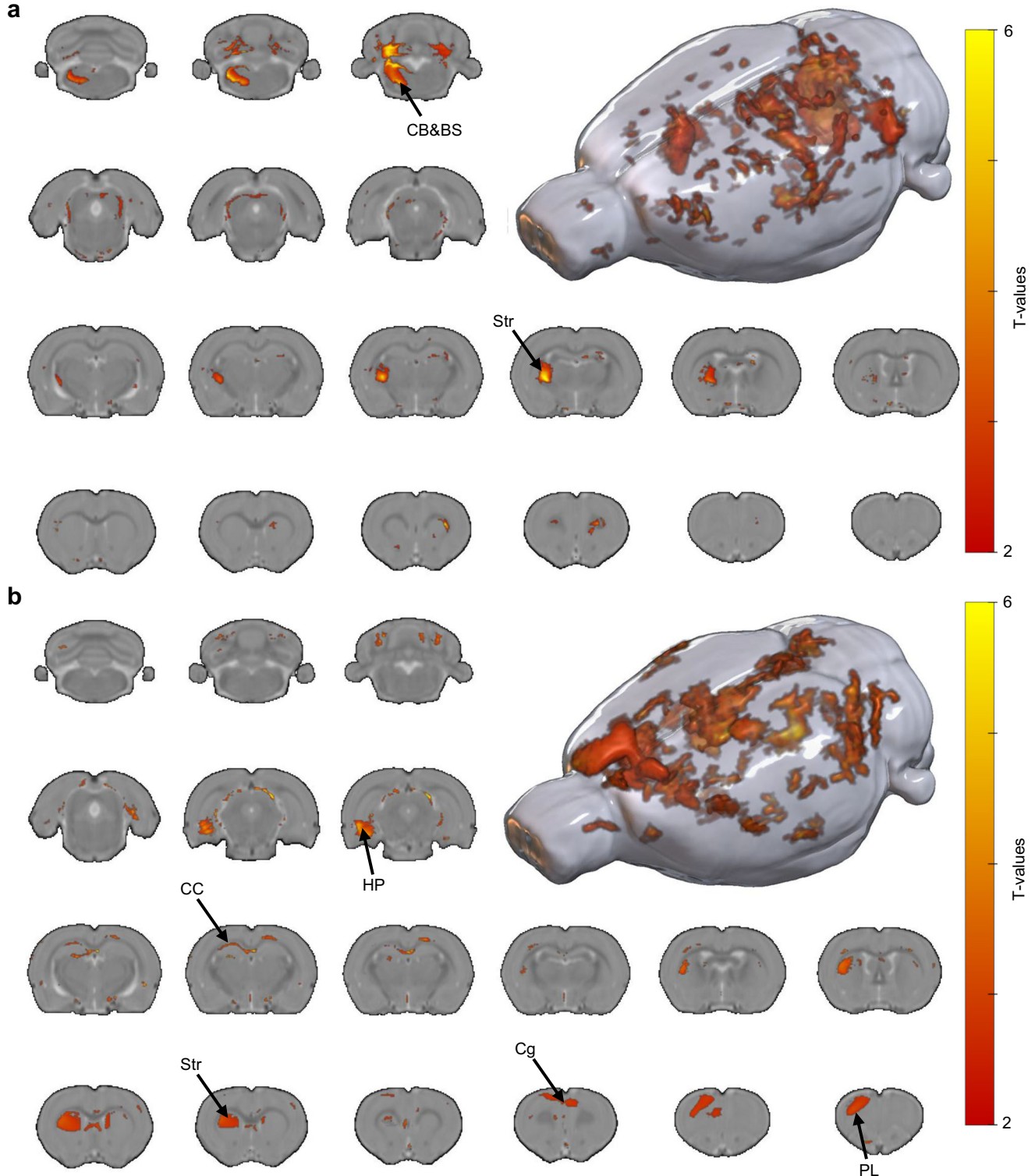

**Fig. 6 | VBM analysis of gray and white matter volume changes induced by ICH-mediated wireless DBS. a** Gray matter volume changes following ICH-mediated wireless DBS in the PD model revealed significant increases in regions such as the cerebellum (CB), brainstem (BS), and striatum (Str), suggesting treatment-induced structural restoration and neuroplastic remodeling. **b** White matter volume changes following ICH-mediated wireless DBS revealed widespread increases in multiple regions, including the hippocampus (HP), corpus callosum (CC), Str, cingulate cortex (Cg), and prelimbic cortex (PL), indicating structural restoration and treatment-induced neuroplasticity. These structural changes reflect the potential neuroprotective and circuit-level remodeling effects of wireless DBS. The color gradient indicates increasing T-values.

PEDOT:PSS with GOx/HRP, PEDOT:PSS with glucose, and PEDOT:PSS with both GOx/HRP and glucose. These precursor systems did not undergo polymerization and served as controls. The prepared samples were lyophilized into a foam-like structure, compressed into thin films, and cut into rectangular sheets (10 mm × 2 mm × 0.1 mm) for testing. The conductivity ($\sigma$) was calculated according to Eq. (2):

$$\sigma = \frac{L \times I}{W \times H \times U} \tag{2}$$

where $I$ denote the current, $U$ is the voltage, and $L$, $W$, and $H$ are the length, width, and the height of the tested samples, respectively.

## Electrochemical characterization of ICHs

The electrochemical impedance spectroscopy (EIS) of the ICHs was measured using a CHI660E electrochemical workstation (Shanghai Chenhua Apparatus Corporation). Cyclic voltammetry (CV) and charge injection capability (CIC) were assessed using a multichannel potentiostat (CHI1040C, Shanghai Chenhua Apparatus Corporation). All measurements were conducted in 0.1 M PBS using a conventional three-electrode configuration. For specimen preparation, 5 μL of ICH precursor solution was drop-cast onto a gold electrode (3 mm diameter), evenly dispersed, and cured to form stable films. Prior to testing, the sample was equilibrated in PBS.

EIS was performed by applying a 5-mV sinusoidal voltage with 0 V DC bias across a frequency range of $10^{-1}$ to $10^5$ Hz. CV measurements were carried out within a potential window of −0.5 V to 0.5 V at a scan rate of 50 mV s$^{-1}$. The charge storage capability (CSC) was calculated from the CV curves according to Eq. (3):

$$\text{CSC} = \int_{E_2}^{E_1} \frac{i(E)}{2\nu A} \, dE \tag{3}$$

where $\nu$ is the scan rate, $E_1$ and $E_2$ are the potential windows, $i$ is the current at each potential, and $A$ is the area of the electrode, respectively.

The CIC was measured by applying biphasic voltage pulses of ±0.5 V at a frequency of 50 Hz. And it was calculated using Eq. (4):

$$\text{CIC} = \frac{Q_a + Q_c}{A} \tag{4}$$

where $Q_a$ represents the charge passed through the anode, $Q_c$ represents the charge passed through the cathode, and $A$ denotes the electrode area.

## Calculation of interfacial polarization

When an alternating electric field is applied, charges tend to accumulate at the ICH-brain tissue interface due to disparity in their electrical properties. This phenomenon is known as Maxwell–Wagner interfacial polarization. The high conductivity of the ICH ($\sigma_h = 30$ S m$^{-1}$) facilitates rapid charge transport toward the interface, whereas the much lower conductivity of brain tissue (0.27 S m$^{-1}$ at 5 MHz) restricts charge movement, leading to charge accumulation on the ICH side. Meanwhile, the high relative permittivity of brain tissue ($\varepsilon_b = 670$ at 5 MHz) allows it to store substantial amounts of charge, attracting opposite charges to its side of the interface. This contrast in conductivity and permittivity enhances charge separation and strengthens interfacial polarization. The interface behaves like a capacitor, forming an electric dipole that increases the intensity of the local electric field.

The discontinuity in the electric displacement field, caused by the permittivity mismatch ($\varepsilon_b \gg \varepsilon_h$), gives rise to a surface charge density ($\rho_s$), as defined in Eq. (5). The relative permittivity $\varepsilon_h$ was -1.35 (calculated by capacitance measurements):

$$\rho_s = \varepsilon_b \varepsilon_0 E_b - \varepsilon_h \varepsilon_0 E_h \tag{5}$$

where $E_b$ and $E_h$ are the electric field strengths near the interface in brain tissue and ICHs, respectively. Tissue-specific electrical parameters, including conductivity ($\sigma_b$) and relative permittivity ($\varepsilon_b$), were obtained from Sim4Life (www.sim4life.swiss).

The current density remains continuous across the interface, ensuring steady charge flow, and the current continuity at the interface ($J_1 = J_2$, where $J_1$ and $J_2$ represent the interfacial current densities in ICHs and brain tissue, respectively.) leads to the following expression for the interfacial electric field ratio, as shown in Eq. (6):

$$E_h/E_b = (J_1/Y_h)/(J_2/Y_b) = Y_b/Y_h = (\sigma_b + j\omega\varepsilon_0\varepsilon_b)/(\sigma_h + j\omega\varepsilon_0\varepsilon_h) \tag{6}$$

where, $Y_h$ and $Y_b$ are the corresponding admittances of ICHs and brain tissue, respectively. Therefore,

$$E_b = \frac{(\sigma_h + j\omega\varepsilon_0\varepsilon_h)}{(\sigma_b + j\omega\varepsilon_0\varepsilon_b)} E_h \tag{7}$$

where $\omega$ is the angular frequency, $\varepsilon_0$ is the vacuum permittivity ($8.85 \times 10^{-12}$ F m$^{-1}$), and $j$ is the imaginary unit. Then, the surface charge density was calculated as $\rho_s = 0.053\, E_h$, where $E_h$ is in V m$^{-1}$ and $\rho_s$ is in nC cm$^{-2}$. Since $E_h$ at the interface cannot be directly measured with high accuracy, two insulated copper wires were placed at the center and edge of the ICH (radius = 0.85 mm). From the observed potential difference of 280 mV under a 2.5 V, 5 MHz capacitive drive, $E_h$ was calculated as 329 V m$^{-1}$ and thus the estimated $\rho_s \approx 17.4$ nC cm$^{-2}$. Considering potential interfacial polarization between the copper wires and the ICH, the measurement was repeated using insulating PEDOT:PSS micro-fibers at the same locations. The 267 mV voltage drop in this configuration yielded $E_h = 314$ V m$^{-1}$, leading to $\rho_s \approx 16.8$ nC cm$^{-2}$. In both cases, the estimated $\rho_s$ exceeded the theoretical threshold of 15 nC cm$^{-2}$ required for neuron cell activation.

## Calculation of specific absorption rate

The total impedance $Z$ of each tissue layer, namely the scalp, skull, and brain, was calculated according to Eq. (8):

$$Z = \frac{d}{\sigma A} + \frac{1}{\sigma + j\omega\varepsilon_0\varepsilon(A/d)} \tag{8}$$

where $d$ denotes the thickness of each layer. The layer thicknesses were defined as follows: insulation layer (5 μm), scalp (1 mm), skull (1 mm), and brain tissue (10 mm). The tissue area was defined as $A = 0.78 \times 10^{-5}$ m$^2$, corresponding to the coverage area of Au foil transmitter electrode (diameter 10 mm).

The total current ($I$) flowing through the tissue was determined based on the root-mean-square voltage ($V_{rms}$), as described by Eq. (9)

$$I = \frac{V_{rms}}{|Z|} \tag{9}$$

The current density $J$ and electric field intensity $E$ were calculated using Eq. (10) and Eq. (11), respectively:

$$J = \frac{I}{A} \tag{10}$$

$$E = \frac{J}{\sigma + j\omega\varepsilon_0\varepsilon} \tag{11}$$

where $A$ for $J_1$ represents the projected surface area of ICHs (0.0445 cm²), while for $J_2$ computations, $A$ corresponds to the effective coverage area of gold foil-coated tissue quantified as 0.785 cm².

The specific absorption rate (SAR) was calculated using Eq. (12):

$$SAR = \sigma \frac{E^2}{\rho} \tag{12}$$

where $\sigma$ is conductivity of brain tissue, $E$ is electric field strength of brain tissue, and $\rho$ is mass density of brain tissue.

## Finite element modeling

A three-dimensional finite element model was constructed using COMSOL Multiphysics® (v6.3) to simulate electric field distributions within a multilayer system (Supplementary Fig. 12). The geometry consisted of six vertically stacked components: (1) a base rectangular chamber (20 mm × 20 mm × 20 mm) filled with a conductive medium representing brain tissue; (2) a 1 mm-thick rectangular shell simulating the skull; (3) a 1 mm-thick outermost rectangular shell representing scalp tissue; (4) a 1.7 mm-diameter spherical inclusion, mimicking the ICH, positioned 7.5 mm above the grounded base along the central vertical axis; (5) a 12.5 mm-diameter insulating barrier (0.1 mm thick) coating on the tissue scalp layer; and (6) a 10 mm-diameter coaxial gold foil electrode placed atop the insulating layer. Boundary conditions were defined by applying a 5 MHz sinusoidal voltage (±2.5 V amplitude, 0° phase) to the gold electrode, with the chamber's bottom surface grounded (0 V). All other exterior boundaries were electrically insulated. The AC/DC module with the electric currents interface was employed for transient solver analysis. In addition, the Solid Heat Transfer in Solids interface was added, and the model was extended to a coupled multiphysics study by enabling the Electromagnetic Heating coupling so that electromagnetic losses produced by the applied AC fields are passed directly into the heat-transfer simulation. A transient solver was used to compute the electromagnetic and thermal response simultaneously: snapshots of changes in electric field density, current, and charge distribution are shown at 0.06 μs, while the temperature evolution is presented at 30 min. A tetrahedral mesh with local refinement was applied to the ICHs, insulating layer, and gold electrode regions to ensure numerical convergence and solution accuracy. Surface charge density simulation results indicate that the maximum surface charge density on the ICHs approached 20 nC cm⁻², larger than the 15 nC cm⁻² threshold required to initiate neural activation.

## Measurement of the electrical outputs

Brain tissue, skull, and scalp harvested from freshly euthanized adult SD rats were rinsed in phosphate-buffered saline (PBS, pH 7.4) to remove residual blood and connective tissue. The dissected skull and scalp were anatomically reassembled over the cerebral surface, followed by placement of a polyurethane-encapsulated gold foil electrode (10 mm diameter) on the reconstructed tissue complex. Electrical grounding was established at the basal cerebral surface. Voltage and current transmission characteristics were assessed using a sinusoidal signal (±2.5 V output) applied to the gold foil (wearable electrode) using a function generator (Tektronix, AFG3021C) and a power amplifier (ATA-1200B, Aigtek). Voltage was recorded via a parallel circuit using an oscilloscope (DHO1204, RIGOL) across the brain, while current was measured in series using a current amplifier (OE4102, Sine Scientific Instruments) between the brain base and ground. Frequency-dependent responses were evaluated from 100 kHz to 7 MHz.

For ex vivo electrical output measurements of ICH, the cerebral hemispheres were bisected along the central sulcus, and 2.5 μL of ICH was injected into the region at the same depth as the STN in the right hemisphere. Two insulated copper wires with exposed tips were implanted: one within the ICH and the other laterally in the adjacent brain tissue. $U_1$ was recorded using an oscilloscope under 5 MHz excitation at incremental input voltages (0.5–5 V, 0.5 V steps). At the same time, $U_2$ was assessed by inserting electrodes below the ICH maintaining identical inter-electrode spacing parameters to those employed in the $U_1$ measurement protocol. To evaluate current focusing, two ICHs (1.25 μL each) were injected into the right STN region, separated by a 5 μm polyurethane insulation barrier. One insulated copper wire was implanted into each ICH, and $I_1$ was quantified using the OE4102 amplifier under 5 MHz excitation with stepwise voltage increments (0.5–5 V, 0.5 V steps). $I_2$ was measured in series between the cerebral base and ground. $I_3$ was measured between the center of the ICHs and the adjacent brain tissue. $I_4$ was measured as a control for $I_1$, with electrodes placed in brain tissue but separated by an insulating PU membrane, mimicking an interface without ICH-mediated conduction. For comparison, cortical voltage ($U_3$) and current density ($J_5$) were recorded using the same relative electrode depth difference as the deep-brain measurements ($U_2$ and $J_4$). Current densities ($J$, mA cm⁻²) were calculated using Equation 13.

## Cell culture

SH-SY5Y (catalog no. QS-H036, Keycell Biotechnology) cell lines were used for calcium imaging, and PC12 (catalog no. QS-R009, Keycell Biotechnology) cell lines were used for viability/apoptosis analyzes. Cells were maintained in high-glucose DMEM (Solarbio) supplemented with 10% fetal bovine serum (Umedium, Hefei, China) and 1% penicillin-streptomycin (Solarbio) at 37 °C with 5% CO₂. SH-SY5Y cells were seeded onto patterned ITO electrodes and PC12 cells were seeded onto ICH-coated ITO. All cells were cultured until ~80% confluence before further process.

## Electrical stimulation setup

WES was applied via capacitive coupling using a polyurethane-insulated Au foil as the transmitter electrode. Unless otherwise specified, stimulation was delivered at 5 MHz with a pulse width of 60 μs and a repetition rate of 130 Hz. For ex vivo experiments, the entire brain containing the implanted ICH was used as the receiving unit to preserve the ICH-tissue interface, which is essential for capacitive coupling. The brain-ICH complex was connected via a copper wire to the ITO microelectrodes or ICH-coated ITO to complete the external-internal coupling circuit for wireless stimulation.

## Calcium imaging

Microelectrodes were patterned on ITO glass (3 mm width) by HCl etching and insulated with polyimide tape, leaving 3 × 3 mm² exposed tips for electrical contact. The ITO substrate was integrated with a PLA mold and PDMS to form a custom chamber. The exposed ITO tips were connected via Cu wire to the implanted ICH (in the rat's brain) for ex vivo stimulation. SH-SY5Y cells were stained with Fluo-4 AM (37 °C, 30 min), rinsed with PBS, and imaged using confocal microscopy (FV3000, Olympus). Calcium signals were recorded for 50 s at baseline followed by 50 s during WES (5 MHz, ±5 V). Cellular activation was defined as fluorescence intensity changes ($\Delta F/F_0 \geq 50\%$) relative to baseline, a threshold derived from consistent response amplitudes across independent replicates.

## Cell viability and apoptosis analysis

The brain-ICH complex was connected via a copper wire to the ICH-coated ITO. WES was applied to PC12 cells at different current amplitudes (200, 400, and 600 μA, corresponding to ±2, ±4, and ±6 V). Cell viability was evaluated by CCK-8 assay (PARK 10M, TECAN) and live/dead staining using Calcein-AM/PI under confocal microscopy (FV3000, Olympus). The number of Calcein-AM-positive cells was quantified using ImageJ (v 1.5 m) across $n = 5$ independent samples per condition. For apoptosis assays, cells received 30 min WES per day for 3 days (5 MHz, ±2.5 V). After stimulation, cells were collected, washed,

and stained with Annexin V-FITC/PI following the manufacturer's instructions, and analyzed using a flow cytometer (CytoFLEX, Beckman Coulter). Data were quantified based on viable (Annexin V⁻/PI⁻), early apoptotic (Annexin V⁺/PI⁻), and late apoptotic/necrotic (Annexin V⁺/PI⁺) populations. For flow cytometry analysis, the ICH (−) WES (−) and ICH (−) WES (+) control groups were tested using the same procedure, except that the Cu wires were directly implanted into normal brain tissue without ICH while all stimulation parameters remained identical to those used in the ICH (+) groups.

### ICH-mediated neuromodulation in vivo
The SD rats (male, ~200 g, 7-week-old) were purchased from Hubei Medical Laboratory Animal Center and raised in an SPF barrier system. The rats were fasted for 12 h and anesthetized using 1% pentobarbital sodium (30 mg kg⁻¹) before surgery. The anesthetized rats were secured in a stereotactic apparatus (68025, RWD) throughout the surgery. The ICH was implanted unilaterally into the STN (in the left hemisphere) (anteroposterior (AP): −3.5 mm, mediolateral (ML): −2.5 mm, dorsoventral (DV): −7.7 mm from dura) of the rats. Specifically, 2.5 μL of ICH precursor was injected through a 25 μL micro-syringe at a rate of 0.5 μL min⁻¹, and the micro-syringe was maintained in place for 5 min before retracting.

One week after ICH injection, rats were divided into 2 groups (n = 5 per group), one without WES and another one with WES. Two control groups (n = 5 per group) were included for comparison: rats receiving WES stimulation without ICH implantation (ICH(−) WES(+)) and rats without ICH implantation and without WES stimulation (ICH(−) WES(−)). Each rat in the WES group was implanted with 8-channel microwire electrodes (KD-MWA-8, Kedou Brain computer technology, China) in the primary motor cortex (M1; AP: 1.8 mm, ML: −2.5 mm, DV: −2.5 mm) and the internal segment of the globus pallidus (GPi; AP: −1.2 mm, ML: −3.2 mm, DV: −7 mm). These electrodes were fixed to the skull using dental cement, and the additional ground wire connected to the screws for reference. These implanted microwire electrodes served for electrophysiological recording to confirm neural modulation induced by wireless electrostimulation 1 week after implantation. The input stimulation burst was applied by wearable capacitive-coupling setup, with parameters set to a wave frequency of 5 MHz, a pulse width of 60 μs, a pulse repetition rate of 130 Hz, and an amplitude of ±2.5 V. The electrophysiological signal was recorded before, during, and after ICH-mediated WES using a multi-channel electrophysiological recording system (RHS Stim/Recording System, Intan Technologies). The collected electrophysiological data were analyzed using NeuroExplorer (Nex Technologies, v5) and Offline Sorter (Plexon, v4.7.3). Spike sorting and LFP analysis were performed to assess neuronal activity changes in response to the stimulation.

To evaluate the effectiveness of ICH-mediated neuromodulation, Rats were euthanized 90 min after 30 min WES, and then the brains were immediately collected. Collected brain tissue samples (n = 20 rats in total) were fixed in 4% paraformaldehyde, dehydrated, embedded in paraffin, and sectioned for further analysis. The immunohistochemical staining of c-Fos (M00297-6, Bosterbio, dilution 1:250) was employed to evaluate neural activity in the STN regions. One section was collected from each rat, and one representative image was acquired per section for quantification. ImageJ software was employed for counting the c-Fos⁺ neural cells.

### Evaluation of thermal effects
To evaluate the thermal effects induced by WES in vivo, a thermal monitoring setup was established as shown in Supplementary Fig. 21. An adhesive polyurethane (PU) patch integrated with Au electrode was affixed to the scalp of anesthetized Sprague-Dawley (SD) rats, with the electrode positioned above the subthalamic nucleus (STN) injection site. The input and grounding terminals were connected to an external power source, which delivered stimulation at a frequency of 5 MHz, a

pulse width of 60 μs, a repetition rate of 130 Hz, and an amplitude of ±2.5 V. Real-time surface temperature was recorded using an infrared thermographic imaging system (E5 Pro, FLIR) at baseline and at 10, 20, and 30 min after the onset of stimulation to capture any heat accumulation at the electrode-tissue interface.

### Biocompatibility of ICH
To investigate the interface biocompatibility, we implanted the ICH into the STN of rats. And tungsten wires (diameter of 500 μm) were used as positive control. After 4-week implantation, rats were euthanized, and the brains were immediately collected. Brain tissue samples (n = 10 rats in total) were fixed in 4% paraformaldehyde, dehydrated, embedded in paraffin, and sectioned for further analysis. Iba-1 (GB113502, Servicebio, dilution 1:500) and GFAP (GB15100, Servicebio, dilution 1:1000) were used as glial markers for immunofluorescence staining to assess inflammation at the hydrogel-tissue interface in the STN region. DAPI was used to stain cell nuclei. ImageJ software was employed for analyzing their fluorescence intensity. One section was collected from each rat, and one representative image was acquired per section for quantification. ImageJ software was employed for counting the c-Fos⁺ neural cells.

### ICH-mediated DBS for PD treatment
To induce PD in rats, a unilateral injection of 6-OHDA was administered into the medial forebrain bundle. Specifically, 6 μL of 6-OHDA (2 μg μL⁻¹, dissolved in 0.02% ascorbate saline, Sigma-Aldrich) was injected at a rate of 0.5 μL min⁻¹ using a 10 μL micro-syringe. The injection coordinates were AP (from bregma): −4.4 mm, ML: −1.5 mm, and DV: −7.8, −7.9 and −8.0 mm (2 μL for each site). After the injection, the micro-syringe was left in place for 5 min before being retracted. Two weeks after stereotactic localization surgery, the dopaminergic lesion was assessed by the apomorphine-induced rotation test. Briefly, rats injected with apomorphine (0.05 mg kg⁻¹, Sigma-Aldrich) made more than 200 turns contralateral to the lesion side in 30 min were of nearly complete (>90%) DA denervation in striatum, which can be selected as PD model. Rats that passed the screening were divided into 4 groups (n = 5 per group), including:

Animals were divided into 4 groups: ICH (−) WES (−) (PD model with no treatment), ICH (-) WES (+) group (WES treatment without ICH), ICH (+) WES (−) (implanted with ICH but without WES stimulation), and ICH (+) WES (+) group (ICH-mediated DBS treatment). The input stimulation burst was applied by wearable capacitive-coupling setup, with parameters set to a wave frequency of 5 MHz, a pulse width of 60 μs, a pulse repetition rate of 130 Hz, and an amplitude of ±2.5 V. For the implantation of ICH, the precursor of ICHs was injected unilaterally into the STN (in the lesioned hemisphere) of a hemi-PD rat 2 weeks after PD modeling. 2.5 μL ICHs precursor was injected through a 25 μL micro-syringe at a rate of 0.5 μL min⁻¹, and the micro-syringe was maintained in place for 5 min before retracting.

### Behavioral tests
The efficacy of the ICH-mediated DBS was evaluated through open-field testing on PD rats. In a typical test, a rat was placed in a square box measuring 75 cm × 75 cm with a height of 50 cm. The movement of the rat's body center was tracked for 5 min using a digital video camera mounted directly above the arena, in conjunction with smart/video tracking software (SMART 3.0, Panlab). The motor performance of hemi-PD rats across different groups was then compared. Motor behavioral indices, including total trajectory, distance travel, number of movements, and average speed, were recorded and analyzed using smart/video tracking software. Active time was defined as the cumulative duration during which instantaneous velocity exceeded a preset threshold of 5 cm s⁻¹ in the tracking software. During the open-field experiment, conditions were controlled to maintain a temperature of ~25 °C, appropriate lighting, and a quiet environment to minimize

external interference with the rats. After each recording, the test area was thoroughly cleaned to remove any excreta and odor left by the previous rat, ensuring consistency and accuracy in subsequent tests.

## Histological investigation of PD rats

Following 4-week ICH-mediated DBS and behavioral assessments at 4 weeks post-lesion, PD rats were deeply anesthetized and transcranially perfused with heparinized saline, followed by 100 mL of 4% paraformaldehyde in phosphate buffer. Brains ($n = 15$ rats in total) were carefully extracted, post-fixed in 4% paraformaldehyde, dehydrated, embedded in paraffin, and sectioned for subsequent analysis. Immunofluorescence staining was performed using NeuN (GB11138, Servicebio, dilution 1:500) as a neuronal marker to assess neuronal viability in the SNc. Glial activation was evaluated using Iba-1 (GB113502, Servicebio; 1:500) and GFAP (GB15100, Servicebio; 1:1000) as microglial and astrocytic markers, respectively. Immunohistochemical staining was conducted to assess tyrosine hydroxylase (TH, GB11181-100, Servicebio; 1:500) and brain-derived neurotrophic factor (BDNF, GB11559-100, Servicebio; 1:200) expression. One section was collected from each rat, and one representative image was acquired per section for quantification. ImageJ software was used to quantify NeuN$^+$ cell counts, the positive staining area of Iba-1 and GFAP, and average optical density (AOD) of BDNF staining. For TH analysis, the Weka segmentation plugin in ImageJ was employed to remove background signals, allowing for accurate sorting and counting of TH$^+$ neurons.

## fMRI data acquisitions

fMRI was performed using a 7T MRI scanner (BioSpin, Bruker) equipped with rat head cooling coil (MRI Cryo Probe, Bruker). All experiments consisted of 9 animals, with one rat excluded due to fixation issues that compromised image quality. Anesthesia was induced with 3–4% isoflurane, followed by an intraperitoneal bolus injection of dexmedetomidine hydrochloride ($0.1\,mg\,kg^{-1}\,h^{-1}$). The rats were secured in the MRI cradle, and the isoflurane concentration was adjusted as needed to maintain light anesthesia based on physiological responses. After induction, dexmedetomidine was continuously infused via a subcutaneous needle at $0.1\,mg\,kg^{-1}\,h^{-1}$, with total infusion duration limited to under 1 h to minimize movement and ensure physiological stability during scanning. A circulating water bath maintained rectal temperature at $37 \pm 0.5\,°C$. Physiological parameters including temperature, heart rate, and respiratory rate were continuously monitored using a small animal monitoring system (Model 1025; Small Animal Instruments Inc.) and a Dräger Infinity Delta monitor. Isoflurane inhalation levels were fine-tuned to maintain a respiratory rate between 75 and 100 breaths per minute.

During image acquisition, global shimming was first performed to optimize magnetic field homogeneity, followed by local second-order shim optimization using the MAPSHIM protocol. Structural images for VBM analysis were acquired using a Turbo-RARE sequence with the following parameters: field of view (FOV) = $20.35 \times 21.00\,mm^2$, matrix size = $256 \times 256$, RARE factor = 8, repetition time (TR) = 6800.24 ms, echo time (TE) = 12 ms, effective TE = 36 ms, averages = 8, flip angle = 180°, slice thickness = 0.5 mm. The total acquisition time was ~29 min. In the core scanning area, local field homogeneity was further refined using a pre-scanned field map. Functional images were obtained using single-shot echo planar imaging (EPI) with the following settings: TR = 2000 ms, TE = 15.32 ms, segments = 1, matrix size = $64 \times 64$, nominal in-plane resolution = $125 \times 125\,µm^2$, slice thickness = 500 µm, 32 slices, with 600 EPI volumes acquired over 20 min.

## fMRI data preprocessing

The fMRI data were processed using various software packages, including AFNI (Analysis of Functional NeuroImages), FSL (Analysis Group, FMRIB, Oxford, UK), and ANTS (Advanced Normalization Tools; v2.4.4) available on GitHub (https://github.com/ANTsX/ANTs),

supplemented by MATLAB scripts (R2019b). Initially, the MRI data was converted to NIFTI format using Bru2Nii (Bruker). This was followed by semi-manual skull striping of T2 images using ITK-SNAP (v4.0.2; http://www.itksnap.org/). Brain extraction and masking were performed using the fslmaths tool in FSL, and EPI images were registered to T2 images using the antsRegistrationSyN.sh script from the ANTS software suite (http://picsl.upenn.edu/software/ants). Image quality was enhanced by applying N4BiasFieldCorrection and DenoiseImage. Subsequently, brain regions were segmented using a study-specific mouse template (https://atlas.brain-map.org/) for group analysis, ensuring the masks were aligned with the EPI images for anatomical consistency. Time series data were corrected for motion and temporal shifts using antsMotionCorr and 3dTshift in AFNI. Finally, all EPI images were spatially normalized to the TMBTA mouse brain template using both linear and nonlinear transformations to ensure anatomical localization, followed by spatial smoothing with a Gaussian kernel set at twice the voxel size (0.4 mm isotropic Gaussian kernel). Time series extraction for regions of interest (ROIs) was then conducted on images registered to the TMBTA standard template using the 3dNetCorr tool in AFNI, setting the stage for subsequent functional network analyzes.

## Functional connectivity analysis

Functional connectivity analysis was conducted on a set of 59 regions-of-interest, which were manually merged based on the SIGMA Rat Brain Templates and Atlases (https://www.nitrc.org/projects/sigma_template, v1-2-1) using FSL. The ROIs are listed in Supplementary Table 1. For each ROI, an average time series was generated by averaging the smooth fMRI data across all voxels within that ROI using the 3dNetCorr tool. Connectivity matrices were constructed by calculating Pearson's correlation coefficients between all ROIs ipsilateral to the stimulation site, resulting in a $59 \times 59$ matrix for each animal. To identify significant connectivity differences between groups, two-sample t-tests with family-wise error (FWE) correction were performed on the matrices, and brain region pairs with significant differences ($p < 0.05$) were identified. These significant brain region pairs were visualized using chord diagrams or BrainNet Viewer (https://www.nitrc.org/projects/bnv, v1.6) to illustrate the whole-brain network changes. Additionally, bar plots were generated for each significantly altered brain region pair, showing the distribution of connectivity values along with the t-test results.

## VBM analysis

The VBM-DARTEL method was employed to measure cortical volumes in rat brains. High-resolution T2-weighted MRI images were resized to match the SIGMA Rat Brain Templates and Atlases. The images were segmented into gray and white matter using the unified segmentation methods in SPM. The DARTEL algorithm was then applied for precise nonlinear normalization, creating group-specific templates. After normalization, a 250 µm Gaussian smoothing kernel was used to improve data reliability. Cortical volumes of specific brain subregions were calculated based on a standard atlas, and statistical analyzes were performed on gray and white matter separately using T-tests in SPM. Group differences with significant differences ($p < 0.05$) were identified, and the significant voxels were visualized in the coronal plane to highlight regions of interest.

## Statistical analysis

All statistical analyzes, except for MRI-related analyzes (which were conducted using SPM, as described in the corresponding sections above), were performed using Origin 2023 software. The statistical analysis was conducted at least three times ($n \geq 3$) for each sample by one-way ANOVA first, and then by the Tukey's post hoc test. Data are presented as the mean ± standard deviation. The significance threshold was presented as * for $p \leq 0.05$, ** for $p \leq 0.01$, and *** for $p \leq 0.001$, respectively. NS not significant.

## Ethics

Every experiment involving animals have been carried out following a protocol approved by an ethical commission (Institutional Animal Care and Use Committee of Huazhong University of Science and Technology, TJH-202212040). Only male rats were used to minimize variability associated with hormonal cycles and estrogen-related neuroprotection, which can influence dopaminergic signaling and behavioral outcomes in females[59,60].

## Reporting summary

Further information on research design is available in the Nature Portfolio Reporting Summary linked to this article.

## Data availability

All data supporting the findings of this study are available within the article and its supplementary files. Any additional requests for information can be directed to, and will be fulfilled by the corresponding authors. Source data are provided with this paper.

## Code availability

The custom pipeline used for fMRI data preprocessing and functional connectivity analysis in this study integrates established software packages and does not involve core analytical algorithms. The code is therefore available from the corresponding author upon reasonable request.

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

## Acknowledgements

This work was supported by the funds from the National Natural Science Foundation of China under grant No. 32471387 (awarded to Zhiqiang Luo) and No. 325B2052 (awarded to R.S.), and by the Ministry of Science and Technology of China under grant No. 2023YFF0714204 (awarded to J.W.). We would like to thank ZMT ZurichMedTech AG for providing Sim4Life software.

## Author contributions

Zhiqiang Luo, J.W., and C.Y. supervised the project. M.Y., W.L., P.C., Zhuang Liu, and R.S. designed the ICH and experiments. M.Y., W.L., P.C., N.Y., and Zhikun Li conducted fabrication and testing of materials. M.Y., W.L., Q.W., Bingqing Xue, and Dingke Zhang conducted the in vitro experiments. M.Y., R.S., C.G., and J.S conducted the in vivo rat experiments. M.Y. and Zhuang Liu conducted MRI experiments. M.Y., W.L., P.C., Zhuang Liu, R.S., C.Y., J.W., and Zhiqiang Luo prepared the manuscript. M.Y., W.L., Zhuang Liu, R.S., and Baochun Xu processed the data and drew the figures. C.M., C.G., Donghui Zhang, and J.F. polished the manuscript. All authors discussed and agreed with the final version.

## Competing interests

The authors declare no competing interests.
