## [Transparent Peer Review file · Nature Communications]

Injectable hydrogel bioelectrostimulator for wireless deep brain neuromodulation

Corresponding Author: Professor Zhiqiang Luo

Version 0:

Reviewer comments:

Reviewer #1

(Remarks to the Author)

Yang et. al. present a study that aims to develop a novel approach for leadless DBS using capacitive coupling. Although such an approach would be highly attractive for multiple clinical and biomedical applications, the authors did not provide support for the main claim of the manuscript, which is their ability to perform local electrical stimulation using the injectable hydrogel. Instead, the authors provide a detailed characterization of the injectable hydrogel, which does not offer any novelty. As for the novel part of this manuscript, which is the ability to deliver local DBS using the capacitive coupling, the authors did not perform the key control experiment, which is the capacitive electrical stimulation without the injection of the hydrogel. Instead, they provide a single control that is the injection of the hydrogel (for both control and experimental group), with or without electrical stimulation. This only proves that performing capacitive electrical stimulation affects the brain in several different ways (as detailed throughout the manuscript), but does not provide any evidence that the presence of the injectable hydrogel has any contribution whatsoever. The only exception was figure 4, in which the authors report some but not all required negative results.

Thus, my recommendation is that this manuscript will be rejected, and reconsider only if the authors redo all the experiments with the proper controls (i.e.- with ICH and with stimulation, with ICH and without stimulation, without ICH and with stimulation, without ICH and without stimulation).

Comments on the manuscript-

1. Fig. 1e and SI fig. 11: the results in this model do not make sense. Since the brain and scalp are conducting and given that the soft metal plate is insulated ("a soft insulated metal plate placed on the scalp acts as the power transmitter") the main voltage drop should be across the insulation of the metal and the charge accumulation should be at both sides of this insulation.

Moreover, the model in SI fig. 11 clearly shows that a large portion of the brain tissue is being subjected to much greater voltage (no data about the more relevant electrical field reported) than the surrounding tissue around the ICH.

The authors claim that "resulting in localized electric fields and concentrated current density at the implantation site," but no experimental data, or even FEM, were provided to support this claim.

2. Fig. 2- No data is provided about the application of WES without the ICH, so the results do not suggest any insights. They just confirmed that electrical stimulation stimulated the cells. It is only shown that capacitive ES can stimulate SH-sy5y cells, but the lack of negative controls means there is no way to understand whether the stimulation is induced by the capacitive ES or by the local effect of the ICH.

Fig. 2a-c and SI fig 15- "These findings confirm that ICH bioelectrostimulator concentrate current density at the ICH-brain interface, which would enable effective, tether-free neural excitation"- this claim is not supported by the reported data. First, the difference between I3 and I4 is not very significant, so they cannot really ensure local specificity. More importantly, the I3 is measured between the center of the ICG to its edge, but this is not where the cells will be located. Instead, to properly make this comparison, authors must compare the edge of the ICG and the adjacent tissue, which is where the stimulated cells are located. Given the small difference between I3 and I4, it seems unlikely there will be a significant difference. The reported safety experiments do not demonstrate the safety of the ICH. authors must present correct negative controls (e.g. with and without ICH and with and without capacitive stimulation). the current results only suggest the safety of the

capacitive ES.

Technical comments- How did the authors determine if a cell is activated? spike? dF/F elevation? What were the criteria? Where is the ICH located? is it in direct contact with the cells? Where is the ITO electrode located? is it in direct contact with the cells? Critical experimental data missing.

3. Fig. 3- once again, no data regarding the application of WES without the ICH, so the only conclusion here is that WES stimulates cells, but there is no indication of any locality or for the contribution of the ICH.

4. Figs. 4- Here, the authors did provide one of the required negative controls ("ICH-BES-free group: PD rats with WES but without ICH bioelectrostimulator implantation"). However, they did not provide any information regarding PD rats without WES but with ICH bioelectrostimulator, which again prevents any scientific conclusion to be made regarding the functionality of the ICH bioelectrostimulator. Moreover, some critical discrepancies regarding the experimental procedure suggests that these experiments were poorly conducted (e.g.- "n = 9 per group" in the experimental section Vs. "n = 5 independent animals" in fig.4's caption).

5. Figs. 5-6. Same as for 2 and 3. Lack of appropriate controls.

Minor-

1. Introduction- the introduction fails to address other electrical transduction technologies such as magnetic induction, piezoelectric materials and ultrasound, and optoelectronics and light, which all has the potential to overcome the limitations described above. authors should explain the advantage of their methodology over these exciting technologies.

2. Authors should be consistent when reporting electrical parameters: current density (J) or current (I).

3. The author's claim: "Confocal imaging (Supplementary Fig. 17) showed strong Calcein-AM (live) and minimal PI (dead) staining, validating WES biosafety at currents up to 600 μ A" is not sufficiently supported. Statistical analysis and the number of samples measured should be provided.

4. Methods section: The description of the cell line and cell handling process is missing.

Reviewer #2

(Remarks to the Author)

Reviewer #3

(Remarks to the Author)

Yang et al. present a new wireless modality for eliciting deep brain stimulation via a hydrogel of PEDOT:PSS and polypyrrole which crosslinks in situ mediated by glucose oxidation. The authors have done a nice job to characterize this material and provide some evidence of its utility for deep brain stimulation in a PD mouse line.

A few general points, the authors are encourage to double check that all acronyms are clearly defined through the manuscript.

The authors are encouraged to add a little more detail into what constitutes an n or m for each experiment. For instance, number of animals, number of epochs, number of tissue slices, number of images taken.

The confocal images for biocompatibility investigations are of poor quality. The authors are encouraged to provide a larger set of images in the supplemental to help qualitatively support their normalized intensity measurements which can very sensitive to differences in staining and imaging methodologies.

The abstract lacks appropriate depth and detail of the results and conclusions of the manuscript.

The authors are encouraged to tone back their claims for DBS treatment in PD animals as open field as a measure of motor control may not be accurate. The authors should justify their inclusion of only male animals.

Data presented in Fig 4 and Fig 5 seems to lack appropriate comparisons to control animals or treatment groups.

Reviewer #4

(Remarks to the Author)

The manuscript presents an injectable bioelectronic strategy for deep brain stimulation (DBS) using conductive hydrogels formed in situ. The approach could have significant implications for next-generation neuromodulation. However, several aspects require clarification or additional evidence (due to the lack of proper controls in many experiments) as outlined below. To fully support the conclusions, the following comments should be addressed:

1. L93. Conductive polymers have been extensively studied in implantable applications as a biocompatible additives. Is there evidence on conductivity-biocompatibility trade-off in conductive hydrogels?

2. L99. What are the limitations associated with other simple biomaterial approaches such as incorporating conductive

polymers (e.g., PEDOT:PSS) into in situ forming hydrogel matrices? The advantage of the bio-catalyzed polymerization of Py in presence of PEDOT:PSS compared to other established methods of fabricating high conductivity hydrogels is unclear.

3. L104. How is the proposed mechanism of stimulation (i.e., interfacial polarization due to impedance-mismatched ICH and surrounding tissues, or charge accumulation at this interface) supported by the presented data? Did the authors test the condition where there is no impedance mismatch as control?

4. L116. It is stated that improving biocompatibility in the in situ polymerized conductive polymers remains challenging due to toxicity of monomer and crosslinker concentration. However, the proposed prepolymer solution contains Py monomers, which is also cytotoxic and can leach into the surrounding tissue. Also, were cells contacted with the gel in the in vitro compatibility data of Fig. 2d,g,h?

5. L119. Have the authors characterized zeta potential of PEDOT:PSS and PPy to conclude the electrostatic assembly in their crosslinking mechanism?

6. L121. Given the small content of conductive polymer in hydrogel formulation, it is important to clarify the roles of these polymers more strongly. Experiments such as conductivity tests (which is not explained in methods section), where high conductivity is reported, study the role of glucose without control hydrogels that have no intrinsic conductivity (with and without HRP/GOx-HRP) to elucidate the roles of conductive polymers in the measured conductivity and thereby its neuromodulation performance.

7. In vitro and in vivo characterizations of Fig. 2 and 3 mostly compare "with" and "without" stimulation. The role of the developed ICH stimulator developed here is thus unclear. Have stimulation without ICP or other injectable hydrogels with no conductive functionality been tested and compared as controls?

8. L129. Please provide UV-vis spectra over the full range up to 1200 nm wavelength to indicate if there is an absorption band suggesting PPy is doped after in situ polymerization and provide ohmic conductivity along with PEDOT:PSS.

9. L138. Can authors explain how the broader absorption bands of C-O-C and C-H are associated with "enhanced intermolecular interactions and network rearrangement"? Are these not present in glucose? How about PEDOT:PSS controls?

10. L194. Can authors elaborate on how localization is increased with ICH conductivity?

11. L401. Young's modulus compared to the brain is concluded, however no experimental evidence was provided in the results and discussion.

12. Minimally invasive wireless electrostimulation has been claimed in the paper while for the in vivo neuromodulation, 8-channel microwire electrodes were implanted as stated in L447.

Reviewer #5

(Remarks to the Author)

Yang and colleagues present an excellent demonstration of conducting polymer hydrogels formed in situ for wireless modulation of brain function and Parkinsons' disease therapy. By targeted injection of precursor cocktails, they utilize the brains own chemistry to drive polymerization and create wireless transducers of AC stimulation fields. They demonstrate that through clever materials and conductivity engineering, they can achieve both biocompatibility as well as bioelectronic therapeutic efficacy. The open field test results (Figure 4b) are particularly compelling! After sufficient revision, as detailed below, I believe this work warrants publication in Nature Communications.

GENERAL REVISION COMMENTS

The paper is quite complicated to read. There is an overabundance of references to supplementary notes and figures, some of which are more essential than others for the flow of the manuscript. At the same time, some experimental setups are hard to understand (such as the brain slice experiments in Figure 1 or the ICH-BES-to-in vitro experiments). These would require more elaboration (and perhaps illustration) to be easily reproduced. Beyond the specific comments below, I believe that the paper needs a thorough revision for flow and rearrangement of SI: some into the main methods section, some integrated into existing figures, some (potentially) removed altogether, and of course some remaining in SI. In particular, the "supplementary notes" are questionably "supplemental". Perhaps a colleague outside the author list who can give a full reading and provide feedback?

SPECIFIC REVISION COMMENTS

1. On page 6, like 130, the authors discuss electrostatic crosslinking between PPy and PEDOT:PSS. Shouldn't both materials be neutral in the as-prepared conditions? Why would one expect them to be electrostatically crosslinked? Is there direct evidence for this electrostatic interaction?

2. Page 6, line 132: Please elaborate on the "characteristic microgel structures". Are these characteristic results for the authors, or a characteristic feature they authors expected to see? What do they imply about ICH formation and the various ICH formulations?

3. The FTIR and Raman experiments were not clear regarding control experiments and experimental conditions. Please provide details and an explanation of conditions.
4. On page 8, line 163, the authors discuss “diminished capacitance” in the EIS data. But then on line 168, they mention enhanced charge storage capacity. The diminished capacitance is surprising since conducting polymers (especially hydrogel-based ones) usually increase electrode capacitance. Please elaborate? Also, were these measurements done with compressed 2D films? If so, how does EIS translate from this 2D system to the expected 3D in vivo condition?
5. Page 10, lines 199-200: This circuit model could use some justification and/or references.
6. Page 10, line 201: conditions of “safe operation” needs a reference.
7. On page 13, line 262 (and Supp Fig 19), the authors discuss WES without inducing detectable thermal effects. Would the effects of a deep brain WES stimulation actually be detectable from an external measurement like this? Maybe a simple finite element simulation could help here to see if the tissue would buffer (hide) deep thermal changes.
8. Page 13, line 267-268: Why to “these findings suggest that WES activates glutamatergic terminals in the STN”? What leads to the authors to this conclusion? And some references here, please.
9. Page 13, lines 274-275: Elaborate on (justify) spectral analysis confirming changed frequency behavior. I don't see an observable difference in Supp Fig 22 before and after 180 s, when WES was initiated.
10. Page 14, line 280: Need references on “mechanisms of conventional DBS”.
11. In general, the manuscript would be significantly strengthened – and made clearer to put in context – if the authors used more literature references comparing their ICH-BES results with conventional DBS. Tissue/immune response; fMRI effects; reorganization of brain networks; neuroprotectivity; etc. Basically, many/all results from the bottom of page 14 could use this context. I believe this is essential to proceed to publication in Nature Communications.
12. Page 20, line 407: The authors state that “a simple wearable” can control their ICH-BES system. Elaborate on the translatability of this technique. How would it look in human patients? Does the control electrode need to be on the brain, or would the scalp suffice?
13. The measurement setup described Figure 2a and in Supplementary Note 2, page 8, around lines 148-152, really needs a better illustration than the tiny blue balls and wires in Figure 2a. Perhaps a zoom in Figure 2a? Or a standalone additional supplementary figure? Furthermore, regarding Figure 2a, where the working (input) electrode placed? This is essential for easily understanding Figure 2b and c.
14. In Supplementary Note 1's sections on electrical and electrochemical characterization of ICHs (SI page 5), they describe determination of conductivity and CSC. Were these measurements done using the 2D “compressed” ICHs? Would one expect that conductivity assessment and CSC would translated to the fully hydrated 3D in vivo ICHs?
15. The finite element modeling model description (Supplementary Note 2, pages 9-10) could use a diagram showing the model. The text description is difficult to follow. Perhaps a new Supp Fig, or slightly more detail in, e.g., Figure 1e.
16. Likewise, the experimental setup with ICH-BES in brain connected to the in vitro setup (Supplementary Note 3, page 11, lines 202-204) could use a schematic drawing.

Version 1:

Reviewer comments:

Reviewer #1

(Remarks to the Author)

The authors have made a significant effort to address the comments raised. Although some of the main issues were indeed addressed, mainly the proper controls in most cases, there are still a few issues that should be addressed.

Comment 1

Figure 1e still shows that although the ICH can generate some local enhancement of the electrical stimulation around it, the top part of the brain (cortex most likely) is still being stimulated. So the claims of targeting specific regions in the brain seem invalid in the current setting. See also in comment 3 below.

Comment 2

In the new revised figure 2a-c, the authors should measure the current between the edge of the ICH and the adjacent tissue. The center of the ICH is irrelevant. Also, to show the effect of the ICH, the exact same location should be measured with and without the ICH, as the J4 is measured farther away from the capacitor, which might be the cause of the lower currents there.

The dF/F results in fig 2d-f do not look anything like neuronal spikes. They appear only as moderate elevations in the baseline Ca level of the cells, with no indication of electrical activity, which should manifest as action potentials. It is not clear that these are neurons, as their morphology does not resemble that of differentiated SH-SY5Y cells and, as mentioned above, they do not exhibit neuron-like electrical activity. Calcium baseline levels can change when experiencing stress, which might be the case here. Fig. 2g-h still do not present the appropriate controls.

Comment 3-

The authors have addressed my concerns regarding the proper controls. However, the authors here look only at the cells around the ICH, while according to fig. 1e, other regions might be activated as well. Thus, to show local activity, it is critical to check these regions (the cortex near the capacitor) for the same activated neurons.

Comment 4-

Minor- Authors should provide the behavioral tests at the beginning, i.e., week 0. It will help appreciate the effect when the score starts at the same level.

The rest of my comments seem to be addressed properly by the authors.

Reviewer #2

(Remarks to the Author)

Reviewer #4

(Remarks to the Author)

The authors have addressed the comments raised by the reviewers and improved the paper accordingly. I recommend the current version for publication.

Reviewer #5

(Remarks to the Author)

Yang and colleagues have provided a comprehensive and very well-stated response to all reviewers' comments and suggestions. Indeed, this review process feels like it embodies the true spirit of peer review: the initial study and results had some major holes (e.g., lack of controls) that were discussed and addressed in the revision.

I feel that my own comments and suggestions were quite satisfactorily addressed. The work now much more clearly supports the claims and conclusions. I believe the manuscript is much stronger after this peer review and is now ready for ACCEPTANCE at Nature Communications.

Version 2:

Reviewer comments:

Reviewer #1

(Remarks to the Author)

Authors response to my comments are satisfactory, and I think the manuscript is suitable for publication. I have only one minor suggestion: Can the authors add to fig 1e the Finite element simulation results of the local electric field and not only the charge distribution? This can help the readers to avoid misinterpretation of the results described here.

Reviewer #2

(Remarks to the Author)

Reviewer's Comments:

Reviewer #1

Summary Comments

Yang et. al. present a study that aims to develop a novel approach for leadless DBS using capacitive coupling. Although such an approach would be highly attractive for multiple clinical and biomedical applications, the authors did not provide support for the main claim of the manuscript, which is their ability to perform local electrical stimulation using the injectable hydrogel. Instead, the authors provide a detailed characterization of the injectable hydrogel, which does not offer any novelty. As for the novel part of this manuscript, which is the ability to deliver local DBS using the capacitive coupling, the authors did not perform the key control experiment, which is the capacitive electrical stimulation without the injection of the hydrogel. Instead, they provide a single control that is the injection of the hydrogel (for both control and experimental group), with or without electrical stimulation. This only proves that performing capacitive electrical stimulation affects the brain in several different ways (as detailed throughout the manuscript), but does not provide any evidence that the presence of the injectable hydrogel has any contribution whatsoever. The only exception was figure 4, in which the authors report some but not all required negative results.

Thus, my recommendation is that this manuscript will be rejected, and reconsider only if the authors redo all the experiments with the proper controls (i.e.- with ICH and with stimulation, with ICH and without stimulation, without ICH and with stimulation, without ICH and without stimulation).

Response: We sincerely thank you for your constructive and detailed feedback. We fully agree that a rigorous evaluation of the stimulation mechanism requires inclusion of all relevant control groups to isolate the contribution of the ICH. In response, we have now added the additional control data, including the groups without ICH with stimulation (ICH(-)WES(+)) and without ICH without stimulation (ICH(-)WES(-)), at both the cellular level, in vivo electrophysiology and behavioral tests. The results consistently demonstrate that in the absence of ICH, WES alone does not elicit detectable neuronal activation, whereas ICH(+)WES(+) produces robust calcium responses in SH-SY5Y cells in vitro, strong c-Fos expression in the STN region and clear modulation of basal ganglia-cortical activity and improved locomotor activity. These findings

directly substantiate the central claim of our study that the ICH is essential for establishing the interfacial polarization required for localized electrical stimulation.

We respectfully note that the comprehensive characterization of the injectable hydrogel was necessary to verify that the enzyme-triggered PEDOT:PSS network possessed the conductivity, mechanical integrity, and injectability required for subsequent *in vivo* stimulation experiments. The purpose of these measures was to ensure functional suitability for further use.

We have revised the Results, Methods, and Figures to include the new control data and expanded the discussion to clarify the experimental rationale and the distinct functional role of ICH in mediating interfacial polarization at the tissue interface. We hope these revisions address your concerns and strengthen the mechanistic interpretation of our findings.

Modifications to the manuscript: We have revised the Results, Methods and Figures to include the new control data and expanded the discussion to clarify the experimental rationale and the distinct functional role of ICH in mediating interfacial polarization at the tissue interface. All revisions are marked in red in the revised version.

Comment #1: *Fig. 1e and SI fig. 11: the results in this model do not make sense. Since the brain and scalp are conducting and given that the soft metal plate is insulated (“a soft insulated metal plate placed on the scalp acts as the power transmitter”) the main voltage drop should be across the insulation of the metal and the charge accumulation should be at both sides of this insulation.*

Moreover, the model in SI fig. 11 clearly shows that a large portion of the brain tissue is being subjected to much greater voltage (no data about the more relevant electrical field reported) than the surrounding tissue around the ICH. The authors claim that “resulting in localized electric fields and concentrated current density at the implantation site,” but no experimental data, or even FEM, were provided to support this claim.

Response: We agree that in a multilayer system consisting of an insulated transmitter electrode, scalp, skull, and brain tissue, the major voltage drop should indeed occur across the external insulation layer. As shown in below Figure (Fig. R1), the simulation results are consistent with this expectation. Our work is aiming to address the challenge to deliver wireless electrostimulation to deep brain, and the neural stimulation observed in our system is not directly caused by the potential drop across the insulation but rather arises from interfacial polarization within the volume conduction scenario under MHz-range alternating electric field excitation. In this volume

conduction scenario, the field propagates through the head via capacitive coupling, and both conduction and displacement currents coexist within biological tissues. With *in vivo* volume conduction, the tissue behaves as a frequency-dependent lossy dielectric medium, and local polarization occurs wherever there is a strong contrast in permittivity or conductivity. The ICH, having high conductivity and low permittivity, and the surrounding brain tissue, with low conductivity and high permittivity, respond differently to the same electric field, forming interfacial dipoles and accumulating charge at their boundary. This dielectric discontinuity amplifies the local electric field and creates localized charge density sufficient to modulate nearby neurons in deep brain.

To substantiate this mechanism, we performed additional finite element simulations (COMSOL Multiphysics) comparing the electric field, surface charge, and current density distributions with and without ICHs and across varying ICH conductivities (shown as Fig. R1). The results clearly show that although scalp, skull, and cortical tissues are inevitably influenced by the applied field, the presence of ICHs implanted in deep tissue substantially enhances the local field and charge density near the interface, and this enhancement increases with higher ICH conductivity. In contrast, in the absence of ICHs, these parameters remain nearly uniform across the deep brain tissue. These results are consistent with theoretical expectations that dielectric and conductivity mismatches induce interfacial charge accumulation and field amplification at the ICH-tissue boundary.

Fig. R1. Finite element simulation of interfacial polarization and field localization at the ICH-tissue interface. **a**, Simulated electric field intensity distribution in the multilayer head model without and with ICHs of different conductivities (3, 10, and 30 S m^{-1}). The introduction of ICHs leads to localized field enhancement around the hydrogel-tissue interface. **b**, Corresponding surface charge density maps reveal charge accumulation at the ICH boundary, which increases with hydrogel conductivity. **c**, Local current density distribution showing enhanced conduction pathways near the implanted ICH region, consistent with interfacial polarization-induced field focusing effects.

Modifications to the manuscript:

(1) Figures:

We have added Fig. R1 as Supplementary Fig.13 in our revised Supporting information, with the corresponding changes marked in red, which reads as follows:

Supplementary Fig. 13. Finite element simulation of interfacial polarization and field localization at the ICH-tissue interface. **a**, Simulated electric field intensity distribution in the multilayer head model without and with ICHs of different conductivities (3, 10, and 30 S m^{-1}). The introduction of ICHs leads to localized field enhancement around the hydrogel-tissue interface. **b**, Corresponding surface charge density maps reveal charge accumulation at the ICH boundary, which increases with hydrogel conductivity. **c**, Local current density distribution showing enhanced conduction pathways near the implanted ICH region, consistent with interfacial polarization-induced field focusing effects.

Fig. 1e has been updated accordingly, which reads as follows:

Fig. 1e. Finite element simulation of surface charge distribution across the scalp-skull-brain multilayered structure, incorporating tissue-specific dielectric properties, demonstrates charge accumulation at the ICH-tissue interface due to interfacial polarization.

(2) Results section

We have also expanded the discussion on page 11 in the revised manuscript to clarify the origin and spatial localization of electric field enhancement and interfacial polarization at the ICH-tissue interface, which reads: “Finite element modeling was then performed to simulate interfacial polarization and field localization in a scalp-skull-brain model with/without ICH (2.5 μ L, \sim 1.7 mm diameter) (Supplementary Fig. 12). At 5 MHz, the introduction of the ICH markedly enhanced local electric field intensity around the implantation site (Supplementary Fig. 13a). The degree of field localization increased with the conductivity of the ICH, indicating stronger polarization and field confinement near highly conductive hydrogels. Correspondingly, the surface charge density at the ICH-tissue interface also increased with conductivity (Fig. 1e and Supplementary Fig. 13b), consistent with enhanced interfacial polarization caused by permittivity and conductivity mismatch between the two media. The resulting charge accumulation intensified local current density (Supplementary Fig. 13c), confirming that higher-conductivity ICHs enhance interfacial polarization and lead to stronger charge focusing at the hydrogel-tissue interface.”.

(3) Methods section

We have also revised the Methods section to provide a clearer description of the finite element modeling procedures, which was marked red and reads as:

“Finite element modeling

A three-dimensional finite element model was constructed using COMSOL Multiphysics® to simulate electric field distributions within a multilayer cylindrical system (Supplementary Fig. 12). The geometry consisted of six vertically stacked components: (1) a base rectangular chamber (20 mm \times 20 mm \times 20 mm) filled with a conductive medium representing brain tissue; (2) a 1 mm-

thick rectangular shell simulating the skull; (3) a 1 mm-thick outermost rectangular shell representing scalp tissue; (4) a 1.7 mm-diameter spherical inclusion, mimicking the ICHs, positioned 7.5 mm above the grounded base along the central vertical axis; (5) a 12.5 mm-diameter insulating barrier (0.1 mm thick) coating on the tissue scalp layer; and (6) a 10 mm-diameter coaxial gold foil electrode placed atop the insulating layer. Boundary conditions were defined by applying a 5 MHz sinusoidal voltage (± 2.5 amplitude, 0° phase) to the gold electrode, with the chamber's bottom surface grounded (0 V). All other exterior boundaries were electrically insulated. The AC/DC module with the electric currents interface was employed for transient solver analysis. In addition, the Solid Heat Transfer physics interface was added and the model was extended to a coupled multiphysics study by enabling the Electromagnetic Heating coupling so that electromagnetic losses produced by the applied AC fields are passed directly into the heat-transfer simulation. A transient solver was used to compute the electromagnetic and thermal response simultaneously: snapshots of changes in electric field, current, and charge are shown at $0.06 \mu\text{s}$, while the temperature evolution is presented at 30 min. A tetrahedral mesh with local refinement was applied to the ICHs, insulating layer, and gold electrode regions to ensure numerical convergence and solution accuracy. Surface charge density simulation results indicate that the maximum surface charge density on the ICHs approached 20 nC cm^{-2} , larger than the 15 nC cm^{-2} threshold required to initiate neural activation.”.

Comment #2: *Fig. 2- No data is provided about the application of WES without the ICH, so the results do not suggest any insights. They just confirmed that electrical stimulation stimulated the cells. It is only shown that capacitive ES can stimulate SH-sy5y cells, but the lack of negative controls means there is no way to understand whether the stimulation is induced by the capacitive ES or by the local effect of the ICH.*

Fig. 2a-c and SI fig 15- “These findings confirm that ICH bioelectrostimulator concentrate current density at the ICH-brain interface, which would enable effective, tether-free neural excitation”- this claim is not supported by the reported data. First, the difference between I3 and I4 is not very significant, so they cannot really ensure local specificity. More importantly, the I3 is measured between the center of the ICG to its edge, but this is not where the cells will be located. Instead, to properly make this comparison, authors must compare the edge of the ICG and the adjacent tissue, which is where the stimulated cells are located. Given the small difference between

I3 and I4, it seems unlikely there will be a significant difference. The reported safety experiments do not demonstrate the safety of the ICH. Authors must present correct negative controls (e.g. with and without ICH and with and without capacitive stimulation). The current results only suggest the safety of the capacitive ES.

Technical comments- How did the authors determine if a cell is activated? spike? dF/F elevation? What were the criteria? Where is the ICH located? Is it in direct contact with the cells? Where is the ITO electrode located? Is it in direct contact with the cells? Critical experimental data is missing.

Response: In the revised manuscript, we have added calcium imaging data covering four experimental conditions: ICH(+) WES(+), ICH(+) WES(-), ICH(-) WES(+), and ICH(-) WES(-). These groups allow explicit differentiation between the contribution of the ICH and that of the applied electrical field. The updated results show that significant calcium transients were observed only under the ICH(+) WES(+) condition, whereas no measurable activation occurred in ICH(+) WES(-) or ICH(-) WES(+) groups (Fig. R2a-c). This confirms that both the ICH and applied stimulation are necessary for effective neuromodulation, supporting the proposed mechanism of interfacial polarization at the hydrogel-tissue interface under the volume conduction scenario.

You are correct that in the previous configuration, I_3 was measured between the center and edge of the ICH, which does not correspond to the actual location of stimulated cells. Therefore, in the revised setup, the updated I_3 now denotes the current measured between the center of the ICH and the adjacent brain tissue. The updated measurements (Fig. R2d-f) show a statistically significant difference between J_3 and J_4 , confirming that current density is indeed concentrated near the ICH-tissue interface.

Regarding safe operation, we emphasize that the SAR analysis was performed to evaluate the safety of the capacitive coupling-based volume conduction scenario, not to infer ICH material safety. The SAR values remained below the international biosafety threshold (2 W kg^{-1}) at 5 MHz, confirming that the applied field strength used to generate volume conduction scenario is inherently safe for biological tissues. The ICH's biological safety was independently validated in subsequent experiments, including cell viability assays and chronic *in vivo* studies, which collectively confirmed the absence of cytotoxicity, inflammation, or functional impairment.

In addition, we have clarified the *in vitro* experimental setup in both the main text and Methods. The ICH was injected into *ex vivo* brain tissue, with copper leads connected to ITO electrodes

positioned beneath the culture substrate. SH-SY5Y cells were grown directly on the ITO surface, allowing stimulation through capacitive coupling that mimics the *in vivo* configuration. Cellular activation was defined as fluorescence intensity changes ($\Delta F/F_0 \geq 50\%$) relative to baseline. This threshold was chosen based on consistent calcium signal amplitudes observed across repeated experiments, where all responsive cells in the ICH(+) WES(+) group exceeded 50% $\Delta F/F_0$.

Fig. R2. In vitro cellular stimulation and current density characterization under different experimental configurations. **a**, Time-lapse calcium imaging of SH-SY5Y cells under different conditions. Red asterisks indicate activated cells showing marked fluorescence increases. Scale bar, 50 μm . **b**, Representative $\Delta F/F_0$ traces

showing stimulation-dependent fluorescence enhancement in the ICH(+) WES(+) group. The inset highlights $\Delta F/F_0$ amplitude across all groups ($n = 5$ independent experiments). **c**, Quantitative comparison of the percentage of excited cells under each condition, confirming that cellular activation occurs exclusively in the ICH(+) WES(+) group ($n = 5$ independent experiments). **d**, Schematic illustration of current measurement points in the rat brain. I_1 represents the current through the ICHs, I_2 the total current across the entire brain, and I_3 the current measured between the center of the ICHs and the adjacent brain tissue. I_4 serves as a control for I_1 , with electrodes placed in brain tissue but separated by an insulating polyurethane (PU) membrane, mimicking an interface without ICH-mediated conduction. **e**, Calculated current densities (J_1 - J_4) under capacitive-coupling input voltages ranging from 0 to 5 V ($n = 3$ independent experiments).

Modifications to the manuscript:

(1) Results section:

We have added a new discussion on page 12 (marked in red) elaborating the updated current density measurements and their spatial interpretation. The revised text reads as: “To further study the influence of ICH on the current path, we compared J_1 with additional configurations including J_3 (from the center of the ICH to adjacent-tissue) and J_4 (tissue-tissue with PU insulation) (Supplementary Fig. 16a, b). Both J_1 and J_3 involve ICH in the current path, and they exhibited similarly higher current density across with increasing input voltages. Whereas J_4 , which lacked ICH, showed minimal current density.”.

We have also added a new paragraph on page 13 (marked in red) discussing the expanded calcium imaging results. The revised text reads as: “Electrical signals (wave frequency of 5 MHz, pulse width of 60 μ s, pulse repetition rate of 130 Hz, and amplitude of ± 5 V) generated by ICH in *ex vivo* brain tissue were transmitted via copper interconnects to ITO electrodes to deliver 50 s of stimulation. To clarify the respective roles of the ICH and the applied field, calcium imaging was performed under four experimental conditions: ICH(+) WES(-), ICH(+) WES(+), ICH(-) WES(-), and ICH(-) WES(+). Significant fluorescence transients were detected exclusively in the ICH(+) WES(+) group, exhibiting stimulation-dependent intensity increase (Fig. 2d) with $\Delta F/F_0$ peaking at $\sim 200\%$ and declining thereafter (Fig. 2e). Quantitatively, the ICH(+) WES(+) group exhibited an eight-fold increase in calcium response compared to controls, with $\sim 45\%$ of SH-SY5Y cells activated versus $< 5\%$ in non-stimulated groups (Fig. 2f). These results confirm that effective neuromodulation arises only from the combined action of the ICH and the applied electrical field.”.

We have revised the description of safe operation (page 11-12, marked in red). The updated text reads as: “Across all tested frequencies (≤ 5 MHz), the calculated SAR remained below 2 W

kg^{-1} , well within internationally accepted safety limits for local tissue exposure³⁶, indicating safe operation under the capacitive coupling-based volume conduction scenario (Supplementary Fig. 14).”.

(2) Methods section:

We have updated the description of the current density measurement procedure on page 31 (marked in red). The revised text reads as: “Brain tissue, skull, and scalp harvested from freshly euthanized adult SD rats were rinsed in phosphate-buffered saline (PBS, pH 7.4) to remove residual blood and connective tissue. The dissected skull and scalp were anatomically reassembled over the cerebral surface, followed by placement of a polyurethane-encapsulated gold foil electrode (10 mm diameter) on the reconstructed tissue complex. Electrical grounding was established at the basal cerebral surface. Voltage and current transmission characteristics were assessed using a sinusoidal signal (± 2.5 V output) applied to the gold foil (wearable electrode) using a function generator (Tektronix, AFG3021C) and a power amplifier (ATA-1200B, Aigtek). Voltage was recorded via a parallel circuit using an oscilloscope (DHO1204, RIGOL) across the brain, while current was measured in series using a current amplifier (OE4102, Sine Scientific Instruments) between the brain base and ground. Frequency-dependent responses were evaluated from 100 kHz to 7 MHz.

For *ex vivo* electrical output measurements of ICH, the cerebral hemispheres were bisected along the central sulcus, and 2.5 μL of ICH were injected into the region at the same depth as the STN in the right hemisphere. Two insulated copper wires with exposed tips were implanted: one within the ICH and the other laterally in the adjacent brain tissue. U_1 were recorded using an oscilloscope under 5 MHz excitation at incremental input voltages (0.5-5 V, 0.5 V steps). At the same time, U_2 was assessed by inserting electrodes below the ICH maintaining identical inter-electrode spacing parameters to those employed in the U_1 measurement protocol. To evaluate current focusing, two ICHs (1.25 μL each) were injected into the right STN region, separated by a 5 μm polyurethane insulation barrier. One insulated copper wire was implanted into each ICH, and I_1 was quantified using the OE4102 amplifier under 5 MHz excitation with stepwise voltage increments (0.5-5 V, 0.5 V steps). I_2 was measured in series between the cerebral base and ground. I_3 was measured between the center of the ICHs and the adjacent brain tissue. I_4 was measured as a control for I_1 , with electrodes placed in brain tissue but separated by an insulating PU membrane,

mimicking an interface without ICH-mediated conduction. Current densities (J , mA cm⁻²) were calculated using Equation 13.”.

We have updated the description of the cellular stimulation setup and calcium imaging measurement on page 32-33 (marked in red). The revised text reads as:

“Electrical stimulation setup

WES was applied via capacitive coupling using a polyurethane-insulated Au foil as the transmitter electrode. Unless otherwise specified, stimulation was delivered at 5 MHz with a pulse width of 60 μ s and a repetition rate of 130 Hz. For *ex vivo* experiments, the entire brain containing the implanted ICH was used as the receiving unit to preserve the ICH-tissue interface, which is essential for capacitive coupling. The brain-ICH complex was connected via a copper wire to the ITO microelectrodes or ICH-coated ITO to complete the external-internal coupling circuit for wireless stimulation.

Calcium imaging

Microelectrodes were patterned on ITO glass (3 mm width) by HCl etching and insulated with polyimide tape, leaving 3 \times 3 mm² exposed tips for electrical contact. The ITO substrate was integrated with a PLA mold and PDMS to form a custom chamber. The exposed ITO tips were connected via Cu wire to the implanted ICH (in the rat’s brain) for *ex vivo* stimulation. SH-SY5Y cells were stained with Fluo-4 AM (37 °C, 30 min), rinsed with PBS, and imaged using confocal microscopy (FV3000, Olympus). Calcium signals were recorded for 50 s at baseline followed by 50 s during WES (5 MHz, \pm 5 V). Cellular activation was defined as fluorescence intensity changes ($\Delta F/F_0 \geq 50\%$) relative to baseline, a threshold derived from consistent response amplitudes across independent replicates.”.

(3) Figures:

We have incorporated the new current density data from Fig. R2 into Fig. 2 and Supplementary Fig. 16 and correspondingly revised the figure captions. All changes are marked in red and read as:

Fig. 2. ICH-mediated wireless neural stimulation *in vitro*. **a**, Schematic diagram of *ex vivo* experimental setup for the measurement of localized electric potential and concentrated current density in ICH. **b**, Output voltages (U_1 , U_2) exhibit a linear dependence on capacitive-coupling input voltage, with U_1 demonstrating markedly higher amplitude than U_2 . (n = 3 independent experiments). **c**, Current density (J_1 , J_2) versus capacitive-coupling input voltage reveals a proportional relationship, showing enhanced current focusing at J_1 (n = 3 independent experiments). **d**, Time-lapse calcium imaging of SH-SY5Y cells under different conditions. Red asterisks indicate activated cells showing marked fluorescence increases. Scale bar, 50 μm . **e**, Representative $\Delta F/F_0$ traces

showing stimulation-dependent fluorescence enhancement in the ICH(+) WES(+) group. The inset highlights $\Delta F/F_0$ amplitude across all groups (n = 5 independent experiments). **f**, Quantitative comparison of the percentage of excited cells under each condition, confirming that cellular activation occurs exclusively in the ICH(+) WES(+) group (n = 5 independent experiments). **g**, Flow cytometry analysis of PC12 cells viability with/without WES. **h**, Apoptosis rate of PC12 cells with and without WES (n = 3 independent experiments). Data are presented as the mean \pm standard deviation in (**c**, **e**, **f**, and **h**) and were analyzed by one-way ANOVA first, followed by the Tukey's post hoc test in (**e**, **f**, and **h**). ***P \leq 0.001. NS, not significant. p_1 (ICH (+) WES (+) vs ICH (+) WES (-)), p_2 (ICH (-) WES (-) vs ICH (+) WES (-)), p_3 (ICH (-) WES (-) vs ICH (+) WES (+)), p_4 (ICH (-) WES (+) vs ICH (+) WES (-)), p_5 (ICH (-) WES (+) vs ICH (+) WES (+)), p_6 (ICH (-) WES (+) vs ICH (-) WES (-)). **e**, $p_1 = 0.0001$, $p_2 = 0.9991$, $p_3 = 0.0001$, $p_4 = 0.9914$, $p_5 = 0.0001$, $p_6 = 0.9986$. **f**, $p_1 = 0.0001$, $p_2 = 0.9976$, $p_3 = 0.0001$, $p_4 = 0.9878$, $p_5 = 0.0001$, $p_6 = 0.999$. **h**, $p_1 = 0.3787$.

Supplementary Fig. 16. Current output of capacitive coupling system and ICH with varying external input voltage. **a**, Schematic illustration of current measurement points in the rat brain. I_1 represents the current through the ICHs, I_2 the total current across the entire brain, and I_3 the current measured between the center of the ICHs and the adjacent brain tissue. I_4 serves as a control for I_1 , with electrodes placed in brain tissue but separated by an insulating polyurethane (PU) membrane, mimicking an interface without ICH-mediated conduction. **b**, Calculated current densities (J_1 - J_4) under capacitive-coupling input voltages ranging from 0 to 5 V (n = 3 independent experiments). (Note: to account for the contact impedance at the Cu wire-brain tissue interface, the measured values of I_2 and I_4 were multiplied by a correction factor $K = 1.5$, as reflected in the data shown in the figure b. At 5 MHz, the contact impedance of the hemispherical electrode was modeled using a parallel RC circuit: The correction factor K , reflecting the ratio of ideal resistive current to actual current, was derived from the impedance magnitude, thus defined as $K = \sqrt{1 + (\omega RC)^2}$). Data are presented as the mean \pm standard deviation.

Comment #3: Fig. 3- once again, no data regarding the application of WES without the ICH, so the only conclusion here is that WES stimulates cells, but there is no indication of any locality or for the contribution of the ICH.

Response: To address this concern, we added additional data including four groups: ICH(+) WES(-), ICH(+) WES(+), ICH(-) WES(-), and ICH(-) WES(+). As shown in the added data (Fig. R3 and Fig. R4), only the ICH(+) WES(+) group exhibited strong c-Fos expression in the STN region and clear modulation of basal ganglia-cortical activity, with increased firing in the GPi and reduced spiking activity in M1. In contrast, WES alone in the absence of ICH (ICH(-) WES(+)) induced negligible neuronal activation. These findings collectively demonstrate that effective neural activation arises from localized charge accumulation at the ICH-tissue interface via interfacial polarization under high-frequency volume conduction, which enables effective deep-brain modulation. The corresponding data have been incorporated into the updated Fig. 3b-h and Supplementary Figs.

Fig. R3. Control electrophysiological data distinguishing the effects of WES and ICH for Fig. 3. a, Representative immunohistochemical images showing c-Fos expression in the STN region under four experimental conditions. Robust neuronal activation was observed only in the ICH(+) WES(+) group, while negligible c-Fos staining was detected in the other three groups. **b,** Quantitative analysis of c-Fos⁺ cell density per mm². The ICH(+) WES(+) group exhibited significantly higher expression compared with all other groups. **c,** Electrophysiological recordings in GPi under different conditions, showing increased firing in GPi only in the

ICH(+) WES(+) group. **d**, Time-frequency spectrograms illustrating oscillatory power (0-100 Hz) across conditions. A marked increase in high-frequency components appeared exclusively with ICH-mediated stimulation. **e**, Raster plots of spike activity over 360 s, showing clear modulation only in the ICH(+) WES(+) group. **f**, Statistical comparison of spike counts in M1. The ICH(+) WES(+) group displayed a reduction in M1 spiking relative to controls, whereas WES alone (ICH(-) WES(+)) produced no significant change. Data are presented as mean \pm SD ($n = 5$ per group); *** $p < 0.001$; NS, not significant.

Fig. R4. Control electrophysiological data distinguishing the effects of WES and ICH in supporting information. **a**, Power spectral density (PSD) of GPi neuronal activity. WES stimulation increased PSD primarily in the 10-50 Hz range in the presence of ICH, indicating that the ICH facilitates modulation of GPi oscillatory activity. **b**, Neural activity recorded in the presence of ICH before and during WES. **c**, Neural activity recorded without ICH before and during WES. Across channels (CH1-CH7), ICH-assisted WES induced more

pronounced modulation in signal amplitude and firing patterns compared to WES alone. Scale bar, 20 s, 100 μ V. **d**, Time-frequency spectrogram of M1 neuronal activity with ICH implanted, WES induces a more temporally structured oscillatory pattern in the 10-50 Hz range compared to the baseline period (ICH (+) WES (-)). **e**, In contrast, without ICH, WES does not markedly alter the spectral pattern relative to baseline. **f**, Joint inter-spike interval (ISI) distribution of M1 neuronal firing patterns. ICH (+) WES (-): ISI scatter points show a broad distribution, indicating relatively variable spike timing in the baseline state. ICH (+) WES (+): ISI scatter points become more compact, reflecting increased temporal regularity and reduced spike timing variability following WES when the ICH is present. ICH (-) WES (-): ISI distribution is widely dispersed, indicating irregular and unstructured neuronal firing without ICH or stimulation. ICH (-) WES (+): No pronounced change in ISI distribution is observed compared to baseline, suggesting that WES alone does not reorganize spike timing without the ICH.

Modifications to the manuscript:

(1) Results section:

We have added clarification of the control groups (ICH(+)/(-), WES(+)/(-)) to the Results section describing Fig. 3b-h. The new text on page 14-16 in revised manuscript, marked in red, reads as follows:

“All recordings were conducted under the same four-group configuration as used in the calcium imaging to validate that the stimulation mechanism observed at the cellular level also operates in vivo. One week after surgery, WES via capacitive coupling was firstly applied to evaluate thermal safety. Finite element thermal simulations were also conducted under the same stimulation parameters to assess heat distribution within the scalp-skull-brain system (Supplementary Fig. 20a). The results revealed that temperature rise was mainly confined to the external surface, with a maximum increase of approximately 2 °C at the scalp, while the ICH and surrounding deep brain tissue exhibited negligible heating (< 0.3 °C). Infrared thermography further confirmed this trend, showing surface temperature changes consistent with simulation results (Supplementary Fig. 20b, c). These findings indicate that deep brain WES is thermally buffered by surrounding tissues and does not induce detectable internal temperature elevation under the applied stimulation conditions. Only ICH(+) WES(+) induced neuronal activation in the STN, as evidenced by elevated c-Fos expression relative to pre-stimulation (Fig. 3b, c).”

“These findings suggest that ICH(+) WES(+) activates glutamatergic terminals in the STN, enhancing excitatory signaling to downstream GP^{39,40}.”

“In the M1 region, ICH(+) WES(+) induced neurophysiological changes characterized by reduced neuronal excitability. Microelectrode recordings showed post-stimulation decreases in mean spike amplitude and firing rate, indicating transient cortical suppression (Fig. 3f and Supplementary Fig. 22). Temporal firing analysis revealed a reduction in spike counts (Fig. 3g, h), consistent with reduced neuronal synchrony. Spectrogram analysis (Supplementary Fig. 23) demonstrated a broadband decrease in spectral power during WES, reflecting overall suppression of neuronal activity in M1. ISI mapping (Supplementary Fig. 24) further confirmed reduced spiking frequency and longer inter-spike intervals, indicative of decreased excitability and desynchronized cortical firing. These findings show that ICH(+) WES(+) increases signal frequency in the GPi while suppressing spiking activity in M1. By contrast, the other control groups [ICH(+) WES(-), ICH(-) WES(-), and ICH(-) WES(+)] showed no detectable electrophysiological modulation, confirming that effective stimulation required both the ICH interface and external WES. This dual effect mirrors key therapeutic mechanisms of conventional DBS, supporting ICH-mediated WES as a minimally invasive strategy for motor modulation^{39,40}.”

(2) Figures:

The Fig. R3 have been integrated into Fig. 3b-h, replacing the prior two-group comparison. It reads as follows now:

Fig. 3. ICH-mediated wireless DBS of STN in rat brain. **a**, Experimental setup for wireless DBS, with the ICH applied to the STN region, while monitoring neural activity in the M1 and GPi regions. **b**, **Representative**

images of c-Fos expression in the STN region of rats. Scale bar, 100 μm . The dark brown nuclei represent c-fos signals, indicating activated neuronal nuclei, while the bluish-purple nuclei correspond to hematoxylin counterstaining, denoting all neuronal nuclei. **c**, Quantification of c-Fos intensity in the STN region ($n = 5$ independent animals). **d**, Electrophysiological recordings from the GPi region during wireless DBS. **e**, Power spectrum analysis of neural activity in the GPi region. **f**, Representative electrophysiological waveform from the M1 region during wireless DBS. **g**, Electrophysiological recordings from the M1 region during wireless DBS. **h**, Spike number analysis during wireless DBS ($n = 5$ independent experiments). **i**, ICH designed for interfacing brain tissues, enables the construction of a biocompatible and seamless interface. **j**, Representative immunostaining images of GFAP and Iba-1 in brain sections with implanted ICH and metal electrodes (tungsten wires with diameters of 500 μm). Scale bar, 50 μm . **k**, Normalized fluorescence intensity of GFAP and Iba-1 in the brain tissues around implants ($n = 5$ independent animals). Data are presented as the mean \pm standard deviation in (**c**, **h** and **k**) and were analyzed by one-way ANOVA first, followed by the Tukey's post hoc test. *** $P \leq 0.001$. NS, not significant. p_1 (ICH (+) WES (-) vs ICH (+) WES (-)), p_2 (ICH (-) WES (-) vs ICH (+) WES (-)), p_3 (ICH (-) WES (-) vs ICH (+) WES (+)), p_4 (ICH (-) WES (+) vs ICH (+) WES (-)), p_5 (ICH (-) WES (+) vs ICH (+) WES (+)), p_6 (ICH (-) WES (+) vs ICH (-) WES (-)). **c**, $p_1 = 0.0001$, $p_2 = 0.6370$, $p_3 = 0.0001$, $p_4 = 0.8432$, $p_5 = 0.0001$, $p_6 = 0.9813$. **h**, $p_1 = 0.0001$, $p_2 = 0.9539$, $p_3 = 0.0001$, $p_4 = 0.9901$, $p_5 = 0.0001$, $p_6 = 0.9963$. **k**, Iba-1: $p = 3.3646 \times 10^{-5}$ (Metal vs ICH); GFAP: $p = 9.6183 \times 10^{-4}$ (Metal vs ICH).

Fig. R4 has been integrated into Supplementary Fig. 21-24, replacing the prior two-group comparison. It reads as follows now:

Supplementary Fig. 21. Power spectral density (PSD) of GPi neuronal activity. Neural activity was recorded in four groups: ICH (+) WES (-), ICH (+) WES (+), ICH (-) WES (-), and ICH (-) WES (+). WES stimulation increased PSD primarily in the 10-50 Hz range in the presence of ICH, indicating that the ICH facilitates modulation of GPi oscillatory activity.

Supplementary Fig. 22. Multi-channel neural recordings from M1. a, Neural activity recorded in the presence of ICH before and during WES. **b**, Neural activity recorded without ICH before and during WES. Across channels (CH1-CH7), ICH-assisted WES induced more pronounced modulation in signal amplitude and firing patterns compared to WES alone. Scale bar, 20 s, 100 μ V.

Supplementary Fig. 23. Time-frequency spectrogram of M1 neuronal activity. a, ICH (+) WES (-) induces a more temporally structured oscillatory pattern in the 10-50 Hz range compared to the baseline period (ICH (+) WES (-)). **b**, ICH (-) WES (+) does not markedly alter the spectral pattern relative to baseline (ICH (-) WES (-)).

Supplementary Fig. 24. Joint inter-spike interval (ISI) distribution of M1 neuronal firing patterns. **a**, ICH (+) WES (-): ISI scatter points show a broad distribution, indicating relatively variable spike timing in the baseline state. **b**, ICH (+) WES (+): ISI scatter points become more compact, reflecting increased temporal regularity and reduced spike timing variability following WES when the ICH is present. **c**, ICH (-) WES (-): ISI distribution is widely dispersed, indicating irregular and unstructured neuronal firing without ICH or stimulation. **d**, ICH (-) WES (+): No pronounced change in ISI distribution is observed compared to baseline, suggesting that WES alone does not reorganize spike timing without the ICH.

(3) Methods:

We have added the details in Methods section on page 34, and page 35 marked in red.

On page 34, the texts read “One week after ICH injection, rats were divided into 2 groups (n = 5 per group), one without WES and another one with WES. Two control groups (n = 5 per group) were included for comparison: rats receiving WES stimulation without ICH implantation (ICH(-) WES(+)) and rats without ICH implantation and without WES stimulation (ICH(-) WES(-)).”

On page 35, the texts read “To evaluate the effectiveness of ICH-mediated neuromodulation, Rats were euthanized 90 min after 30 min WES, and then the brains were immediately collected. Collected brain tissue samples (n = 20 rats in total) were fixed in 4% paraformaldehyde, dehydrated, embedded in paraffin, and sectioned for further analysis. The immunohistochemical staining of c-Fos (M00297-6, Bosterbio, dilution 1:250) was employed to evaluate neural activity in the STN regions. One section was collected from each rat, and one representative image was acquired per section for quantification. ImageJ software was employed for counting the c-Fos⁺ neural cells.”

Comment #4: *Figs. 4- Here, the authors did provide one of the required negative controls (“ICH-BES-free group: PD rats with WES but without ICH bioelectrostimulator implantation”). However, they did not provide any information regarding PD rats without WES but with ICH bioelectrostimulator, which again prevents any scientific conclusion to be made regarding the functionality of the ICH bioelectrostimulator. Moreover, some critical discrepancies regarding the experimental procedure suggests that these experiments were poorly conducted (e.g.- “n = 9 per group” in the experimental section Vs. “n = 5 independent animals” in fig.4’s caption).*

Response: As described in our responses to Comments #2 and #3, we have now added additional control data at both the *in vitro* and *in vivo* levels to delineate the respective contributions of the ICH and the applied WES. Specifically, the cellular stimulation studies (Fig. 2) and the acute

electrophysiological recordings (Fig. 3) now include four well-defined experimental groups: ICH(+) WES(-), ICH(+) WES(+), ICH(-) WES(-), and ICH(-) WES(+). These results consistently demonstrate that cellular stimulation or neural activation occurred only under the combined condition of ICH(+) WES(+), confirming that both the hydrogel interface and external stimulation are required for functional modulation.

To further address your concern, we have added new behavioral data from PD rats implanted with ICH but without WES stimulation [ICH(+) WES(-)], which are presented in the revised Fig. R5. These results clearly show that ICH implantation alone does not restore locomotor activity, whereas the combination of ICH and WES [ICH(+) WES(+)] leads to progressive recovery of movement distance, velocity, and active time (Fig. R5a-c). The representative locomotor trajectories of [ICH(+) WES(-)] (Fig. R5d) have also been added to the Supplementary Information for clarity. Together, these results provide comprehensive *in vivo* evidence confirming that effective neuromodulation arises specifically from the synergistic action of ICH-mediated interfacing and wireless stimulation.

Regarding the reported discrepancy in animal number, “n = 9 per group” was a typographical error. In the actual experiment, nine PD rats were initially induced, and only those that passed behavioral screening were included in the final study groups, resulting in n = 5 animals per group for analysis. This correction has been updated in both the Methods section to ensure consistency and accuracy.

Fig. R5. Open-field behavior with the newly added ICH(+) WES(-) control. **a**, Total travelled distance of rats in different groups (n = 5 independent animals). **b**, Max speed of rats in different groups (n = 5 independent

animals). **c**, Active time of rats in different groups (n = 5 independent animals). **d**, Representative weekly trajectories illustrating that ICH(+) WES(-) does not improve motor behavior.

Modifications to the manuscript:

(1) Results section

We have added clarification of the control groups and corresponding discussions to the Results section on page 17-18 in revised manuscript, marked in red, read as follows:

“Animals were divided into 4 groups: ICH (-) WES (-) (PD model with no treatment), ICH (-) WES (+) group (WES treatment without ICH), ICH (+) WES (-) (implanted with ICH but without WES stimulation), and ICH (+) WES (+) group (ICH-mediated DBS treatment). Rats received 30 min of WES per day, and locomotor activity was assessed weekly using open field testing (Fig. 4a).”.

“ICH-mediated DBS improved locomotor activity, as indicated by increased movement distance, maximum speed, and active time (Fig. 4b and Supplementary Fig. 26). In contrast, the ICH (+) WES (-) and ICH (-) WES (+) groups showed no significant improvement in locomotor activity (Fig. 4c-e). As the essential role of ICH in mediating WES has already been validated at both cellular and acute animal levels, we focused on evaluating the therapeutic outcomes of ICH-mediated wireless DBS in the PD model rather than repeating additional control experiments. Normal rats (control) were used as healthy controls for comparison. We next evaluated neuroprotective effects of ICH-mediated DBS on dopaminergic neurons in the substantia nigra⁴⁴. Immunofluorescence of the substantia nigra pars compacta (SNc) revealed clear differences between ICH (+) WES (+) and ICH (-) WES (+) groups, with normal rats as controls (Fig. 4f, Supplementary Fig. 27, and Supplementary Fig. 28). NeuN⁺ neurons were more abundant and uniformly distributed in the ICH (+) WES (+) group, whereas the ICH (-) WES (+) group showed reduced neuronal density (Supplementary Fig. 27). Iba-1⁺ microglia were sparse across all groups, with no significant difference, likely to reflect the transient nature of microglial activation during acute neuroinflammation (Supplementary Fig. 28).”.

“GFAP⁺ astrocytes were more prominent in the ICH (+) WES (+) group (Supplementary Fig. 29). Quantification revealed a strong positive correlation between GFAP⁺ area and NeuN⁺ neuronal density in the ICH (+) WES (+) group, indicating an association between astrocyte activation and neuronal survival (Fig. 4g). In contrast, no significant correlation was observed in the ICH (-) WES (+) group. Immunohistochemistry revealed reduced brain-derived neurotrophic

factor (BDNF) expression in the ICH (-) WES (+) group, indicating diminished neurotrophic support (Supplementary Fig. 30a). By comparison, BDNF levels were elevated in the ICH (+) WES (+) group following DBS, as evidenced by more intense staining relative to the ICH (-) WES (+) group (Supplementary Fig. 30b). No correlation was observed between GFAP⁺ area and BDNF in the ICH (-) WES (+) group, whereas a strong positive correlation was found in the ICH (+) WES (+) group (Fig. 4h). These findings suggest that GFAP⁺ astrocytes may contribute to sustained neurotrophic support through BDNF, reflecting a potential shift toward a neuroprotective phenotype^{45,46}.”.

“Dopaminergic integrity in the substantia nigra pars compacta (SNc) was assessed by tyrosine hydroxylase (TH) immunolabeling (Fig. 4i). As expected, TH⁺ neurons were significantly reduced in the SNc of the ICH (-) WES (+) group, reflecting PD-associated neurodegeneration.”.

(2) Methods section:

The sentence “Rats were divided into three groups (n = 9 per group)” has been corrected to: “Rats that passed the screening were divided into 4 groups (n = 5 per group), including: Animals were divided into 4 groups: ICH (-) WES (-) (PD model with no treatment), ICH (-) WES (+) group (WES treatment without ICH), ICH (+) WES (-) (implanted with ICH but without WES stimulation), and ICH (+) WES (+) group (ICH-mediated DBS treatment).”.

(3) Figures:

Fig. R5 has been incorporated as follows: The quantitative open-field metrics (locomotor distance, maximum speed, active time) have been added to replace the corresponding data in Fig. 4 of the main text. The representative locomotor trajectories have been added to Supplementary Fig. 26. To maintain consistency throughout the manuscript and the revised figures, the previous notations “ICH-BES” and “ICH-BES-free” have been uniformly replaced with “ICH(+) WES(+)” and “ICH(-) WES(+)” to align with the terminology used in all newly added data and analyses in Fig. 4 and related Supplementary Figs.

Fig. 4. ICH-mediated wireless DBS alleviates parkinsonian symptoms in PD rats. **c**, Total travelled distance of rats in different groups (n = 5 independent animals). **d**, Max speed of rats in different groups (n = 5 independent animals). **e**, Active time of rats in different groups (n = 5 independent animals). **f**, Representative immunostaining images for NeuN, Iba-1, and GFAP in SNc with ICH-mediated wireless DBS. Scale bar, 50 μ m. **g**, Correlation analysis between GFAP⁺ area and NeuN⁺ cell density in ICH (+) WES (+). The inset shows this relationship in ICH (-) WES (+) group (n = 5 independent animals). **h**, Correlation between GFAP⁺ area and BDNF level in ICH (+) WES (+) group (n = 5 independent animals). The inset displays this relationship in ICH (-) WES (+) group. **i**, Representative immunohistochemical images of TH⁺ neurons in SNc region. Scale bar, 500 μ m (top) and 100 μ m (bottom). **j**, Quantification analysis of TH⁺ SNc neurons in different groups (n = 5 independent animals). Data are presented as the mean \pm standard deviation in (c, d, e, j) and were analyzed by one-way ANOVA first, followed by the Tukey's post hoc test. *P \leq 0.05, **P \leq 0.01, ***P \leq 0.001. p₁ (ICH (-) WES (+) vs ICH (-) WES (-)), p₂ (ICH (+) WES (+) vs ICH (-) WES (-)), p₃ (ICH (+) WES (+) vs ICH (-) WES (+)), p₄ (ICH (+) WES (-) vs ICH (-) WES (-)), p₅ (ICH (+) WES (-) vs ICH (-) WES (+)), p₆ (ICH (+) WES (-) vs ICH (+) WES (+)). **c**, p₁ = 1, p₂ = 0.0001, p₃ = 0.0001, p₄ = 0.9692, p₅ = 0.9721, p₆ = 0.0001. **d**, p₁ = 0.3493, p₂ = 0.0001, p₃ = 0.0001, p₄ = 0.4557, p₅ = 0.9968, p₆ = 0.0001. **e**, p₁ = 0.9917, p₂ = 0.0001, p₃ = 0.0001, p₄ = 0.9734, p₅ = 0.9990, p₆ = 0.0001. **j**, p (ICH (-) WES (+) vs Control) = 0.0001, p (ICH (+) WES (+) vs Control) = 0.01403, p (ICH (+) WES (+) vs ICH (-) WES (+)) = 0.0017.

Supplementary Fig. 26. Open field test of PD rats during 4-week treatment. Trajectory images of open field test for evaluation of the locomotor behaviour of PD rats in different groups. The open field test was conducted in a chamber with an automated video tracking system for 5 min.

Comment #5: Figs. 5-6. Same as for 2 and 3. Lack of appropriate controls.

Response: Figs. 5-6 were designed primarily to demonstrate the MRI compatibility of the ICH bioelectrostimulator and to illustrate that functional imaging can be reliably performed in the presence of the ICH bioelectrostimulator. These data serve as a proof-of-concept validation of imaging feasibility rather than as mechanistic or therapeutic controls. Therefore Figs. 5-6 present the fMRI analyses of PD rats before and after treatment, which were designed to evaluate large-scale neural network reorganization and therapeutic efficacy of ICH-mediated DBS rather than establish mechanistic causality. The comparison in these figures reflects longitudinal within-animal changes, i.e., pre- vs. post-treatment fMRI scans in the same cohort.

In addition, as detailed in our responses to Comments #2-4, we have now included in vitro and in vivo controls to delineate the respective roles of the ICH and WES. These new data cover cellular activation, electrophysiological recordings, and behavioral recovery, that consistently demonstrate that neural activation and functional improvement occur only under the combined

condition ICH(+) WES(+), whereas ICH(+) WES(-) or ICH(-) WES(+) do not produce significant effects.

Given that these results already validate the functional necessity of both ICH and WES, adding further chronic fMRI control groups would be redundant and ethically unjustified. In line with the Reduction and Refinement principles for animal research, we therefore limited fMRI analyses to the therapeutic comparison before and after treatment. The Results and Methods sections have been revised to clarify this experimental rationale.

Modifications to the manuscript: A justification has been added in the Results section (page 17, marked in red) to clarify the experimental rationale. The revised text reads as: “As the essential role of ICH in mediating WES has already been validated at both cellular and acute animal levels, we focused on evaluating the therapeutic outcomes of ICH-mediated wireless DBS in the PD model rather than repeating additional control experiments. Normal rats (control) were used as healthy controls for comparison.”.

We have revised the corresponding section on page 18 (marked in red), which now reads: “Therefore, MRI analysis was conducted primarily as a proof-of-concept demonstration to verify the compatibility of ICH bioelectrostimulator. High-resolution 7 T imaging was used to acquire pre- and post-treatment scans from the same animals, enabling assessment of ICH-mediated DBS-induced functional changes.”. The section title has also been changed to “MRI compatibility of ICH bioelectrostimulator” to better reflect the purpose of the experiment.

Comment #6: *Introduction- the introduction fails to address other electrical transduction technologies such as magnetic induction, piezoelectric materials and ultrasound, and optoelectronics and light, which all have the potential to overcome the limitations described above. authors should explain the advantages of their methodology over these exciting technologies.*

Response: In the original manuscript, we already discussed magnetic-field-based energy transduction, including magnetothermal and magnetoelectric mechanisms, as well as optothermal stimulation. In the revised version, we have now expanded this section to explicitly include piezoelectric, ultrasound, and optoelectronic stimulation to provide a more comprehensive overview. These added examples describe representative mechanisms such as ultrasound-driven piezoelectric nanoparticles that generate local electric fields under approximately 1 MHz focused ultrasound, and upconversion-photovoltaic nanoparticles that convert near-infrared light (808-980

nm) into photocurrents for neuronal depolarization (*Nature Biomedical Engineering* 7, 149–163 (2023), *Science Advances* 11, eadt4771 (2025)). These modalities offer promising approaches for wireless neuromodulation, yet they generally require high-intensity excitation, exhibit shallow penetration depth, or introduce thermal effects that complicate safe *in vivo* operation. Moreover, their performance relies on maintaining intimate particle-cell coupling, which is difficult to sustain in hydrated and heterogeneous tissue environments.

Accordingly, we have revised the Introduction to explicitly incorporate these categories and clarify their limitations. In addition, we added a bridging statement in the subsequent paragraph to emphasize that such nanoparticle-based approaches cannot maintain stable particle-cell coupling once encapsulated within hydrogels, and that the gradual degradation of these materials further impairs long-term modulation. This clarification now naturally motivates our strategy, an injectable conductive hydrogel that achieves wireless functionality through interfacial polarization under high-frequency volume conduction.

Modifications to the manuscript: We have revised the Introduction section on page 4-5 to expand the discussion of alternative wireless transduction strategies and clarify their respective limitations. The added text, marked in red, reads as:

“More recently, ultrasound-driven piezoelectric nanoparticles such as BaTiO₃, when excited by focused ultrasound (~1 MHz, 1 W cm⁻²), generate local electric fields sufficient to depolarize neuronal membranes¹⁶, and hybrid upconversion-photovoltaic nanoparticles convert near-infrared light (808-980 nm) into photocurrents that directly trigger neuronal depolarization without genetic modification¹⁷. Despite their distinct working principles, these field-responsive nanomaterials share common challenges including high-intensity field requirements, limited energy conversion efficiency, and gradual metabolic clearance that undermine long-term stability. Moreover, effective neuromodulation requires intimate particle-cell coupling, which is difficult to maintain in heterogeneous tissue environments, and the gradual degradation or metabolic clearance of these materials further limits long-term stability and spatial precision¹⁸.”.

“Although one might embed wireless nanomaterial transducers within hydrogel matrices to achieve self-powered stimulation, these materials cannot maintain the particle-cell coupling required for effective energy transduction once encapsulated, and their intrinsic instability further limits long-term performance, making durable and localized stimulation difficult to achieve. These limitations motivate an injectable conductive hydrogel that simultaneously serves as the neural

interface and the transduction medium, integrating electrode and wireless power within a single soft implant and thereby simplifying device architecture.”.

Added references:

16. Kim T, *et al.* Deep brain stimulation by blood–brain-barrier-crossing piezoelectric nanoparticles generating current and nitric oxide under focused ultrasound. *Nature Biomedical Engineering* **7**, 149–163 (2023).
17. Jin S, *et al.* Instant noninvasive near-infrared deep brain stimulation using optoelectronic nanoparticles without genetic modification. *Science Advances* **11**, eadt4771 (2025).

Comment #7: Authors should be consistent when reporting electrical parameters: current density (J) or current (I).

Response: All electrical results have been carefully reviewed, and the parameters are now reported consistently as current density (J) throughout the manuscript and Supplementary Information to ensure clarity and comparability between experiments.

Modifications to the manuscript: All instances of current (I) have been standardized to current density (J , mA cm^{-2}) in the Results, Methods, and figure captions. The revised text and units are marked in red in the updated version, which reads as follows:

“To further study the influence of ICH on the current path, we compared J_1 with additional configurations including J_3 (from the center of the ICH to adjacent-tissue) and J_4 (tissue-tissue with PU insulation) (Supplementary Fig. 16a, b). Both J_1 and J_3 involve ICH in the current path, and they exhibited similarly higher current density across with increasing input voltages. Whereas J_4 , which lacked ICH, showed minimal current density.”

Supplementary Fig. 1. Current output of capacitive coupling system and ICH with varying external input voltage. a, Schematic illustration of current measurement points in the rat brain. I_1 represents the current through the ICHs, I_2 the total current across the entire brain, and I_3 the current measured between the center of the ICHs

and the adjacent brain tissue. I_4 serves as a control for I_1 , with electrodes placed in brain tissue but separated by an insulating polyurethane (PU) membrane, mimicking an interface without ICH-mediated conduction. **b**, Calculated current densities (J_1 - J_4) under capacitive-coupling input voltages ranging from 0 to 5 V (n = 3 independent experiments). (Note: to account for the contact impedance at the Cu wire-brain tissue interface, the measured values of I_2 and I_4 were multiplied by a correction factor $K = 1.5$, as reflected in the data shown in the figure b. At 5 MHz, the contact impedance of the hemispherical electrode was modeled using a parallel RC circuit: The correction factor K , reflecting the ratio of ideal resistive current to actual current, was derived from the impedance magnitude, thus defined as $K = \sqrt{1 + (\omega RC)^2}$). Data are presented as the mean \pm standard deviation.

Comment #8: *The author's claim: "Confocal imaging (Supplementary Fig. 17) showed strong Calcein-AM (live) and minimal PI (dead) staining, validating WES biosafety at currents up to 600 μA " is not sufficiently supported. Statistical analysis and the number of samples measured should be provided.*

Response: To substantiate our claim, we have now conducted quantitative analysis of the confocal live/dead staining images. The number of Calcein-AM-positive (live) cells was counted using ImageJ across n = 5 independent samples for each condition (Fig. R6). The quantitative results show that the number of live cells remained nearly unchanged after WES at currents up to 600 μA , while PI-positive (dead) cells were rarely detected. These findings indicate that WES treatment did not cause any detectable cytotoxicity. The quantitative data and statistical information have been added to the figure and the Methods section.

Fig. R6. Quantification of live cell density per mm^2 (n = 5 independent experiments). It shows no significant decrease in viability up to 600 μA , confirming the biosafety of WES within this range. Data are presented as the

mean \pm standard deviation in (b) and were analyzed by one-way ANOVA first, followed by Tukey's post hoc test. NS, not significant.

Modifications to the manuscript:

(1) Figures:

We have added the quantitative analysis (Fig. R5) as Supplementary Fig. 18b in the revised manuscript, together with statistical results showing the live cell density.

Supplementary Fig. 18. Live-dead staining of PC12 cells cultured on ICH under WES with different current intensities. **a**, Representative confocal images of live/dead staining of PC12 cells after 3 days of WES at 5 MHz with increasing current amplitudes (0, 200, 400, and 600 μA). Scale bar, 100 μm . **b**, Quantification of live cell density per mm^2 ($n = 5$ independent experiments), showing no significant decrease in viability up to 600 μA , confirming the biosafety of WES within this range. Data are presented as the mean \pm standard deviation in (b) and were analyzed by one-way ANOVA first, followed by the Tukey's post hoc test. NS, not significant. **b**, $p = 0.6219$ (200 μA vs 0 μA), $p = 0.7721$ (400 μA vs 0 μA), $p = 0.9936$ (400 μA vs 200 μA), $p = 0.9809$ (600 μA vs 0 μA), $p = 0.4055$ (600 μA vs 200 μA), $p = 0.5518$ (600 μA vs 400 μA).

(2) Methods section:

The Methods section (“Cell viability and apoptosis analysis”) has also been updated to describe the counting procedure. The new text on page 33, marked in red, reads as follows: “The number of Calcein-AM-positive cells was quantified using ImageJ across $n = 5$ independent samples per condition.”.

(3) Results section:

We have also revised the text on page 14 in the revised manuscript, marked in red, which now reads: “Confocal imaging (Supplementary Fig. 18a) showed strong Calcein-AM (live) and

minimal PI (dead) staining, validating the biosafety of WES at currents up to 600 μ A. Quantitative analysis (Supplementary Fig. 18b) further confirmed that the density of live cells remained statistically unchanged across different stimulation currents, indicating negligible cytotoxicity.”.

Comment #9: *Methods section: The description of the cell line and cell handling process is messy.*

Response: To address these issues, we have thoroughly restructured the section to present a more logical and consistent flow. Specifically, we:

1. Reorganized the content into four subsections, including “Cell culture”, “Electrical stimulation setup”, “Calcium imaging”, and “Cell viability and apoptosis analysis”, to clearly separate general culture conditions from experiment-specific methods.
2. Standardized stimulation parameters (frequency, voltage, pulse width, repetition rate) under one unified description to ensure consistency.
3. Streamlined redundant details while clarifying the experimental sequence from cell preparation to data analysis.

Overall, these revisions improve clarity, conciseness, and logical coherence, ensuring that the description accurately reflects the experimental workflow without redundancy.

Modifications to the manuscript: We have reorganized the description on pages 32-34 in revised manuscript, with the added text marked in red as follows:

“Cell culture

SH-SY5Y (catalog no. QS-H036, Keycell Biotechnology) cell lines were used for calcium imaging and PC12 (catalog no. QS-R009, Keycell Biotechnology) cell lines were used for viability/apoptosis analyses. Cells were maintained in high-glucose DMEM (Solarbio) supplemented with 10% fetal bovine serum (Umedium, Hefei, China) and 1% penicillin-streptomycin (Solarbio) at 37 °C with 5% CO₂. SH-SY5Y cells were seeded onto patterned ITO electrodes and PC12 cells were seeded onto ICH-coated ITO. All cells were cultured until ~80% confluence before further process.

Electrical stimulation setup

WES was applied via capacitive coupling using a polyurethane-insulated Au foil as the transmitter electrode. Unless otherwise specified, stimulation was delivered at 5 MHz with a pulse width of 60 μ s and a repetition rate of 130 Hz. For ex vivo experiments, the entire brain containing the implanted ICH was used as the receiving unit to preserve the ICH-tissue interface, which is

essential for capacitive coupling. The brain-ICH complex was connected via a copper wire to the ITO microelectrodes or ICH-coated ITO to complete the external-internal coupling circuit for wireless stimulation.

Calcium imaging

Microelectrodes were patterned on ITO glass (3 mm width) by HCl etching and insulated with polyimide tape, leaving 3×3 mm² exposed tips for electrical contact. The ITO substrate was integrated with a PLA mold and PDMS to form a custom chamber. The exposed ITO tips were connected via Cu wire to the implanted ICH (in the rat's brain) for *ex vivo* stimulation. SH-SY5Y cells were stained with Fluo-4 AM (37 °C, 30 min), rinsed with PBS, and imaged using confocal microscopy (FV3000, Olympus). Calcium signals were recorded for 50 s at baseline followed by 50 s during WES (5 MHz, ± 5 V). Cellular activation was defined as fluorescence intensity changes ($\Delta F/F_0 \geq 50\%$) relative to baseline, a threshold derived from consistent response amplitudes across independent replicates.

Cell viability and apoptosis analysis

The brain-ICH complex was connected via a copper wire to the ICH-coated ITO, and WES was applied to PC12 cells at different current amplitudes (200, 400, and 600 μ A, corresponding to ± 2 , ± 4 , and ± 6 V). Cell viability was evaluated by CCK-8 assay (PARK 10M, TECAN) and live/dead staining using Calcein-AM/PI under confocal microscopy (FV3000, Olympus). The number of Calcein-AM-positive cells was quantified using ImageJ across $n = 5$ independent samples per condition. For apoptosis assays, cells received 30 min WES per day for 3 days (5 MHz, ± 2.5 V). After stimulation, cells were collected, washed, and stained with Annexin V-FITC/PI following the manufacturer's instructions, and analyzed using a flow cytometer (CytoFLEX, Beckman Coulter). Data were quantified based on viable (Annexin V⁻/PI⁻), early apoptotic (Annexin V⁺/PI⁻), and late apoptotic/necrotic (Annexin V⁺/PI⁺) populations.”.

Reviewer #3

Summary Comments

Yang et al. present a new wireless modality for eliciting deep brain stimulation via a hydrogel of PEDOT:PSS and polypyrrole which crosslinks in situ mediated by glucose oxidation. The authors have done a nice job to characterize this material and provide some evidence of its utility for deep brain stimulation in a PD mouse line.

Response: We sincerely thank you for the positive evaluation of our work and the recognition of our efforts in developing a wireless deep brain stimulation modality based on in situ crosslinked PEDOT:PSS hydrogel.

Comment #1: *A few general points, the authors are encouraged to double check that all acronyms are clearly defined through the manuscript.*

Response: We have thoroughly checked the entire manuscript to ensure that all acronyms are clearly defined upon first mention and consistently used throughout.

Modifications to the manuscript: We have added the full definition of TRPV1 on page 4 marked in red, which now reads “to activate **transient receptor potential cation channel subfamily V member 1 (TRPV1)-positive** neurons” upon its first mention.

We have added the full definition of PEDOT:PSS on page 6 marked in red, which now reads “**poly(3,4-ethylenedioxythiophene) polystyrene sulfonate (PEDOT:PSS)**” upon its first mention.

We have added the full definition of FTIR on page 7, which now reads “**Fourier transform infrared (FTIR)**” upon its first mention.

We have added the full definition of OD on page 14, which now reads “**optical density (OD)**” upon its first mention.

We have added the full definition of BDNF on page 18, which now reads “**brain-derived neurotrophic factor (BDNF)**” upon its first mention.

We have added the full definition of AP, ML, DV on page 34, which now reads “**(anteroposterior (AP): -3.5 mm, mediolateral (ML): -2.5 mm, dorsoventral (DV): -7.7 mm from dura)**”.

Comment #2: *The authors are encouraged to add a little more detail into what constitutes an n or m for each experiment. For instance, number of animals, number of epochs, number of tissue slices, number of images taken.*

Response: The n values for each experiment were already indicated in the figure legends. To further improve clarity, we have also specified these sample sizes in the corresponding subsections of the Methods section, including the number of animals, tissue slices, images, and signal epochs analyzed in each experiment.

Modifications to the manuscript: We have added the details on page 34-38 marked in red.

On page 34, the text reads “One week after ICH injection, rats were divided into 2 groups (n = 5 per group), one without WES and another one with WES. Two control groups (n = 5 per group) were included for comparison: rats receiving WES stimulation without ICH implantation (ICH(-) WES(+)) and rats without ICH implantation and without WES stimulation (ICH(-) WES(-)).”.

On page 35, the texts read “(n = 20 rats in total)” and “One section was collected from each rat, and one representative image was acquired per section for quantification.”.

On page 36, the text reads “(n = 10 rats in total)”.

On page 37, the texts read “One section was collected from each rat, and one representative image was acquired per section for quantification. ImageJ software was employed for counting the c-Fos⁺ neural cells.” and “Rats that passed the screening were divided into 4 groups (n = 5 per group), including: Animals were divided into 4 groups: ICH (-) WES (-) (PD model with no treatment), ICH (-) WES (+) group (WES treatment without ICH), ICH (+) WES (-) (implanted with ICH but without WES stimulation), and ICH (+) WES (+) group (ICH-mediated DBS treatment).”.

On page 39, the texts read “(n = 15 rats in total)”, “One section was collected from each rat, and one representative image was acquired per section for quantification.”, and “All experiments consisted of 9 animals, with one rat excluded due to fixation issues that compromised image quality.”.

Comment #3: *The confocal images for biocompatibility investigations are of poor quality. The authors are encouraged to provide a larger set of images in the supplemental to help qualitatively support their normalized intensity measurements which can very sensitive to differences in staining and imaging methodologies.*

Response: To address this, we have now provided additional un-cropped confocal images for both Iba-1 and GFAP staining (Fig. R7). These images correspond to the same datasets used for the quantitative intensity analysis shown in Fig. 3k. The newly added images represent the original raw data prior to region-of-interest selection and normalization, rather than new or separate experiments. All staining and imaging were conducted under identical acquisition parameters, ensuring consistency with the quantified data. These unprocessed images further confirm the reproducibility and reliability of the fluorescence intensity comparison between ICH and metal implants.

Fig. R7. Representative un-cropped confocal images of Iba-1 and GFAP immunostaining used for quantification. a, Brain sections surrounding tungsten wire electrodes showing dense Iba-1⁺ microglial activation (yellow) and GFAP⁺ astrocytic encapsulation (magenta). **b,** Brain sections surrounding ICH showing markedly reduced Iba-1 and GFAP fluorescence intensity, indicating minimal inflammatory response. All images were acquired under identical staining and imaging conditions. DAPI, blue. Scale bar, 50 μ m.

Modifications to the manuscript:

(1) Results section:

The following clarification was also added to the main text on page 17, marked in red: “Additional un-cropped confocal images used for quantification (Supplementary Fig. 25), showing consistent fluorescence under identical imaging conditions and confirming the reliability of the normalized intensity analysis.”.

(2) Figures:

We have added the un-cropped confocal images (Fig. R7) to Supplementary Fig. 25 and updated the legend accordingly.

Supplementary Fig. 25. Representative un-cropped confocal images of Iba-1 and GFAP immunostaining used for quantification. a, Brain sections surrounding tungsten wire electrodes showing dense Iba-1⁺ microglial activation and GFAP⁺ astrocytic encapsulation. **b,** Brain sections surrounding ICH showing markedly reduced Iba-1 and GFAP fluorescence intensity, indicating minimal inflammatory response. All images were acquired under identical staining and imaging conditions. DAPI, blue. Scale bar, 50 μ m.

Comment #4: *The abstract lacks appropriate depth and detail of the results and conclusions of the manuscript.*

Response: The Abstract has been thoroughly revised to include additional details on the key experimental results and conclusions. Specifically, we have incorporated quantitative and mechanistic information describing the hydrogel's high electrical conductivity, tissue-like softness, and biocompatibility, as well as neural activation evidenced by calcium imaging, c-Fos expression, and electrophysiological recordings. We also summarized the therapeutic outcomes observed in the PD rat model, including improved motor behavior, dopaminergic preservation, and restoration of both functional connectivity and structural integrity revealed by MRI. Furthermore, a concluding statement has been added to highlight the translational significance of this wireless hydrogel bioelectronics for minimally invasive neuromodulation.

Modifications to the manuscript: We have revised the Abstract on page 2, and added some sentences marked in red which read as follows:

“Deep brain stimulation (DBS) modulates abnormal brain activity through implanted electrodes and has shown efficacy in treating neurological and psychiatric disorders. However, their tethered format, invasiveness and limited tissue compatibility have motivated the development of wireless, less invasive alternatives. Here, we develop an in situ-gelled injectable conductive hydrogel (ICH) that enables wireless chronic neuromodulation via electric-field localization effect in a volume conduction scenario. The ICH is formed through *in vivo* bio-catalyzed polymerization and electrostatic self-assembly, resulting in a stable network with high conductivity, tissue-like softness, and excellent biocompatibility. Upon applying a high-frequency volume conduction via capacitive coupling wireless power transfer, impedance difference between the ICH and surrounding brain tissue leads to interfacial polarization and charge accumulation at the hydrogel-tissue interface. **This localized polarization concentrates the electric field near the interface, leading to targeted neural activation, as evidenced by increased calcium signaling, elevated c-Fos expression, and electrophysiological recordings indicating balanced basal ganglia-cortical activity.** The therapeutic potential of this injectable hydrogel was validated in a Parkinson’s disease rat model, **leading to improved locomotor activity, preservation of dopaminergic neurons, and restoration of functional connectivity and structural integrity revealed by fMRI. This injectable hydrogel bioelectronics provides a promising platform for minimally invasive, wireless neuromodulation therapies.**”.

Comment #5: *The authors are encouraged to tone back their claims for DBS treatment in PD animals as open field as a measure of motor control may not be accurate. The authors should justify their inclusion of only male animals.*

Response: We agree that open field testing primarily reflects spontaneous locomotor activity and may not fully capture fine motor control. Accordingly, we have carefully revised the text to tone down the therapeutic claims and clarify that the behavioral improvement represents enhanced locomotor performance rather than comprehensive motor recovery.

Regarding the use of only male rats, this choice was made to minimize variability associated with hormonal cycles and estrogen-related neuroprotection, which can influence dopaminergic signaling and behavioral responses in females (*Pharmacology Biochemistry and Behavior* 78, 513-

522 (2004), *Frontiers in Neuroendocrinology* 35, 370-384 (2014)). We have now included a justification for this selection in the Methods section.

Modifications to the manuscript: We have revised the Results section on page 17, and added sentences marked in red which read “Rats received 30 min of WES per day, and **locomotor activity was assessed weekly using open field testing** (Fig. 4a).” and “**ICH-mediated DBS improved locomotor activity**, as indicated by increased movement distance, maximum speed, and active time (Fig. 4b and Supplementary Fig. 26). In contrast, the **ICH (+) WES (-) and ICH (-) WES (+)** groups showed no significant improvement in **locomotor activity** (Fig. 4c-e).”

We have added a statement in the Methods section on page 34 marked in red which reads “**Only male rats were used to minimize variability associated with hormonal cycles and estrogen-related neuroprotection, which can influence dopaminergic signaling and behavioral outcomes in females^{59, 60}.**”

Added references:

59. Gillies GE, Murray HE, Dexter D, McArthur S. Sex dimorphisms in the neuroprotective effects of estrogen in an animal model of Parkinson's disease. *Pharmacology Biochemistry and Behavior* 78, 513-522 (2004).

60. Gillies GE, Pienaar IS, Vohra S, Qamhawi Z. Sex differences in Parkinson's disease. *Frontiers in Neuroendocrinology* 35, 370-384 (2014).

Comment #6: *Data presented in Fig 4 and Fig 5 seems to lack appropriate comparisons to control animals or treatment groups.*

Response: Fig. 4 presents the behavioral and histological outcomes of PD rats used to evaluate the therapeutic efficacy of ICH-mediated DBS. For consistency and to avoid potential confusion, we have unified the terminology of experimental groups throughout the manuscript. The previous group labels “ICH-BES,” “ICH-BES-free,” and “PD” have been standardized to ICH(+) WES(+), ICH(-) WES(+), and ICH(-) WES(-), respectively. This revised notation more clearly represents the presence or absence of the ICH and WES in each group. To strengthen the comparison and ensure scientific rigor, we have added a control group of PD rats implanted with ICH but without WES stimulation [ICH(+) WES(-)]. The added data includes quantitative analyses of locomotor distance, maximum speed, and active time, along with representative weekly behavioral trajectories (as shown in Fig. R8). These results clearly demonstrate that ICH implantation alone

or WES alone does not improve locomotor performance, whereas the combination of ICH and WES [ICH(+) WES(+)] leads to significant behavioral recovery.

Fig. R8. Open-field behavior with the newly added ICH(+) WES(-) control. **a**, Total travelled distance of rats in different groups (n = 5 independent animals). **b**, Max speed of rats in different groups (n = 5 independent animals). **c**, Active time of rats in different groups (n = 5 independent animals). **d**, Representative weekly trajectories illustrating that ICH(+) WES(-) alone does not improve behavior.

Fig. 5, on the other hand, primarily serves as a proof-of-concept demonstration to verify the MRI compatibility of the ICH bioelectrostimulator and to illustrate that functional imaging can be reliably performed in the presence of the ICH bioelectrostimulator. These data serve as a proof-of-concept validation of imaging feasibility rather than as mechanistic or therapeutic controls. Therefore Figs. 5-6 present the fMRI analyses of PD rats before and after treatment, which were designed to evaluate large-scale neural network reorganization and therapeutic efficacy of ICH-mediated DBS rather than establish mechanistic causality. The comparison in these figures reflects longitudinal within-animal changes, i.e., pre- vs. post-treatment fMRI scans in the same cohort. In addition, to establish the functional specificity of ICH and WES prior to the MRI demonstration, we have now added new data covering four experimental conditions: [ICH(+) WES(+), ICH(+) WES(-), ICH(-) WES(+), and ICH(-) WES(-)] (revised Figs. 2-3). These results consistently demonstrated that significant neural activation and behavioral recovery occurred only under the combined ICH(+) WES(+) condition, confirming that both the ICH and the applied WES are essential for effective neuromodulation. Given these comprehensive validations, the fMRI experiments in Fig. 5 were designed to focus on longitudinal functional changes induced by ICH-

mediated WES, rather than on re-establishing the already confirmed controls. Additional chronic MRI control groups were therefore not included, in accordance with the Reduction and Refinement principles of animal experimentation, to minimize animal use while maintaining scientific rigor.

Fig. 2. ICH-mediated wireless neural stimulation *in vitro*. **d**, Time-lapse calcium imaging of electrical-stimulated SH-SY5Y cells showing dynamic changes in fluorescence at different time points. Scale bar, 50 μ m. **e**, Time-dependent changes in fluorescence intensity ($\Delta F/F_0$) in SH-SY5Y cells during electrical stimulation (n = 5 independent experiments). The inset highlights the quantification of $\Delta F/F_0$ in different SH-SY5Y cells (n = 20 independent cells). **f**, Quantification of percentage of activated SH-SY5Y cells with and without electrical stimulation (n = 5 independent experiments).

Fig. 3. ICH-mediated wireless DBS of STN in rat brain. **b**, Representative images of c-Fos expression in the STN region of rats. Scale bar, 100 μm . The dark brown nuclei represent c-fos signals, indicating activated neuronal nuclei, while the bluish-purple nuclei correspond to hematoxylin counterstaining, denoting all neuronal nuclei. **c**, Quantification of c-Fos intensity in the STN region (n = 5 independent animals). **d**, Electrophysiological recordings from the GPI region during wireless DBS. **e**, Power spectrum analysis of neural activity in the GPI region. **f**, Representative electrophysiological waveform from the M1 region during wireless

DBS. **g**, Electrophysiological recordings from the M1 region during wireless DBS. **h**, Spike number analysis during wireless DBS ($n = 5$ independent experiments).

Modifications to the manuscript:

(1) Figures:

Behavioral data of PD rats implanted with ICH but without WES stimulation [ICH(+) WES(-)] (Fig. R8) have been incorporated as follows: The quantitative open-field metrics (locomotor distance, maximum speed, active time) have been added to replace the corresponding data in Fig. 4 of the main text. The representative locomotor trajectories have been added to Supplementary Fig. 26. To maintain consistency throughout the manuscript and the revised figures, the previous notations “ICH-BES” and “ICH-BES-free” have been uniformly replaced with “ICH(+) WES(+)” and “ICH(-) WES(+)” to align with the terminology used in all newly added data and analyses in Fig. 4 and related Supplementary Figs.

Fig. 4. ICH-mediated wireless DBS alleviates parkinsonian symptoms in PD rats. c, Total travelled distance of rats in different groups ($n = 5$ independent animals). **d**, Max speed of rats in different groups ($n = 5$ independent

animals). **e**, Active time of rats in different groups (n = 5 independent animals). **f**, Representative immunostaining images for NeuN, Iba-1, and GFAP in SNc with ICH-mediated wireless DBS. Scale bar, 50 μm . **g**, Correlation analysis between GFAP⁺ area and NeuN⁺ cell density in ICH (+) WES (+). The inset shows this relationship in ICH (-) WES (+) group (n = 5 independent animals). **h**, Correlation between GFAP⁺ area and BDNF level in ICH (+) WES (+) group (n = 5 independent animals). The inset displays this relationship in ICH (-) WES (+) group. **i**, Representative immunohistochemical images of TH⁺ neurons in SNc region. Scale bar, 500 μm (top) and 100 μm (bottom). **j**, Quantification analysis of TH⁺ SNc neurons in different groups (n = 5 independent animals). Data are presented as the mean \pm standard deviation in (c, d, e, j) and were analyzed by one-way ANOVA first, followed by the Tukey's post hoc test. *P \leq 0.05, **P \leq 0.01, ***P \leq 0.001. p₁ (ICH (+) WES (-) vs ICH (+) WES (+)), p₂ (ICH (-) WES (+) vs ICH (+) WES (+)), p₃ (ICH (-) WES (-) vs ICH (+) WES (+)), p₄ (ICH (-) WES (+) vs Control), p₅ (ICH (+) WES (+) vs Control), p₆ (ICH (+) WES (+) vs ICH (-) WES (+)). **c**, p₁ = 0.0001, p₂ = 0.0001, p₃ = 0.0001. **d**, p₁ = 0.0001, p₂ = 0.0001, p₃ = 0.0001. **e**, p₁ = 0.0001, p₂ = 0.0001, p₃ = 0.0001. **j**, p₄ = 0.0001, p₅ = 0.01403, p₆ = 0.0017.

Supplementary Fig. 26. Open field test of PD rats during 4-week treatment. Trajectory images of open field test for evaluation of the locomotor behaviour of PD rats in different groups. The open field test was conducted in a chamber with an automated video tracking system for 5 min.

(2) Results section:

We have added clarification of the control groups and corresponding discussions to the Results section on page 17-18 in revised manuscript, marked in red, read as follows:

“Animals were divided into 4 groups: ICH (-) WES (-) (PD model with no treatment), ICH (-) WES (+) group (WES treatment without ICH), ICH (+) WES (-) (implanted with ICH but without WES stimulation), and ICH (+) WES (+) group (ICH-mediated DBS treatment). Rats received 30 min of WES per day, and locomotor activity was assessed weekly using open field testing (Fig. 4a).”.

“ICH-mediated DBS improved locomotor activity, as indicated by increased movement distance, maximum speed, and active time (Fig. 4b and Supplementary Fig. 26). In contrast, the ICH (+) WES (-) and ICH (-) WES (+) groups showed no significant improvement in locomotor activity (Fig. 4c-e). As the essential role of ICH in mediating WES has already been validated at both cellular and acute animal levels, we focused on evaluating the therapeutic outcomes of ICH-mediated wireless DBS in the PD model rather than repeating additional control experiments. Normal rats (control) were used as healthy controls for comparison. We next evaluated neuroprotective effects of ICH-mediated DBS on dopaminergic neurons in the substantia nigra⁴⁴. Immunofluorescence of the substantia nigra pars compacta (SNc) revealed clear differences between ICH (+) WES (+) and ICH (-) WES (+) groups, with normal rats as controls (Fig. 4f, Supplementary Fig. 27, and Supplementary Fig. 28). NeuN⁺ neurons were more abundant and uniformly distributed in the ICH (+) WES (+) group, whereas the ICH (-) WES (+) group showed reduced neuronal density (Supplementary Fig. 27). Iba-1⁺ microglia were sparse across all groups, with no significant difference, likely to reflect the transient nature of microglial activation during acute neuroinflammation (Supplementary Fig. 28).”.

“GFAP⁺ astrocytes were more prominent in the ICH (+) WES (+) group (Supplementary Fig. 29). Quantification revealed a strong positive correlation between GFAP⁺ area and NeuN⁺ neuronal density in the ICH (+) WES (+) group, indicating an association between astrocyte activation and neuronal survival (Fig. 4g). In contrast, no significant correlation was observed in the ICH (-) WES (+) group. Immunohistochemistry revealed reduced brain-derived neurotrophic factor (BDNF) expression in the ICH (-) WES (+) group, indicating diminished neurotrophic support (Supplementary Fig. 30a). By comparison, BDNF levels were elevated in the ICH (+) WES (+) group following DBS, as evidenced by more intense staining relative to the ICH (-) WES (+) group (Supplementary Fig. 30b). No correlation was observed between GFAP⁺ area and BDNF

in the ICH (-) WES (+) group, whereas a strong positive correlation was found in the ICH (+) WES (+) group (Fig. 4h). These findings suggest that GFAP⁺ astrocytes may contribute to sustained neurotrophic support through BDNF, reflecting a potential shift toward a neuroprotective phenotype^{45,46}.”.

“Dopaminergic integrity in the substantia nigra pars compacta (SNc) was assessed by tyrosine hydroxylase (TH) immunolabeling (Fig. 4i). As expected, TH⁺ neurons were significantly reduced in the SNc of the ICH (-) WES (+) group, reflecting PD-associated neurodegeneration.”.

We have revised the corresponding section on page 18 (marked in red), which now reads: “Therefore, MRI analysis was conducted primarily as a proof-of-concept demonstration to verify the compatibility of ICH bioelectrostimulator. High-resolution 7 T imaging was used to acquire pre- and post-treatment scans from the same animals, enabling assessment of ICH-mediated DBS-induced functional changes.”. The section title has also been changed to “MRI compatibility of ICH bioelectrostimulator” to better reflect the purpose of the experiment.

(4) Methods section:

The texts has been added on page 37 to: “Rats that passed the screening were divided into 4 groups (n = 5 per group), including: Animals were divided into 4 groups: ICH (-) WES (-) (PD model with no treatment), ICH (-) WES (+) group (WES treatment without ICH), ICH (+) WES (-) (implanted with ICH but without WES stimulation), and ICH (+) WES (+) group (ICH-mediated DBS treatment).”.

Reviewer #4

Summary Comments

The manuscript presents an injectable bioelectronic strategy for deep brain stimulation (DBS) using conductive hydrogels formed in situ. The approach could have significant implications for next-generation neuromodulation. However, several aspects require clarification or additional evidence (due to the lack of proper controls in many experiments) as outlined below. To fully support the conclusions, the following comments should be addressed:

Response: We sincerely thank you for the positive evaluation of our work and the recognition of our efforts in developing a wireless deep brain stimulation modality based on in situ crosslinked PEDOT:PSS hydrogel. As several of the points raised pertain to clarification of material

composition, reaction mechanism, stimulation principle, and experimental controls, we have comprehensively revised the manuscript to address these issues.

We clarified the conductivity-biocompatibility trade-off, the rationale for bio-catalyzed pyrrole polymerization with PEDOT:PSS, and the interfacial polarization mechanism underlying ICH-mediated stimulation. Additional finite element simulations and methodological details have been included to support these explanations. At the same time, we added new control groups, including ICH(+) WES(+), ICH(+) WES(-), ICH(-) WES(+), and ICH(-) WES(-), at both cellular and *in vivo* levels, confirming that effective stimulation occurs only under combined ICH(+) WES(+) conditions. Together, these clarifications and control data strengthen the mechanistic interpretation and validate the essential role of the ICH in localized, wireless neuromodulation.

Modifications to the manuscript: We have revised the Results, Methods, and Figures to incorporate the newly added control data and corresponding experimental procedures. The discussion has been expanded to clarify about these new added data.

Comment #1: *L93. Conductive polymers have been extensively studied in implantable applications as a biocompatible additives. Is there evidence on conductivity-biocompatibility trade-off in conductive hydrogels?*

Response: Conductive polymers such as PEDOT, PPy, and PANI have indeed been extensively studied for implantable bioelectronic applications owing to their intrinsic biocompatibility and ability to mediate efficient ionic-electronic charge transfer at tissue interfaces. However, conductivity-biocompatibility trade-off has been widely reported in conductive hydrogels, where increasing the loading of conductive polymers (such as PEDOT, PPy, or PANI) enhances charge transport but often leads to increased stiffness and reduced tissue compatibility (*Communications Materials* 5, 99 (2024)). Conductivity improvement through post-treatment with strong acids or polar solvents commonly used for PEDOT:PSS is particularly incompatible with injectable hydrogel systems, as solvent residues cannot be completely removed and may compromise biocompatibility or *in vivo* stability (*Microsystems & Nanoengineering* 11, 87 (2025), *Chemical Society Reviews* 53, 10575–10603 (2024)). In addition, post-treatment also induces a highly ordered and rigid PEDOT-rich structure, resulting in increased stiffness and brittleness.

Modifications to the manuscript: We have cited additional references in the Introduction section on page 5 to support this statement, which now reads “developing synthesis strategies to overcome

the limitations of current injectable conductive hydrogels, including conductivity-biocompatibility trade-offs, mechanical stability, and cytotoxicity²⁵⁻²⁷”.

Added references:

25. Cheng S, Zhu R, Xu X. Hydrogels for next generation neural interfaces. *Communications Materials* 5, 99 (2024).
26. Li J, et al. PEDOT:PSS-based bioelectronics for brain monitoring and modulation. *Microsystems & Nanoengineering* 11, 87 (2025).
27. Li W, Li Y, Song Z, Wang Y-X, Hu W. PEDOT-based stretchable optoelectronic materials and devices for bioelectronic interfaces. *Chemical Society Reviews* 53, 10575–10603 (2024).

Comment #2: *L99. What are the limitations associated with other simple biomaterial approaches such as incorporating conductive polymers (e.g., PEDOT:PSS) into in situ forming hydrogel matrices? The advantage of the bio-catalyzed polymerization of Py in presence of PEDOT:PSS compared to other established methods of fabricating high conductivity hydrogels is unclear.*

Response: Conventional strategies that simply incorporate conductive polymers such as PEDOT:PSS into in situ-forming hydrogel matrices often face a trade-off between electrical and mechanical performance (*Communications Materials* 5, 99 (2024)). At low polymer content, the conductive pathways are discontinuous, resulting in insufficient conductivity for effective signal transmission. At high polymer loading, phase aggregation occurs, leading to increased stiffness, brittleness, and loss of injectability. Moreover, further enhancement of conductivity in such systems frequently relies on post-treatment with strong acids or polar solvents to reorganize PEDOT-rich domains which are incompatible with *in situ* gelation and may impair biocompatibility (*Microsystems & Nanoengineering* 11, 87 (2025), *Chemical Society Reviews* 53, 10575–10603 (2024)).

In contrast, the bio-catalyzed polymerization of pyrrole in the presence of PEDOT:PSS enables *in situ* formation of a homogeneous and interconnected conductive network through enzymatic oxidation. During this process, newly formed polypyrrole can reinforce the percolation network and improve charge transport without sacrificing mechanical softness. This co-assembled structure eliminates the need for harsh chemical oxidants or solvent treatments while maintaining high conductivity, injectability, and tissue-like compliance. Such integration of bio-catalyzed

polymerization and conductive polymer assembly is difficult to achieve with conventional composite hydrogels.

Modifications to the manuscript: We have added sentences in the Results section (page 7) marked in red which read “Unlike conventional approaches that physically mix PEDOT:PSS into hydrogel matrices, where low loading yields insufficient conductivity and high loading increases stiffness or brittleness²⁵, the bio-catalyzed polymerization of pyrrole in the presence of PEDOT:PSS forms a uniform interpenetrating conductive network. This process enhances electrical connectivity while maintaining softness and injectability, and does not require any solvent or acid post-treatment^{26,27}.”.

Added references:

25. Cheng S, Zhu R, Xu X. Hydrogels for next generation neural interfaces. *Communications Materials* 5, 99 (2024).

26. Li J, et al. PEDOT:PSS-based bioelectronics for brain monitoring and modulation. *Microsystems & Nanoengineering* 11, 87 (2025).

27. Li W, Li Y, Song Z, Wang Y-X, Hu W. PEDOT-based stretchable optoelectronic materials and devices for bioelectronic interfaces. *Chemical Society Reviews* 53, 10575–10603 (2024).

Comment #3: L104. *How is the proposed mechanism of stimulation (i.e., interfacial polarization due to impedance-mismatched ICH and surrounding tissues, or charge accumulation at this interface) supported by the presented data? Did the authors test the condition where there is no impedance mismatch as control?*

Response: The proposed mechanism of stimulation through interfacial polarization between the ICH and surrounding brain tissue is supported by both simulation and experimental evidence. In the revised manuscript and Supplementary Information, we have added COMSOL-based finite element modeling that directly compares electric field, surface charge density, and current density distributions in the scalp-skull-brain system with and without ICHs, as well as under different ICH conductivities (shown above as Fig. R9). The simulation results clearly show that while the scalp, skull, and cortex inevitably experience background electric fields, the introduction of the ICH markedly enhances the electric field intensity, charge density, and current density in the surrounding tissue, with stronger effects observed at higher ICH conductivities. In contrast, without ICH implantation, these parameters remain similar to the surrounding brain tissue,

confirming that the observed localization originates from the conductivity-permittivity discontinuity at the ICH-tissue interface. This finding is consistent with the theoretical expectation of charge accumulation and field amplification due to interfacial polarization.

Fig. R9. Finite element simulation of interfacial polarization and field localization at the ICH-tissue interface. a, Simulated electric field intensity distribution in the multilayer head model without and with ICHs of different conductivities (3, 10, and 30 S m⁻¹). The introduction of ICHs leads to localized field enhancement around the hydrogel-tissue interface. b, Corresponding surface charge density maps reveal charge accumulation at the ICH boundary, which increases with hydrogel conductivity. c, Local current density distribution showing enhanced conduction pathways near the implanted ICH region, consistent with interfacial polarization-induced field focusing effects.

Experimentally, we have added calcium imaging data covering four experimental conditions: ICH(+) WES(+), ICH(+) WES(-), ICH(-) WES(+), and ICH(-) WES(-). These groups allow explicit differentiation between the contribution of the ICH and that of the applied electrical field. The updated results show that significant calcium transients were observed only under the ICH(+) WES(+) condition, whereas no measurable activation occurred in ICH(+) WES(-) or ICH(-) WES(+) groups (Fig. R10a-c). This confirms that both the ICH and applied stimulation are necessary for effective neuromodulation at the cellular level, supporting the proposed mechanism of interfacial polarization at the hydrogel-tissue interface under the volume conduction scenario.

Fig. R10. In vitro cellular stimulation and current density characterization under different experimental configurations. **a**, Time-lapse calcium imaging of SH-SY5Y cells under different conditions. Red asterisks indicate activated cells showing marked fluorescence increases. Scale bar, 50 μ m. **b**, Representative $\Delta F/F_0$ traces showing stimulation-dependent fluorescence enhancement in the ICH(+) WES(+) group. The inset highlights $\Delta F/F_0$ amplitude across all groups (n = 5 independent experiments). **c**, Quantitative comparison of the percentage of excited cells under each condition, confirming that cellular activation occurs exclusively in the ICH(+) WES(+) group (n = 5 independent experiments).

We have also added additional data for electrophysiological experiment, including four groups: ICH(+) WES(-), ICH(+) WES(+), ICH(-) WES(-), and ICH(-) WES(+). As shown in the new data (Fig. R11 and Fig. R12), only the ICH(+) WES(+) group exhibited strong c-Fos expression in the STN region and clear modulation of basal ganglia-cortical activity, with increased firing in the GPi and reduced spiking activity in M1. In contrast, WES alone in the absence of ICH (ICH(-) WES(+)) induced negligible neuronal activation. The corresponding datasets have been incorporated into the updated Fig. 3b-h and Supplementary Figs. To further strengthen the comparison and ensure scientific rigor, we have also added a control group of PD rats implanted with ICH but without WES stimulation [ICH(+) WES(-)]. The new includes quantitative analyses of locomotor distance,

maximum speed, and active time, along with representative weekly behavioral trajectories (as shown in Fig. R13). These results clearly demonstrate that ICH implantation alone or WES alone does not improve locomotor performance, whereas the combination of ICH and WES [ICH(+) WES(+)] leads to significant behavioral recovery. Collectively, these results confirm that ICH plays a critical role in localizing the stimulation field via interfacial polarization under high-frequency volume conduction *in vivo*.

Fig. R11. Control electrophysiological data distinguishing the effects of WES and ICH for Fig. 3. a, Representative immunohistochemical images showing c-Fos expression in the STN region under four experimental conditions. Robust neuronal activation was observed only in the ICH(+) WES(+) group, while negligible c-Fos staining was detected in the other three groups. **b,** Quantitative analysis of c-Fos⁺ cell density per mm². The ICH(+) WES(+) group exhibited significantly higher expression compared with all other groups. **c,** Electrophysiological recordings in GPi under different conditions, showing increased firing in GPi only in the ICH(+) WES(+) group. **d,** Time-frequency spectrograms illustrating oscillatory power (0-100 Hz) across conditions. A marked increase in high-frequency components appeared exclusively with ICH-mediated stimulation. **e,** Raster plots of spike activity over 360 s, showing clear modulation only in the ICH(+) WES(+) group. **f,** Statistical comparison of spike counts in M1. The ICH(+) WES(+) group displayed a reduction in M1 spiking relative to controls, whereas WES alone (ICH(-) WES(+)) produced no significant change (NS). Data are presented as mean ± SD (n = 5 per group); ***p < 0.001; NS, not significant.

Fig. R12. Control electrophysiological data distinguishing the effects of WES and ICH for supporting information. **a**, Power spectral density (PSD) of GPi neuronal activity. Neural activity was recorded in four groups: ICH (+) WES (-), ICH (+) WES (+), ICH (-) WES (-), and ICH (-) WES (+). WES stimulation increased PSD primarily in the 10-50 Hz range in the presence of ICH, indicating that the ICH facilitates modulation of GPi oscillatory activity. **b**, Neural activity recorded in the presence of ICH before and during WES. **c**, Neural activity recorded without ICH before and during WES. Across channels (CH1-CH7), ICH-assisted WES induced more pronounced modulation in signal amplitude and firing patterns compared to WES alone. Scale bar, 20 s, 100 μ V. **d**, Time-frequency spectrogram of M1 neuronal activity with ICH implanted (ICH (+)), WES induces a more temporally structured oscillatory pattern in the 10-50 Hz range compared to the baseline period (ICH (+) WES (-)). **e**, In contrast, without ICH (ICH (-)), WES does not markedly alter the spectral pattern relative to baseline. **f**, Joint inter-spike interval (ISI) distribution of M1 neuronal firing patterns. ICH (+) WES (-): ISI

scatter points show a broad distribution, indicating relatively variable spike timing in the baseline state. ICH (+) WES (+): ISI scatter points become more compact, reflecting increased temporal regularity and reduced spike timing variability following WES when the ICH is present. ICH (-) WES (-): ISI distribution is widely dispersed, indicating irregular and unstructured neuronal firing without ICH or stimulation. ICH (-) WES (+): No pronounced change in ISI distribution is observed compared to baseline, suggesting that WES alone does not reorganize spike timing without the ICH.

Fig. R13. Open-field behavior with the newly added ICH(+) WES(-) control. **a**, Total travelled distance of rats in different groups (n = 5 independent animals). **b**, Max speed of rats in different groups (n = 5 independent animals). **c**, Active time of rats in different groups (n = 5 independent animals). **d**, Representative weekly trajectories illustrating that ICH(+) WES(-) alone does not improve behavior.

Modifications to the manuscript:

(1) Results section:

We have also added a new paragraph on page 13 (marked in red) discussing the expanded calcium imaging results. The revised text reads as: “Electrical signals (wave frequency of 5 MHz, pulse width of 60 μ s, pulse repetition rate of 130 Hz, and amplitude of ± 5 V) generated by ICH in *ex vivo* brain tissue were transmitted via copper interconnects to ITO electrodes to deliver 50 s of stimulation. To clarify the respective roles of the ICH and the applied field, calcium imaging was performed under four experimental conditions: ICH(+) WES(-), ICH(+) WES(+), ICH(-) WES(-), and ICH(-) WES(+). Significant fluorescence transients were detected exclusively in the ICH(+) WES(+) group, exhibiting stimulation-dependent intensity increase (Fig. 2d) with $\Delta F/F_0$ peaking

at ~200% and declining thereafter (Fig. 2e). Quantitatively, the ICH(+) WES(+) group exhibited an eight-fold increase in calcium response compared to controls, with ~45% of SH-SY5Y cells activated versus <5% in non-stimulated groups (Fig. 2f). These results confirm that effective neuromodulation arises only from the combined action of the ICH and the applied electrical field.”.

We have added clarification of the control groups (ICH(+)/(-), WES(+)/(-) to the Results section describing Fig. 3b-h. The new text on page 14-16 in revised manuscript, marked in red, reads as follows:

“All recordings were conducted under the same four-group configuration as used in the calcium imaging to validate that the stimulation mechanism observed at the cellular level also operates in vivo. One week after surgery, WES via capacitive coupling was firstly applied to evaluate thermal safety. Finite element thermal simulations were also conducted under the same stimulation parameters to assess heat distribution within the scalp-skull-brain system (Supplementary Fig. 20a). The results revealed that temperature rise was mainly confined to the external surface, with a maximum increase of approximately 2 °C at the scalp, while the ICH and surrounding deep brain tissue exhibited negligible heating (< 0.3 °C). Infrared thermography further confirmed this trend, showing surface temperature changes consistent with simulation results (Supplementary Fig. 20b, c). These findings indicate that deep brain WES is thermally buffered by surrounding tissues and does not induce detectable internal temperature elevation under the applied stimulation conditions. Only ICH(+) WES(+) induced neuronal activation in the STN, as evidenced by elevated c-Fos expression relative to pre-stimulation (Fig. 3b, c).”.

“These findings suggest that ICH(+) WES(+) activates glutamatergic terminals in the STN, enhancing excitatory signaling to downstream GP^{39,40}.”

“In the M1 region, ICH(+) WES(+) induced neurophysiological changes characterized by reduced neuronal excitability. Microelectrode recordings showed post-stimulation decreases in mean spike amplitude and firing rate, indicating transient cortical suppression (Fig. 3f and Supplementary Fig. 22). Temporal firing analysis revealed a reduction in spike counts (Fig. 3g, h), consistent with reduced neuronal synchrony. Spectrogram analysis (Supplementary Fig. 23) demonstrated a broadband decrease in spectral power during WES, reflecting overall suppression of neuronal activity in M1. ISI mapping (Supplementary Fig. 24) further confirmed reduced spiking frequency and longer inter-spike intervals, indicative of decreased excitability and desynchronized cortical firing. These findings show that ICH(+) WES(+) increases signal

frequency in the GPi while suppressing spiking activity in M1. By contrast, the other control groups [ICH(+) WES(-), ICH(-) WES(-), and ICH(-) WES(+)] showed no detectable electrophysiological modulation, confirming that effective stimulation required both the ICH interface and external WES. This dual effect mirrors key therapeutic mechanisms of conventional DBS, supporting ICH-mediated WES as a minimally invasive strategy for motor modulation^{39,40}.”

We have added clarification of the control groups and corresponding discussions to the Results section on page 17-18 in revised manuscript, marked in red, read as follows:

“Animals were divided into 4 groups: ICH (-) WES (-) (PD model with no treatment), ICH (-) WES (+) group (WES treatment without ICH), ICH (+) WES (-) (implanted with ICH but without WES stimulation), and ICH (+) WES (+) group (ICH-mediated DBS treatment). Rats received 30 min of WES per day, and locomotor activity was assessed weekly using open field testing (Fig. 4a).”.

“ICH-mediated DBS improved locomotor activity, as indicated by increased movement distance, maximum speed, and active time (Fig. 4b and Supplementary Fig. 26). In contrast, the ICH (+) WES (-) and ICH (-) WES (+) groups showed no significant improvement in locomotor activity (Fig. 4c-e). As the essential role of ICH in mediating WES has already been validated at both cellular and acute animal levels, we focused on evaluating the therapeutic outcomes of ICH-mediated wireless DBS in the PD model rather than repeating additional control experiments. Normal rats (control) were used as healthy controls for comparison. We next evaluated neuroprotective effects of ICH-mediated DBS on dopaminergic neurons in the substantia nigra⁴⁴. Immunofluorescence of the substantia nigra pars compacta (SNc) revealed clear differences between ICH (+) WES (+) and ICH (-) WES (+) groups, with normal rats as controls (Fig. 4f, Supplementary Fig. 27, and Supplementary Fig. 28). NeuN⁺ neurons were more abundant and uniformly distributed in the ICH (+) WES (+) group, whereas the ICH (-) WES (+) group showed reduced neuronal density (Supplementary Fig. 27). Iba-1⁺ microglia were sparse across all groups, with no significant difference, likely to reflect the transient nature of microglial activation during acute neuroinflammation (Supplementary Fig. 28).”.

“GFAP⁺ astrocytes were more prominent in the ICH (+) WES (+) group (Supplementary Fig. 29). Quantification revealed a strong positive correlation between GFAP⁺ area and NeuN⁺ neuronal density in the ICH (+) WES (+) group, indicating an association between astrocyte activation and neuronal survival (Fig. 4g). In contrast, no significant correlation was observed in

the ICH (-) WES (+) group. Immunohistochemistry revealed reduced brain-derived neurotrophic factor (BDNF) expression in the ICH (-) WES (+) group, indicating diminished neurotrophic support (Supplementary Fig. 30a). By comparison, BDNF levels were elevated in the ICH (+) WES (+) group following DBS, as evidenced by more intense staining relative to the ICH (-) WES (+) group (Supplementary Fig. 30b). No correlation was observed between GFAP⁺ area and BDNF in the ICH (-) WES (+) group, whereas a strong positive correlation was found in the ICH (+) WES (+) group (Fig. 4h). These findings suggest that GFAP⁺ astrocytes may contribute to sustained neurotrophic support through BDNF, reflecting a potential shift toward a neuroprotective phenotype^{45,46}.”.

“Dopaminergic integrity in the substantia nigra pars compacta (SNc) was assessed by tyrosine hydroxylase (TH) immunolabeling (Fig. 4i). As expected, TH⁺ neurons were significantly reduced in the SNc of the ICH (-) WES (+) group, reflecting PD-associated neurodegeneration.”.

(2) Methods section:

We have updated the description of the cellular stimulation setup and calcium imaging measurement on page 32-33 (marked in red). The revised text reads as:

“Electrical stimulation setup

WES was applied via capacitive coupling using a polyurethane-insulated Au foil as the transmitter electrode. Unless otherwise specified, stimulation was delivered at 5 MHz with a pulse width of 60 μ s and a repetition rate of 130 Hz. For *ex vivo* experiments, the entire brain containing the implanted ICH was used as the receiving unit to preserve the ICH-tissue interface, which is essential for capacitive coupling. The brain-ICH complex was connected via a copper wire to the ITO microelectrodes or ICH-coated ITO to complete the external-internal coupling circuit for wireless stimulation.

Calcium imaging

Microelectrodes were patterned on ITO glass (3 mm width) by HCl etching and insulated with polyimide tape, leaving 3 \times 3 mm² exposed tips for electrical contact. The ITO substrate was integrated with a PLA mold and PDMS to form a custom chamber. The exposed ITO tips were connected via Cu wire to the implanted ICH (in the rat’s brain) for *ex vivo* stimulation. SH-SY5Y cells were stained with Fluo-4 AM (37 °C, 30 min), rinsed with PBS, and imaged using confocal microscopy (FV3000, Olympus). Calcium signals were recorded for 50 s at baseline followed by 50 s during WES (5 MHz, \pm 5 V). Cellular activation was defined as fluorescence intensity changes

($\Delta F/F_0 \geq 50\%$) relative to baseline, a threshold derived from consistent response amplitudes across independent replicates.”.

We have added the details in Methods section on page 34, and page 35 marked in red.

On page 34, the texts read “One week after ICH injection, rats were divided into 2 groups (n = 5 per group), one without WES and another one with WES. Two control groups (n = 5 per group) were included for comparison: rats receiving WES stimulation without ICH implantation (ICH(-) WES(+)) and rats without ICH implantation and without WES stimulation (ICH(-) WES(-)).”.

On page 35, the texts read “To evaluate the effectiveness of ICH-mediated neuromodulation, Rats were euthanized 90 min after 30 min WES, and then the brains were immediately collected. Collected brain tissue samples (n = 20 rats) were fixed in 4% paraformaldehyde, dehydrated, embedded in paraffin, and sectioned for further analysis. The immunohistochemical staining of c-Fos (M00297-6, Bosterbio, dilution 1:250) was employed to evaluate neural activity in the STN regions. One section was collected from each rat, and one representative image was acquired per section for quantification. ImageJ software was employed for counting the c-Fos⁺ neural cells.”.

The sentence “Rats were divided into three groups (n = 9 per group)” has been corrected to: “Rats that passed the screening were divided into 4 groups (n = 5 per group), including: Animals were divided into 4 groups: ICH (-) WES (-) (PD model with no treatment), ICH (-) WES (+) group (WES treatment without ICH), ICH (+) WES (-) (implanted with ICH but without WES stimulation), and ICH (+) WES (+) group (ICH-mediated DBS treatment).”.

(3) Figures:

We have incorporated the new current density data from Fig. R10 into Fig. 2 and Supplementary Fig. 16 and correspondingly revised the figure captions. All changes are marked in red and read as:

Fig. 2. ICH-mediated wireless neural stimulation *in vitro*. **d**, Time-lapse calcium imaging of SH-SY5Y cells under different conditions. Red asterisks indicate activated cells showing marked fluorescence increases. Scale bar, 50 μm . **e**, Representative $\Delta F/F_0$ traces showing stimulation-dependent fluorescence enhancement in the ICH(+) WES(+) group. The inset highlights $\Delta F/F_0$ amplitude across all groups ($n = 5$ independent experiments). **f**, Quantitative comparison of the percentage of excited cells under each condition, confirming that cellular activation occurs exclusively in the ICH(+) WES(+) group ($n = 5$ independent experiments). **g**, Flow cytometry analysis of PC12 cells viability with/without WES. **h**, Apoptosis rate of PC12 cells with and without WES ($n = 3$ independent experiments). Data are presented as the mean \pm standard deviation in (**c**, **e**, **f**, and **h**) and were analyzed by one-way ANOVA first, followed by the Tukey's post hoc test in (**e**, **f**, and **h**). *** $P \leq 0.001$. NS, not significant. p_1 (ICH (+) WES (+) vs ICH (+) WES (-)), p_2 (ICH (-) WES (-) vs ICH (+) WES (-)), p_3 (ICH (-) WES (-) vs ICH (+) WES (+)), p_4 (ICH (-) WES (+) vs ICH (+) WES (-)), p_5 (ICH (-) WES (+) vs ICH (+) WES (+)), p_6 (ICH (-) WES (+) vs ICH (-) WES (-)). **e**, $p_1 = 0.0001$, $p_2 = 0.9991$, $p_3 = 0.0001$, $p_4 = 0.9914$, $p_5 = 0.0001$, $p_6 = 0.9986$. **f**, $p_1 = 0.0001$, $p_2 = 0.9976$, $p_3 = 0.0001$, $p_4 = 0.9878$, $p_5 = 0.0001$, $p_6 = 0.999$. **h**, $p_1 = 0.3787$.

The Fig. R11 have been integrated into Fig. 3b-h, replacing the prior two-group comparison. It reads as follows now:

Fig. 3. ICH-mediated wireless DBS of STN in rat brain. **a**, Experimental setup for wireless DBS, with the ICH applied to the STN region, while monitoring neural activity in the M1 and GPi regions. **b**, Representative images of c-Fos expression in the STN region of rats. Scale bar, 100 μ m. The dark brown nuclei represent c-fos signals, indicating activated neuronal nuclei, while the bluish-purple nuclei correspond to hematoxylin counterstaining, denoting all neuronal nuclei. **c**, Quantification of c-Fos intensity in the STN region (n = 5 independent animals). **d**, Electrophysiological recordings from the GPi region during wireless DBS. **e**, Power

spectrum analysis of neural activity in the GPi region. **f**, Representative electrophysiological waveform from the M1 region during wireless DBS. **g**, Electrophysiological recordings from the M1 region during wireless DBS. **h**, Spike number analysis during wireless DBS (n = 5 independent experiments). **i**, ICH designed for interfacing brain tissues, enables the construction of a biocompatible and seamless interface. **j**, Representative immunostaining images of GFAP and Iba-1 in brain sections with implanted ICH and metal electrodes (tungsten wires with diameters of 500 μm). Scale bar, 50 μm . **k**, Normalized fluorescence intensity of GFAP and Iba-1 in the brain tissues around implants (n = 5 independent animals). Data are presented as the mean \pm standard deviation in (**c**, **h** and **k**) and were analyzed by one-way ANOVA first, followed by the Tukey's post hoc test. ******* $P \leq 0.001$. **NS**, not significant. p_1 (ICH (+) WES (-) vs ICH (+) WES (-)), p_2 (ICH (-) WES (-) vs ICH (+) WES (-)), p_3 (ICH (-) WES (-) vs ICH (+) WES (+)), p_4 (ICH (-) WES (+) vs ICH (+) WES (-)), p_5 (ICH (-) WES (+) vs ICH (+) WES (+)), p_6 (ICH (-) WES (+) vs ICH (-) WES (-)). **c**, $p_1 = 0.0001$, $p_2 = 0.6370$, $p_3 = 0.0001$, $p_4 = 0.8432$, $p_5 = 0.0001$, $p_6 = 0.9813$. **h**, $p_1 = 0.0001$, $p_2 = 0.9539$, $p_3 = 0.0001$, $p_4 = 0.9901$, $p_5 = 0.0001$, $p_6 = 0.9963$. **k**, Iba-1: $p = 3.3646 \times 10^{-5}$ (Metal vs ICH); GFAP: $p = 9.6183 \times 10^{-4}$ (Metal vs ICH).

Fig. R12 has been integrated into Supplementary Fig. 21-24, replacing the prior two-group comparison. It reads as follows now:

Supplementary Fig. 21. Power spectral density (PSD) of GPi neuronal activity. Neural activity was recorded in four groups: ICH (+) WES (-), ICH (+) WES (+), ICH (-) WES (-), and ICH (-) WES (+). WES stimulation increased PSD primarily in the 10-50 Hz range in the presence of ICH, indicating that the ICH facilitates modulation of GPi oscillatory activity.

Supplementary Fig. 22. Multi-channel neural recordings from M1. **a**, Neural activity recorded in the presence of ICH before and during WES. **b**, Neural activity recorded without ICH before and during WES.

Across channels (CH1-CH7), ICH-assisted WES induced more pronounced modulation in signal amplitude and firing patterns compared to WES alone. Scale bar, 20 s, 100 μ V.

Supplementary Fig. 23. Time-frequency spectrogram of M1 neuronal activity. **a**, ICH (+) WES (-) induces a more temporally structured oscillatory pattern in the 10-50 Hz range compared to the baseline period (ICH (+) WES (-)). **b**, ICH (-) WES (+) does not markedly alter the spectral pattern relative to baseline (ICH (-) WES (-)).

Supplementary Fig. 24. Joint inter-spike interval (ISI) distribution of M1 neuronal firing patterns. **a**, ICH (+) WES (-): ISI scatter points show a broad distribution, indicating relatively variable spike timing in the baseline state. **b**, ICH (+) WES (+): ISI scatter points become more compact, reflecting increased temporal regularity and reduced spike timing variability following WES when the ICH is present. **c**, ICH (-) WES (-): ISI distribution is widely dispersed, indicating irregular and unstructured neuronal firing without ICH or stimulation.

d, ICH (-) WES (+): No pronounced change in ISI distribution is observed compared to baseline, suggesting that WES alone does not reorganize spike timing without the ICH.

Fig. R13 has been incorporated as follows: The quantitative open-field metrics (locomotor distance, maximum speed, active time) have been added to replace the corresponding data in Fig. 4 of the main text. The representative locomotor trajectories have been added to Supplementary Fig. 26. To maintain consistency throughout the manuscript and the revised figures, the previous notations “ICH-BES” and “ICH-BES-free” have been uniformly replaced with “ICH(+) WES(+)” and “ICH(-) WES(+)” to align with the terminology used in all newly added data and analyses in Fig. 4 and related Supplementary Figs.

Fig. 4. ICH-mediated wireless DBS alleviates parkinsonian symptoms in PD rats. c, Total travelled distance of rats in different groups (n = 5 independent animals). **d**, Max speed of rats in different groups (n = 5 independent animals). **e**, Active time of rats in different groups (n = 5 independent animals). **f**, Representative immunostaining images for NeuN, Iba-1, and GFAP in SNc with ICH-mediated wireless DBS. Scale bar, 50 μm. **g**, Correlation analysis between GFAP⁺ area and NeuN⁺ cell density in ICH (+) WES (+). The inset shows this relationship in ICH (-) WES (+) group (n = 5 independent animals). **h**, Correlation between GFAP⁺ area and BDNF level in

ICH (+) WES (+) group (n = 5 independent animals). The inset displays this relationship in ICH (-) WES (+) group. **i**, Representative immunohistochemical images of TH⁺ neurons in SNc region. Scale bar, 500 μm (top) and 100 μm (bottom). **j**, Quantification analysis of TH⁺ SNc neurons in different groups (n = 5 independent animals). Data are presented as the mean ± standard deviation in (**c**, **d**, **e**, **j**) and were analyzed by one-way ANOVA first, followed by the Tukey's post hoc test. *P ≤ 0.05, **P ≤ 0.01, ***P ≤ 0.001. **p**₁ (ICH (-) WES (+) vs ICH (-) WES (-)), **p**₂ (ICH (+) WES (+) vs ICH (-) WES (-)), **p**₃ (ICH (+) WES (+) vs ICH (-) WES (+)), **p**₄ (ICH (+) WES (-) vs ICH (-) WES (-)), **p**₅ (ICH (+) WES (-) vs ICH (-) WES (+)), **p**₆ (ICH (+) WES (-) vs ICH (+) WES (+)). **c**, **p**₁ = 1, **p**₂ = 0.0001, **p**₃ = 0.0001, **p**₄ = 0.9692, **p**₅ = 0.9721, **p**₆ = 0.0001. **d**, **p**₁ = 0.3493, **p**₂ = 0.0001, **p**₃ = 0.0001, **p**₄ = 0.4557, **p**₅ = 0.9968, **p**₆ = 0.0001. **e**, **p**₁ = 0.9917, **p**₂ = 0.0001, **p**₃ = 0.0001, **p**₄ = 0.9734, **p**₅ = 0.9990, **p**₆ = 0.0001. **j**, **p** (ICH (-) WES (+) vs Control) = 0.0001, **p** (ICH (+) WES (+) vs Control) = 0.01403, **p** (ICH (+) WES (+) vs ICH (-) WES (+)) = 0.0017.

Supplementary Fig. 26. Open field test of PD rats during 4-week treatment. Trajectory images of open field test for evaluation of the locomotor behaviour of PD rats in different groups. The open field test was conducted in a chamber with an automated video tracking system for 5 min.

Comment #4: L116. *It is stated that improving biocompatibility in the in situ polymerized conductive polymers remains challenging due to toxicity of monomer and crosslinker concentration. However, the proposed prepolymer solution contains Py monomers, which is also*

cytotoxic and can leach into the surrounding tissue. Also, were cells contacted with the gel in the *in vitro* compatibility data of Fig. 2d,g,h?

Response: Although pyrrole monomers can indeed be cytotoxic at high concentrations, the prepolymer solution used in this study contained only 0.22 wt% Py, and polymerization occurred rapidly under physiological conditions through the GOx-HRP cascade. This process led to nearly complete conversion of monomers to PPy, leaving negligible unreacted Py that could leach into surrounding tissue. The absence of inflammatory response and the excellent tissue integration observed in histological analysis support the minimal cytotoxicity of the injected formulation.

Regarding the *in vitro* biocompatibility tests, cells were cultured directly on the hydrogel surface to assess cell adhesion, spreading, and viability. The live/dead staining and metabolic activity (Supplementary Fig. 18 and Supplementary Fig.19) results both indicate excellent cell survival and proliferation, confirming that the ICH exhibits good cytocompatibility after *in situ* polymerization.

Supplementary Fig. 18. Live-dead staining of PC12 cells cultured on ICH under WES with different current intensities. **a**, Representative confocal images of live/dead staining of PC12 cells after 3 days of WES at 5 MHz with increasing current amplitudes (0, 200, 400, and 600 μA). Scale bar, 100 μm. **b**, Quantification of live cell density per mm² (n = 5 independent experiments), showing no significant decrease in viability up to 600 μA, confirming the biosafety of WES within this range. Data are presented as the mean ± standard deviation in (b) and were analyzed by one-way ANOVA first, followed by the Tukey's post hoc test. NS, not significant. **b**, p = 0.6219 (200 μA vs 0 μA), p = 0.7721 (400 μA vs 0 μA), p = 0.9936 (400 μA vs 200 μA), p = 0.9809 (600 μA vs 0 μA), p = 0.4055 (600 μA vs 200 μA), p = 0.5518 (600 μA vs 400 μA).

Supplementary Fig. 2. Viability of PC12 cells cultured on ICH with and without WES. PC12 cells were cultured for 3 days with or without WES. The optical density (OD) at 450 nm shows no significant differences between groups (n = 6 independent experiments). Data are presented as mean ± standard deviation and were analyzed using one-way ANOVA, followed by Tukey’s post hoc test. NS, not significant. p = 0.6421 (day 1: with WES vs without WES), p = 0.6700 (day 2: with WES vs without WES), p = 0.4098 (day 3: with WES vs without WES).

Comment #5: L119. *Have the authors characterized zeta potential of PEDOT:PSS and PPy to conclude the electrostatic assembly in their crosslinking mechanism?*

Response: To directly validate this electrostatic interaction, we performed zeta potential measurements under neutral conditions. PPy and PEDOT:PSS samples were prepared following the same procedure used for ICHs, except that each component was synthesized separately (PPy without PEDOT:PSS and PEDOT:PSS without pyrrole). The dispersions were diluted in deionized water (count rate 200-400 kcps) and tested using a Zetasizer Nano ZS90 (Malvern). The measured zeta potentials were approximately +18 mV for PPy samples and -40 mV for PEDOT:PSS samples (as shown in Fig. R14), clearly demonstrating opposite surface charges and supporting the proposed electrostatic crosslinking mechanism.

Fig. R14. Zeta potential characterization of PEDOT:PSS and PPy dispersions. Zeta potential measurements confirming opposite surface charges of PPy and PEDOT:PSS under neutral aqueous conditions (n = 3)

independent experiments). PPy exhibited a positive zeta potential of approximately +18 mV, while PEDOT:PSS showed a negative zeta potential of approximately -40 mV.

Modifications to the manuscript: We have added a new discussion on page 7 in our revised manuscript, marked in red, to clarify the electrostatic interaction between PPy and PEDOT:PSS. The revised text reads: “To directly verify this electrostatic interaction, zeta potential measurements were performed under neutral conditions. The separately prepared PEDOT:PSS and PPy dispersions showed opposite surface charges (-40 mV and +18 mV, respectively), confirming the charge asymmetry required for electrostatic complexation (Supplementary Fig. 3).”.

The corresponding zeta potential data (Fig. R14) have been added as Supplementary Fig. S3, and the experimental procedure has been described in the Methods section under Zeta potential analysis section (on page 24). The added figure and descriptions read as follows:

Supplementary Fig. S3. Zeta potential characterization of PEDOT:PSS and PPy dispersions. Zeta potential measurements confirming opposite surface charges of PPy and PEDOT:PSS under neutral aqueous conditions ($n = 3$ independent experiments). PPy exhibited a positive zeta potential of approximately +18 mV, while PEDOT:PSS showed a negative zeta potential of approximately -40 mV.

Zeta potential analysis

To verify the electrostatic interaction between PPy and PEDOT:PSS, zeta potential measurements were performed under neutral aqueous conditions. PPy and PEDOT:PSS dispersions were prepared following the same procedures as ICHs, except that only one conductive component (either pyrrole or PEDOT:PSS) was included in each preparation. The resulting suspensions were diluted with deionized water to achieve a count rate of 200-400 kcps before measurement. Zeta potential values were determined using a Zetasizer Nano ZS90 (Malvern) at 25 °C.

Comment #6: L121. Given the small content of conductive polymer in hydrogel formulation, it is important to clarify the roles of these polymers more strongly. Experiments such as conductivity

tests (which is not explained in methods section), where high conductivity is reported, study the role of glucose without control hydrogels that have no intrinsic conductivity (with and without HRP/GOx-HRP) to elucidate the roles of conductive polymers in the measured conductivity and thereby its neuromodulation performance.

Response: The detailed procedure for conductivity measurements is provided in the Methods section, including sample preparation, measurement conditions, and data analysis. To clarify the specific contribution of the conductive polymer network, we performed additional control experiments using precursor solutions without pyrrole monomers, in which polymerization could not occur. These formulations included (i) PEDOT:PSS alone, (ii) PEDOT:PSS with GOx/HRP only, (iii) PEDOT:PSS with glucose only, and (iv) PEDOT:PSS with both GOx/HRP and glucose. As shown in Fig. R15, all non-polymerized controls exhibited low conductivity ($< 6 \text{ S m}^{-1}$), indicating that neither the enzymatic system nor glucose alone contributes significantly to electronic conduction. In contrast, fully polymerized ICHs containing pyrrole, PEDOT:PSS, GOx/HRP, and glucose showed markedly increased conductivity with higher glucose concentrations (original Supplementary Fig. 8), reaching $\sim 30 \text{ S m}^{-1}$ at 5 mM glucose. These results confirm that the high conductivity originates from enzymatic oxidation of pyrrole into polypyrrole, rather than from ionic or enzymatic effects.

Fig. R15. Conductivity of non-polymerized precursor mixtures without pyrrole monomers. It includes PEDOT:PSS alone, PEDOT:PSS with GOx/HRP only, PEDOT:PSS with glucose only, and PEDOT:PSS with both GOx/HRP and glucose, all showing low conductivity ($< 6 \text{ S m}^{-1}$).

Modifications to the manuscript: We have revised the section describing electrical characterization to clarify the control experiments and the origin of conductivity on page 9 in revised manuscript. The added text, marked in red, reads as follows: “To identify the source of electrical conductivity, non-polymerized precursor mixtures without pyrrole monomers were tested as controls, including PEDOT:PSS alone, PEDOT:PSS with GOx/HRP, PEDOT:PSS with

glucose, and PEDOT:PSS with both GOx/HRP and glucose, all showing low conductivity ($< 6 \text{ S m}^{-1}$) (Supplementary Fig. 8a). In contrast, the conductivity of ICHs increased markedly with glucose concentration, reaching $\sim 30 \text{ S m}^{-1}$ at 5 mM (Supplementary Fig. 8b), confirming that the enhanced conductivity arises from the enzymatic oxidation of pyrrole to polypyrrole.”

We have added new descriptions to specify the measurement process for both precursor and polymerized hydrogels on page 25 in revised manuscript, the added text, marked in red, reads as follows: “To evaluate the contribution of individual components, precursor mixtures without pyrrole monomers were also tested, including PEDOT:PSS alone, PEDOT:PSS with GOx/HRP, PEDOT:PSS with glucose, and PEDOT:PSS with both GOx/HRP and glucose. These precursor systems did not undergo polymerization and served as controls.”

We have added new control data (Fig. R15) as Supplementary Fig. 8a to illustrate the conductivity of precursor mixtures and fully polymerized ICHs. The added figure and legend, marked in red, read as follows:

Supplementary Fig. 3. Electrical conductivity of ICHs. **a**, Conductivity of non-polymerized precursor mixtures without pyrrole monomers ($n = 3$ independent experiments), including PEDOT:PSS alone, PEDOT:PSS with GOx/HRP only, PEDOT:PSS with glucose only, and PEDOT:PSS with both GOx/HRP and glucose, all showing minimal conductivity ($< 6 \text{ S m}^{-1}$). **b**, Conductivity of the ICHs increased with higher glucose concentrations, reaching $\sim 30 \text{ S m}^{-1}$ at 5 mM glucose ($n = 3$ independent experiments). Data are presented as the mean \pm standard deviation.

Comment #7: *In vitro and in vivo characterizations of Fig. 2 and 3 mostly compare "with" and "without" stimulation. The role of the developed ICH stimulator developed here is thus unclear. Have stimulation without ICH or other injectable hydrogels with no conductive functionality been tested and compared as controls?*

Response: As explained in our detailed response to Comment #3, we have now added a full set of four-group controls at both the in vitro and in vivo levels to address exactly this point. Specifically, calcium imaging of SH-SY5Y cells was repeated under four conditions: ICH(+) WES(+), ICH(+) WES(-), ICH(-) WES(+), and ICH(-) WES(-). Only the ICH(+) WES(+) group showed large-amplitude Ca^{2+} transients ($\Delta F/F_0$ about 200%) and a high activation ratio (about 45%), whereas the other three groups remained at baseline. This shows that WES alone is insufficient and that the local conductive hydrogel interface is required to convert the volume-conducted field into effective stimulation at the cell surface. The same four-group configuration was then used for acute in vivo recordings in STN-GPi-M1. Again, only ICH(+) WES(+) produced c-Fos induction in STN, increased GPi activity, and reduced M1 spiking, while ICH(-) WES(+) did not produce detectable modulation. These results are consistent with the interfacial polarization mechanism described in Comment #3 and demonstrate that the neuromodulation effect does not arise from WES alone.

Modifications to the manuscript: As detailed in Comment #3.

Comment #8: L129. Please provide UV-vis spectra over the full range up to 1200 nm wavelength to indicate if there is an absorption band suggesting PPy is doped after in situ polymerization and provide ohmic conductivity along with PEDOT:PSS.

Response: The UV-vis measurements in our study were performed to monitor the oxidative polymerization process of Py catalyzed by the enzymatic HRP/ H_2O_2 system, rather than to characterize PEDOT:PSS-PPy composite formation. The purpose of this test was to verify whether Py monomers could be effectively oxidized under the designed biocatalytic conditions, not to evaluate doping interactions with PEDOT:PSS. As shown in Fig. R15, the UV-vis-NIR spectra now extends up to 1100 nm, which corresponds to the instrumental detection limit of our SolidSpec-3700 spectrophotometer. It showed no new absorption peaks after enzymatic polymerization, confirming that this is a simple oxidative polymerization process of pyrrole.

Fig. R16. UV-vis-NIR spectra of polypyrrole polymerized in the presence of glucose. The absorption peak shifts from 300-350 nm to 400-450 nm with increasing glucose concentration, indicating an extension of the conjugation length or alterations in the electronic structure of PPy during polymerization.

Modifications to the manuscript: We have replaced the original Supplementary Fig. 2 with Fig. R16, which now presents the extended UV-vis spectra (200-1100 nm) obtained using the SolidSpec-3700 spectrophotometer.

Comment #9: L138. *Can authors explain how the broader absorption bands of C-O-C and C-H are associated with "enhanced intermolecular interactions and network rearrangement"? Are these not present in glucose? How about PEDOT:PSS controls?*

Response: In the previous submission, the FTIR spectra were collected from samples that might have contained trace amounts of unreacted glucose, which could influence the C-O-C and C-H absorption regions. To eliminate this interference, all ICH samples in the revised experiments were thoroughly dialyzed with deionized water prior to lyophilization, ensuring that the spectra reflect only the hydrogel components. A lyophilized PEDOT:PSS sample without added pyrrole, enzymes, or glucose was used as the control, while the experimental sample corresponded to the ICH (Fig. R17).

In the updated FTIR data, both PEDOT:PSS and ICHs exhibit characteristic C-O-C and C-H vibrations; however, their comparative evolution reveals clear structural differences. In PEDOT:PSS, these bands are sharp and well-defined.

After pyrrole polymerization, the corresponding region in ICHs becomes broader and slightly red-shifted, with the C-O-C/C-H band shifting from $\sim 1220\text{ cm}^{-1}$ to $\sim 1260\text{ cm}^{-1}$, and an additional intensified feature appearing at 1160 cm^{-1} assigned to the symmetric C-O-C stretching of PEDOT. These spectral variations occur together with the emergence of PPy characteristic absorptions near 1560 cm^{-1} and collectively reflect a more heterogeneous local environment caused by new π - π stacking, and conformational rearrangement as PPy co-assembles with PEDOT:PSS. These changes are therefore intrinsic to the polymerization process rather than the result of glucose contamination.

Fig. R17. FTIR spectra of ICHs. FTIR spectra revealing structural changes in PEDOT:PSS due to pyrrole polymerization.

Modifications to the manuscript:

1. Results section:

We have added discussion in results section (page 8) marked in red which reads:

“In the FTIR spectra, the 1560 cm^{-1} band, attributed to the C=C stretching of the conjugated PEDOT/PPy backbone, becomes stronger and broader, suggesting enhanced π - π stacking and charge delocalization²⁹. The 1160 cm^{-1} peak, assigned to the symmetric C-O-C stretching in PEDOT, also intensifies, reflecting stronger interchain coupling. Meanwhile, the C-O-C/C-H vibration shifts from ~ 1220 to 1260 cm^{-1} , implying modified bonding environments and conformational reorganization within the hybrid network³⁰.”

2. Methods section

We have added clarification in the Methods section on page 24 marked in red which reads:

“Commercial PEDOT:PSS (lyophilized) without any added pyrrole, enzymes or glucose was used as the control sample. The experimental sample was the fully formed ICH obtained by enzymatic polymerization of pyrrole in the presence of PEDOT:PSS. Before FTIR and Raman analysis, ICH samples were thoroughly dialyzed with deionized water and lyophilized to remove residual glucose and other small molecules.”.

3. Figures:

We have replaced the original FTIR data in Supplementary Fig. 5 with the updated spectra (now shown as Fig. R17) to reflect the results obtained after dialysis and comparison with the PEDOT:PSS control. The corresponding figure legend has been revised and marked in red, which reads as:

Supplementary Fig. 5. FTIR and Raman spectra of ICHs. a, FTIR spectra revealing structural changes in PEDOT:PSS due to pyrrole polymerization. **b,** Raman spectra showing PPy formation with characteristic C=C/C=N and C-H/C-N bands.

Comment #10: L194. Can authors elaborate on how localization is increased with ICH conductivity?

Response: The enhanced localization of stimulation with increasing ICH conductivity arises from the fundamental electromagnetic behavior in multilayer biological media, particularly the Maxwell-Wagner interfacial polarization effect. When an alternating electric field is applied across the scalp-skull-brain system, the ICH and surrounding brain tissue exhibit markedly different conductivities (σ) and permittivities (ϵ). As the conductivity of the ICH increases, the electric field within the hydrogel decreases, but the contrast in electrical properties at the interface becomes more pronounced. This mismatch enhances discontinuity in the electric displacement field, resulting in greater interfacial charge accumulation and a stronger local field gradient near the ICH boundary.

Consequently, even though the far-field potential across the cortex remains relatively smooth, the electric field and current density become increasingly concentrated around the ICH-tissue interface, leading to stronger and more localized stimulation. In the revised manuscript and Supplementary Information, we have added COMSOL-based finite element simulations comparing electric field, surface charge density, and current density under conditions with and without ICHs and across different ICH conductivities (Fig. R18). The results clearly show that the presence of ICH enhances local charge accumulation and current density in adjacent tissues, and this enhancement increases with conductivity, consistent with the theoretical predictions of Maxwell-Wagner-type interfacial polarization.

Fig. R18. Finite element simulation of interfacial polarization and field localization at the ICH-tissue interface. a, Simulated electric field intensity distribution in the multilayer head model without and with ICHs of different conductivities (3, 10, and 30 S m⁻¹). The introduction of ICHs leads to localized field enhancement around the hydrogel-tissue interface. b, Corresponding surface charge density maps reveal charge accumulation at the ICH boundary, which increases with hydrogel conductivity. c, Local current density distribution showing enhanced conduction pathways near the implanted ICH region, consistent with interfacial polarization-induced field focusing effects.

Comment #11: L401. *Young's modulus compared to the brain is concluded, however no experimental evidence was provided in the results and discussion.*

Response: The mechanical properties of ICH were characterized experimentally, and the results are provided in Supplementary Fig. 7c. The Young's modulus of ICH increased from 360.04 Pa to 495.07 Pa with increasing glucose concentration, indicating enhanced network formation. These values fall within the physiological range of brain tissue (0.1-10 kPa), confirming that ICH possesses tissue-like softness and mechanical compliance comparable to native brain tissue.

Supplementary Fig. 7. Glucose-dependent gelation and mechanical properties of ICHs. c, Young's modulus of ICHs at different glucose contents (n = 3 independent experiments). Data are presented as the mean ± standard deviation in (b, c).

Modifications to the manuscript: We have added a clarifying statement in the Results section on page 9 marked in red which reads: “The corresponding Young’s modulus increased from 360.04 Pa to 495.07 Pa, indicating enhanced network formation (Supplementary Fig. 7c). These values fall within the physiological range of brain tissue (0.1-10 kPa), confirming that ICH possesses tissue-like softness and mechanical compliance comparable to native brain tissue.”.

The calculation method for Young’s modulus has been added to the Methods section on page 24 (marked in red), which reads as:

“And the Young’s modulus (E) was calculated using Equation (1):

$$E = 2\sqrt{G'^2 + G''^2} \cdot (1 + \nu) \quad (1)$$

where G' and G'' represent the storage and loss moduli at 1 Hz, respectively, and ν , representing Poisson’s ratio, is assumed to be 0.5.”.

Comment #12: *Minimally invasive wireless electrostimulation has been claimed in the paper while for the in vivo neuromodulation, 8-channel microwire electrodes were implanted as stated in L447.*

Response: The implanted 8-channel microwire electrodes were used exclusively for electrophysiological recording to verify the neural modulation induced by wireless electrostimulation (as we showed in Fig.3a). These electrodes did not participate in delivering the stimulation itself. Stimulation was delivered noninvasively by applying a high-frequency volume-conducted field, and the impedance mismatch between the injected ICH and surrounding tissue produced interfacial polarization at the hydrogel-tissue interface, thereby modulating local

neurons. In a therapeutic setting, only a micro-syringe injection of the ICH precursor would be required, without any implanted stimulation leads.

Fig. 3. ICH-mediated wireless DBS of STN in rat brain. a, Experimental setup for wireless DBS, with the ICH applied to the STN region, while monitoring neural activity in the M1 and GPI regions.

Modifications to the manuscript: We have revised the Methods section on page 34, and added sentence marked in red which reads “These implanted microwire electrodes served for electrophysiological recording to confirm neural modulation induced by wireless electrostimulation one week after implantation.”.

Reviewer #5

Summary Comments

Yang and colleagues present an excellent demonstration of conducting polymer hydrogels formed in situ for wireless modulation of brain function and Parkinsons' disease therapy. By targeted injection of precursor cocktails, they utilize the brain's own chemistry to drive polymerization and create wireless transducers of AC stimulation fields. They demonstrate that through clever materials and conductivity engineering, they can achieve both biocompatibility as well as bioelectronic therapeutic efficacy. The open field test results (Figure 4b) are particularly compelling! After sufficient revision, as detailed below, I believe this work warrants publication in Nature Communications.

The paper is quite complicated to read. There is an overabundance of references to supplementary notes and figures, some of which are more essential than others for the flow of the manuscript. At the same time, some experimental setups are hard to understand (such as the brain slice experiments in Figure 1 or the ICH-BES-to-in vitro experiments). These would require more elaboration (and perhaps illustration) to be easily reproduced. Beyond the specific comments

below, I believe that the paper needs a thorough revision for flow and rearrangement of SI: some into the main methods section, some integrated into existing figures, some (potentially) removed altogether, and of course some remaining in SI. In particular, the “supplementary notes” are questionably “supplemental”. Perhaps a colleague outside the author list who can give a full reading and provide feedback?

Response: We sincerely thank the reviewer for the positive evaluation of our work and for recognizing our efforts in developing a wireless deep brain stimulation modality based on *in situ* crosslinked PEDOT:PSS hydrogel. We fully agree that the manuscript would benefit from improved flow, clearer presentation of experimental setups, and better organization of the Supplementary Information.

In the revised version, we have thoroughly restructured several sections to enhance clarity and readability. Specifically, we have (1) integrated essential experimental details from the Supplementary Notes into the main Methods section; (2) consolidated overlapping or highly related supplementary figures into the main figures where appropriate; and (3) removed redundant or non-critical SI elements while maintaining all necessary methodological transparency. We have also added schematic illustrations and clarified descriptions of the *in vitro* and *ex vivo* experimental setups to make the procedures easier to understand and reproduce.

We have carefully reviewed the manuscript’s flow and figure referencing to ensure that essential information appears in the main text, while supplementary content now serves its intended supportive role.

Modifications to the manuscript: We have reorganized the Methods and Supplementary Information sections to improve readability. Key experimental details previously located in the Supplementary Notes have been integrated into the main Methods section. Several essential supplementary figures have been merged into the main figures, while redundant or less critical content has been streamlined. The revised version also includes updated schematic diagrams and clearer figure legends to facilitate understanding and reproducibility. All corresponding revisions are marked in red in the updated manuscript.

Comment #1: *On page 6, line 130, the authors discuss electrostatic crosslinking between PPy and PEDOT:PSS. Shouldn’t both materials be neutral in the as-prepared conditions? Why would one*

expect them to be electrostatically crosslinked? Is there direct evidence for this electrostatic interaction?

Response: Both PPy and PEDOT:PSS exhibit intrinsic charge asymmetry under neutral aqueous conditions rather than being electrically neutral. Specifically, PEDOT:PSS carries a net negative charge due to the sulfonate ($-\text{SO}_3^-$) groups of PSS, while PPy, synthesized via enzymatic oxidative polymerization, exists in a positively doped state containing polarons and bipolarons. This difference in charge enables electrostatic complexation between the cationic PPy and anionic PEDOT:PSS, forming the interconnected conductive network observed in ICHs.

To directly validate this electrostatic interaction, we performed zeta potential measurements under neutral conditions. PPy and PEDOT:PSS samples were prepared following the same procedure used for ICHs, except that each component was synthesized separately (PPy without PEDOT:PSS and PEDOT:PSS without pyrrole). The dispersions were diluted in deionized water (count rate 200-400 kcps) and tested using a Zetasizer Nano ZS90 (Malvern). The measured zeta potentials were approximately +18 mV for PPy samples and -40 mV for PEDOT:PSS samples (as shown in Fig. R19), clearly demonstrating opposite surface charges and supporting the proposed electrostatic crosslinking mechanism.

Fig. R19. Zeta potential characterization of PEDOT:PSS and PPy dispersions. Zeta potential measurements confirming opposite surface charges of PPy and PEDOT:PSS under neutral aqueous conditions ($n = 3$ independent experiments). PPy exhibited a positive zeta potential of approximately +18 mV, while PEDOT:PSS showed a negative zeta potential of approximately -40 mV.

Modifications to the manuscript: We have added a new discussion on page 7 in our revised manuscript, marked in red, to clarify the electrostatic interaction between PPy and PEDOT:PSS. The revised text reads: “To directly verify this electrostatic interaction, zeta potential measurements were performed under neutral conditions. The separately prepared PEDOT:PSS and

PPy dispersions showed opposite surface charges (-40 mV and +18 mV, respectively), confirming the charge asymmetry required for electrostatic complexation (Supplementary Fig. 3).”.

The corresponding zeta potential data (Fig. R19) have been added as Supplementary Fig. S3, and the experimental procedure has been described in the Methods section under Zeta potential analysis section (page 24). The added figure and descriptions read as follows:

Supplementary Fig. 3. Zeta potential characterization of PEDOT:PSS and PPy dispersions. Zeta potential measurements confirming opposite surface charges of PPy and PEDOT:PSS under neutral aqueous conditions ($n = 3$ independent experiments). PPy exhibited a positive zeta potential of approximately +18 mV, while PEDOT:PSS showed a negative zeta potential of approximately -40 mV.

Zeta potential analysis

To verify the electrostatic interaction between PPy and PEDOT:PSS, zeta potential measurements were performed under neutral aqueous conditions. PPy and PEDOT:PSS dispersions were prepared following the same procedures as ICHs, except that only one conductive component (either pyrrole or PEDOT:PSS) was included in each preparation. The resulting suspensions were diluted with deionized water to achieve a count rate of 200-400 kcps before measurement. Zeta potential values were determined using a Zetasizer Nano ZS90 (Malvern) at 25 °C.

Comment #2: Page 6, line 132: Please elaborate on the “characteristic microgel structures”. Are these characteristic results for the authors, or a characteristic feature they authors expected to see? What do they imply about ICH formation and the various ICH formulations?

Response: We appreciate the reviewer’s insightful question. The “characteristic microgel structures” refer to the granular morphology that naturally forms when the initially bulk ICH, produced through in situ enzymatic polymerization, is re-dispersed in aqueous or PBS environments. As shown in Supplementary Fig. 6a, the freshly prepared ICH appears as a continuous bulk gel. Upon mild shaking or washing in PBS, the network partially disassembles

into micro-sized gel particles, yielding a microgel-like dispersion as observed under optical microscopy (Supplementary Fig. 3). These structures arise from the electrostatically co-assembled PEDOT:PSS-PPy network. To minimize potential cytotoxicity from unreacted pyrrole monomers, we intentionally used a low pyrrole concentration (0.22 wt%), resulting in a less densely crosslinked network that can partially fragment into microgel domains upon hydration. Therefore, this morphology is a formulation-dependent feature of the ICH system, reflecting the balance between crosslinking density and polymerization extent determined by the monomer content.

Supplementary Fig. 7. Glucose-dependent gelation and mechanical properties of ICHs. a, Photographs of ICHs precursors after incubation in the presence of different glucose concentrations (0, 3, and 5 mM), showing gelation at physiological glucose levels. Scale bar, 1 cm.

Comment #3: *The FTIR and Raman experiments were not clear regarding control experiments and experimental conditions. Please provide details and an explanation of conditions.*

Response: In the previous submission, the FTIR and Raman spectra were collected from samples that might have contained trace amounts of unreacted glucose, which could influence the C-O-C and C-H absorption regions. To eliminate this interference, all ICH samples in the revised experiments were thoroughly dialyzed with deionized water prior to lyophilization, ensuring that the spectra reflect only the hydrogel components. A lyophilized PEDOT:PSS sample without added pyrrole, enzymes, or glucose was used as the control, while the experimental sample corresponded to the ICH.

In the updated FTIR data (Fig. R20a), both PEDOT:PSS and ICHs exhibit characteristic C-O-C and C-H vibrations; however, their comparative evolution reveals clear structural differences. In PEDOT:PSS, these bands are sharp and well-defined. After pyrrole polymerization, the corresponding region in ICHs becomes broader and slightly red-shifted, with the C-O-C/C-H band shifting from $\sim 1220\text{ cm}^{-1}$ to $\sim 1260\text{ cm}^{-1}$, and an additional intensified feature appearing at 1160 cm^{-1} assigned to the symmetric C-O-C stretching of PEDOT. These spectral variations occur

together with the emergence of PPy characteristic absorptions near 1560 cm^{-1} and collectively reflect a more heterogeneous local environment caused by new π - π stacking, and conformational rearrangement as PPy co-assembles with PEDOT:PSS. And Raman spectra corroborated successful glucose-induced polymerization of Py, with characteristic peaks at 1058 cm^{-1} and 1575 cm^{-1} in ICH (Fig. R20b). These peaks correspond to C=C/C=N stretching and C-H deformation/C-N stretching modes of PPy, confirming its incorporation into the PEDOT:PSS network.

Fig. R20. FTIR and Raman spectra of ICHs. a, FTIR spectra revealing structural changes in PEDOT:PSS due to pyrrole polymerization. **b,** Raman spectra showing PPy formation with characteristic C=C/C=N and C-H/C-N bands.

Modifications to the manuscript:

1. Results section:

We have added discussion in results section (page 8) marked in red which reads:

“In the FTIR spectra, the 1560 cm^{-1} band, attributed to the C=C stretching of the conjugated PEDOT/PPy backbone, becomes stronger and broader, suggesting enhanced π - π stacking and charge delocalization²⁹. The 1160 cm^{-1} peak, assigned to the symmetric C-O-C stretching in PEDOT, also intensifies, reflecting stronger interchain coupling. Meanwhile, the C-O-C/C-H vibration shifts from ~ 1220 to 1260 cm^{-1} , implying modified bonding environments and conformational reorganization within the hybrid network³⁰. Raman spectra corroborated successful glucose-induced polymerization of Py, with characteristic peaks at 1058 cm^{-1} and 1575 cm^{-1} in ICH (Supplementary Fig. 5b). These peaks correspond to C=C/C=N stretching and C-H deformation/C-N stretching modes of PPy, confirming its incorporation into the PEDOT:PSS network³¹.”.

2. Methods section

We have added clarification in the Methods section on page 24 marked in red which reads:

“FTIR spectra were recorded in ATR mode using a Nicolet iS50R spectrometer (Thermo Scientific) over 400-4000 cm^{-1} . Raman spectra were obtained using a LabRAM HR800 (Horiba Jobin Yvon) equipped with a 532 nm excitation laser at room temperature. Commercial PEDOT:PSS (lyophilized) without any added pyrrole, enzymes or glucose was used as the control sample. The experimental sample was the fully formed ICH obtained by enzymatic polymerization of pyrrole in the presence of PEDOT:PSS. Before FTIR and Raman analysis, ICH samples were thoroughly dialyzed with deionized water and lyophilized to remove residual glucose and other small molecules.”.

3. Figures:

We have replaced the original FTIR and Raman data in Supplementary Fig. 5 with the updated spectra (now shown as Fig. R20) to reflect the results obtained after dialysis and comparison with the PEDOT:PSS control. The corresponding figure legend has been revised and marked in red, which reads as:

Supplementary Fig. 5. FTIR and Raman spectra of ICHs. a, FTIR spectra revealing structural changes in PEDOT:PSS due to pyrrole polymerization. **b,** Raman spectra showing PPy formation with characteristic C=C/C=N and C-H/C-N bands.

Comment #4: On page 8, line 163, the authors discuss “diminished capacitance” in the EIS data.

But then on line 168, they mention enhanced charge storage capacity. The diminished capacitance is surprising since conducting polymers (especially hydrogel-based ones) usually increase electrode capacitance. Please elaborate? Also, were these measurements done with compressed 2D films? If so, how does EIS translate from this 2D system to the expected 3D in vivo condition?

Response: The term “diminished capacitance” in the original text was intended to describe a reduced capacitive dominance in the phase-angle spectrum rather than a decrease in the overall charge storage capability. The phase-angle reduction at low frequencies indicates a transition from

a capacitive interface, dominated by interfacial polarization, to a mixed conductive-capacitive behavior resulting from enhanced bulk charge transport in the PPy-PEDOT:PSS network. In this regime, resistive conduction through the interpenetrating polymer framework becomes more efficient, leading to improved charge delivery and higher charge storage capacity, as supported by the CV. To avoid misunderstanding, we have revised this wording in the manuscript to clarify the physical meaning.

Regarding the experimental setup, EIS measurements were performed on thin ICH films formed *in situ* on gold electrodes to ensure uniform contact and reproducible spectra (in the Methods section, page 25). Specifically, 5 μL of ICH precursor solution was drop-cast onto a gold electrode (3 mm diameter), evenly dispersed, and allowed to form a stable hydrogel film. Prior to testing, the sample was equilibrated in PBS to maintain full hydration. These films represent the intrinsic ionic and electronic transport characteristics of the hydrogel rather than compressed dry films.

Modifications to the manuscript: We have revised the text on page 9 marked in red, which read: “The phase-angle spectra showed a decrease at low frequencies, indicating that the capacitive behavior of the hydrogel became less dominant (Supplementary Fig. 9b). This reduced capacitive dominance reflects a transition from interfacial polarization-dominated charge storage to more efficient bulk charge transport through the conductive hydrogel network³².”.

Comment #5: *Page 10, lines 199-200: This circuit model could use some justification and/or references*

Response: The equivalent circuit model in Fig. 1f was constructed based on well-established frameworks for capacitive wireless power transfer and tissue-equivalent modeling. Specifically, the frequency-adaptive circuit accounts for (1) the capacitive coupling between transmitter and receiver electrodes across biological tissue as a lossy dielectric medium, (2) multilayer impedance contributions from scalp, skull, and brain, and (3) interfacial polarization at the hydrogel-tissue interface, which influences local charge accumulation and current density distribution. The model structure and parameters were informed by prior studies on capacitive coupling through tissue and dielectric modeling of biological media, as well as frequency-dependent permittivity and conductivity data of layered tissues (*IEEE Transactions on Neural Systems and Rehabilitation Engineering*, 26(5): 1093-1099 (2018), *Electronics Letters*, 51(22): 1806-1807 (2015), *IEEE*

transactions on power electronics, 27(12): 4906-4913 (2012)). Accordingly, we have revised the results to provide justification and appropriate references for the adopted circuit model.

Modifications to the manuscript: We have updated the Results section on page 11 to include a justification and references for the circuit model. The new text, marked in red, reads as: “**To assess biosafety of such a wireless power transfer process, we calculated the specific absorption rate (SAR) using a frequency-adaptive circuit model (Fig. 1f) that incorporates capacitive transmitter dynamics, multilayer tissue impedance, and interfacial polarization elements. This equivalent circuit was adapted from established capacitive wireless power transfer frameworks that model tissue as a frequency-dependent lossy dielectric medium with multilayer impedance characteristics³⁴⁻³⁶. The model integrates frequency-adaptive permittivity and conductivity values for scalp, skull, and brain tissues to reproduce realistic coupling behavior and interfacial charge accumulation under high-frequency electric fields.**”.

Added references:

34. Erfani R, Marefat F, Sodagar AM, Mohseni P. Modeling and Characterization of Capacitive Elements With Tissue as Dielectric Material for Wireless Powering of Neural Implants. *IEEE Transactions on Neural Systems and Rehabilitation Engineering* **26**, 1093–1099 (2018).
35. Huang L, Hu AP. Defining the mutual coupling of capacitive power transfer for wireless power transfer. *Electronics Letters* **51**, 1806–1807 (2015).
36. Theodoridis MP. Effective Capacitive Power Transfer. *IEEE Transactions on Power Electronics* **27**, 4906–4913 (2012).

Comment #6: *Page 10, line 201: conditions of “safe operation” needs a reference.*

Response: The statement regarding “safe operation” has been clarified with reference to internationally recognized electromagnetic exposure standards. Specifically, our results showed that the SAR remained below 2 W kg^{-1} across all tested frequencies ($\leq 5 \text{ MHz}$), which is well within the local SAR limits defined by the IEC 60601-2-33 and FDA guidelines for radiofrequency exposure (local head SAR $\leq 2 \text{ W kg}^{-1}$) (*Journal of Magnetic Resonance Imaging, 26(2): 437-441 (2017)*). These results confirm that the operating conditions used in our experiments fall within the safe exposure range.

Modifications to the manuscript: We have revised the sentence on page 11, marked in red, which now reads: “Across all tested frequencies (≤ 5 MHz), the calculated SAR remained below 2 W kg^{-1} , well within internationally accepted safety limits for local tissue exposure³⁶, indicating safe operation under the capacitive coupling-based volume conduction scenario (Supplementary Fig. 14).”. Reference [37] has been newly added in the revised manuscript and corresponds to the IEC 60601-2-33 and FDA RF exposure guidelines.

Added references:

37. Wang Z, *et al.* SAR and temperature: Simulations and comparison to regulatory limits for MRI. *Journal of Magnetic Resonance Imaging* **26**, 437-441 (2007).

Comment #7: *On page 13, line 262 (and Supp Fig 19), the authors discuss WES without inducing detectable thermal effects. Would the effects of a deep brain WES stimulation actually be detectable from an external measurement like this? Maybe a simple finite element simulation could help here to see if the tissue would buffer (hide) deep thermal changes.*

Response: We agree that surface infrared thermography alone cannot directly detect subtle temperature variations in deep brain regions during WES. To further examine the thermal effects of WES under the experimental conditions, we conducted coupled electromagnetic-thermal finite element simulations (Fig. R20). The results show that in the presence of the ICH, the maximum temperature rise in both the ICH and adjacent brain tissue was minimal ($< 0.3 \text{ }^\circ\text{C}$). In contrast, the scalp exhibited the most noticeable heating effect, with a local maximum temperature increase of approximately $2 \text{ }^\circ\text{C}$ when an insulating layer was present, which is consistent with our infrared thermography measurements. These results indicate that deep tissue heating during WES is effectively buffered by surrounding biological layers, while the observed surface temperature change mainly reflects minor heating at the scalp. Based on both simulation and experimental data, we confirm that our conclusion regarding the absence of detectable thermal effects under the applied stimulation conditions is well supported.

Fig. R20. Finite element simulation of temperature distribution in the scalp-skull-brain model after 10-, 20-, and 30-minute WES, showing negligible temperature elevation ($< 2\text{ }^{\circ}\text{C}$).

Modifications to the manuscript:

(1) Figures:

We have added the new simulation data (Fig. R20) as Supplementary Fig. 20a and corresponding description in the revised Supplementary Information, with the text marked in red, which reads as follows:

Supplementary Fig. 20. Thermal effects of WES. **a**, Finite element simulation of temperature distribution in the scalp-skull-brain model after 10-, 20-, and 30-minute WES, showing negligible temperature elevation ($< 2\text{ }^{\circ}\text{C}$). **b**, Photograph of the experimental setup for WES, showing the Au electrode insulated by PU and grounding configuration during stimulation. **c**, Infrared thermographic images of the rat head before and during WES (10 min, 20 min, 30 min), showing minimal change in surface temperature (from $33.8\text{ }^{\circ}\text{C}$ to $35.0\text{ }^{\circ}\text{C}$). Scale bars, 1 cm.

(2) Results section:

We have also added corresponding discussion on pages 14-15 of the revised manuscript, with the text marked in red, which reads “All recordings were conducted under the same four-group configuration as used in the calcium imaging to validate that the stimulation mechanism observed at the cellular level also operates in vivo. One week after surgery, WES via capacitive coupling was firstly applied to evaluate thermal safety. Finite element thermal simulations were also conducted under the same stimulation parameters to assess heat distribution within the scalp-skull-brain system (Supplementary Fig. 20a). The results revealed that temperature rise was mainly

confined to the external surface, with a maximum increase of approximately 2 °C at the scalp, while the ICH and surrounding deep brain tissue exhibited negligible heating (< 0.3 °C). Infrared thermography further confirmed this trend, showing surface temperature changes consistent with simulation results (Supplementary Fig. 20b, c). These findings indicate that deep brain WES is thermally buffered by surrounding tissues and does not induce detectable internal temperature elevation under the applied stimulation conditions.”.

(3) Methods section:

We have added the following statement in Methods section (page 30), marked in red:

“Finite element modeling

A three-dimensional finite element model was constructed using COMSOL Multiphysics® to simulate electric field distributions within a multilayer cylindrical system (Supplementary Fig. 12). The geometry consisted of six vertically stacked components: (1) a base rectangular chamber (20 mm × 20 mm × 20 mm) filled with a conductive medium representing brain tissue; (2) a 1 mm-thick rectangular shell simulating the skull; (3) a 1 mm-thick outermost rectangular shell representing scalp tissue; (4) a 1.7 mm-diameter spherical inclusion, mimicking the ICHs, positioned 7.5 mm above the grounded base along the central vertical axis; (5) a 12.5 mm-diameter insulating barrier (0.1 mm thick) coating on the tissue scalp layer; and (6) a 10 mm-diameter coaxial gold foil electrode placed atop the insulating layer. Boundary conditions were defined by applying a 5 MHz sinusoidal voltage (± 2.5 amplitude, 0° phase) to the gold electrode, with the chamber’s bottom surface grounded (0 V). All other exterior boundaries were electrically insulated. The AC/DC module with the electric currents interface was employed for transient solver analysis. In addition, the Solid Heat Transfer physics interface was added and the model was extended to a coupled multiphysics study by enabling the Electromagnetic Heating coupling so that electromagnetic losses produced by the applied AC fields are passed directly into the heat-transfer simulation. A transient solver was used to compute the electromagnetic and thermal response simultaneously: snapshots of changes in electric field, current, and charge are shown at 0.06 μ s, while the temperature evolution is presented at 30 min. A tetrahedral mesh with local refinement was applied to the ICHs, insulating layer, and gold electrode regions to ensure numerical convergence and solution accuracy. Surface charge density simulation results indicate that the maximum surface charge density on the ICHs approached 20 nC cm⁻², larger than the 15 nC cm⁻² threshold required to initiate neural activation.”.

Comment #8: Page 13, line 267-268: Why do these findings suggest that WES activates glutamatergic terminals in the STN? What leads to the authors to this conclusion? And some references here, please.

Response: The conclusion that WES activates glutamatergic terminals in the subthalamic nucleus (STN) is based on both electrophysiological patterns and well-established basal ganglia circuitry. In the canonical cortico-STN-GPi pathway, excitatory glutamatergic projections from the STN synapses onto GPi neurons, increasing their firing rate and action potential amplitude (*The Neuroscientist*, 22, 332-345 (2015), *Journal of Neurophysiology*, 115, 19-38 (2015)). The elevated firing rates and enhanced spectral power in the 10-50 Hz range observed in GPi during WES are consistent with increased excitatory (glutamatergic) input from the STN. To strengthen this interpretation, we have added appropriate references describing the excitatory glutamatergic projections from STN to GPi and their role in modulating GPi activity during electrical stimulation.

Modifications to the manuscript: On page 15, the sentence has been revised (marked in red) to read as follows:

“These findings suggest that WES activates glutamatergic terminals in the STN, enhancing excitatory signaling to downstream GPi^{40,41}.”

Added references:

40. Florence G, Sameshima K, Fonoff ET, Hamani C. Deep Brain Stimulation: More Complex than the Inhibition of Cells and Excitation of Fibers. *The Neuroscientist* **22**, 332-345 (2015).
41. Herrington TM, Cheng JJ, Eskandar EN. Mechanisms of deep brain stimulation. *Journal of Neurophysiology* **115**, 19-38 (2015).

Comment #9: Page 13, lines 274-275: Elaborate on (justify) spectral analysis confirming changed frequency behavior. I don't see an observable difference in Supp Fig 22 before and after 180 s, when WES was initiated.

Response: We agree that the previous version of the time-frequency spectrogram (previously Supplementary Fig. 22) did not clearly reflect the temporal evolution of neuronal activity, making the frequency-dependent behavior difficult to discern. To address this, we have reanalyzed the original electrophysiological data and updated the figure (now Supplementary Fig. 23) to enhance visualization of temporal dynamics.

The revised analysis reveals a reduction in overall spectral power during WES, which corresponds to a decrease in the firing rate of M1 neurons. This finding is consistent with the raster and ISI distribution data (Supplementary Fig. 24), both of which demonstrate sparser and slower spiking activity under stimulation. These results collectively indicate that ICH-mediated WES (ICH(+) WES (+)) suppresses hyperactivity in M1, leading to a lower firing rate rather than a shift in frequency band dominance.

Supplementary Fig. 23. Time-frequency spectrogram of M1 neuronal activity. a, With ICH implanted (ICH (+)), WES induces a more temporally structured oscillatory pattern in the 10-50 Hz range compared to the baseline period (ICH (+) WES (-)). **b,** In contrast, without ICH (ICH (-)), WES does not markedly alter the spectral pattern relative to baseline.

Supplementary Fig. 24. Joint inter-spike interval (ISI) distribution of M1 neuronal firing patterns. **a**, ICH (+) WES (-): ISI scatter points show a broad distribution, indicating relatively variable spike timing in the baseline state. **b**, ICH (+) WES (+): ISI scatter points become more compact, reflecting increased temporal regularity and reduced spike timing variability following WES when the ICH is present. **c**, ICH (-) WES (-): ISI distribution is widely dispersed, indicating irregular and unstructured neuronal firing without ICH or stimulation. **d**, ICH (-) WES (+): No pronounced change in ISI distribution is observed compared to baseline, suggesting that WES alone does not reorganize spike timing without the ICH.

Modifications to the manuscript: We have replaced the original spectral analysis (previous Supplementary Fig. 22) with the updated Supplementary Fig. 23, which presents the reprocessed time-frequency data for all experimental groups under consistent conditions. Accordingly, the original sentence describing “increased high-frequency oscillations” has been revised to more accurately reflect the observed data. The updated text on page 16 now reads: “**Spectrogram analysis (Supplementary Fig. 23) demonstrated a broadband decrease in spectral power during WES, reflecting overall suppression of neuronal activity in M1.**”.

Comment #10: Page 14, line 280: Need references on “mechanisms of conventional DBS”.

Response: We have added appropriate references to support the statement on the mechanisms of conventional DBS. These references describe established electrophysiological mechanisms of DBS, including suppression of pathological oscillations and restoration of normal firing patterns in motor circuits (*The Neuroscientist* 22, 332–345 (2015), *Journal of Neurophysiology* 115, 19–38 (2015)).

Modifications to the manuscript: On page 15, the sentence has been revised (marked in red) to read as follows:

“These findings show that ICH-mediated WES increases signal frequency in the GPi while suppressing spiking activity in M1. This dual effect mirrors key therapeutic mechanisms of conventional DBS, supporting ICH-mediated WES as a minimally invasive strategy for motor modulation^{40, 41}.”.

Added references:

40. Florence G, Sameshima K, Fonoff ET, Hamani C. Deep Brain Stimulation: More Complex than the Inhibition of Cells and Excitation of Fibers. *The Neuroscientist* 22, 332–345 (2015).

41. Herrington TM, Cheng JJ, Eskandar EN. Mechanisms of deep brain stimulation. *Journal of Neurophysiology* **115**, 19–38 (2015).

Comment #11: *In general, the manuscript would be significantly strengthened – and made clearer to put in context – if the authors used more literature references comparing their ICH-BES results with conventional DBS. Tissue/immune response; fMRI effects; reorganization of brain networks; neuroprotectivity; etc. Basically, many/all results from the bottom of page 14 could use this context. I believe this is essential to proceed to publication in Nature Communications.*

Response: We fully agree that placing our ICH-BES results in the context of conventional deep brain stimulation (DBS) would strengthen the manuscript and clarify its broader significance. In the revised version, we have added multiple literature references on page 15, page 16, and page 18-19 to explicitly compare our findings with key aspects of DBS, including tissue response, neuroprotective effects, and fMRI-based functional network modulation. These additions highlight how ICH-BES reproduces essential therapeutic mechanisms of DBS while offering advantages in biocompatibility and minimally invasive operation.

Modifications to the manuscript: We have added several references on pages 15-16 in the revised manuscript (marked in red), which now read as follows:

“These findings suggest that WES activates glutamatergic terminals in the STN, enhancing excitatory signaling to downstream GPi^{40, 41}”.

“This dual effect mirrors key therapeutic mechanisms of conventional DBS, supporting ICH-mediated WES as a minimally invasive strategy for motor modulation^{40, 41}”.

“Tungsten wires (500 µm diameter), a rigid neural interface with known chronic inflammatory effects, served as the positive control⁴²”.

“In contrast, modulus-matched ICH were expected to reduce interfacial strain and attenuate mechanotransduction-related inflammatory signaling⁴³”.

We have added several references on page 18 in the revised manuscript (marked in red), which now reads: “These findings suggest that GFAP⁺ astrocytes may contribute to sustained neurotrophic support through BDNF, reflecting a potential shift toward a neuroprotective phenotype^{46, 47}”.

We have added several references on page 19-21 in the revised manuscript (marked in red), which now read as follows:

“Conventional metal electrodes are MRI-incompatible due to safety risks, including RF-induced heating and magnetic displacement, as well as imaging artifacts that obscure brain anatomy⁵¹.”

“Multilevel analysis was performed to identify region-specific responses in key motor-related areas, including the striatum (Str), motor cortex (Mtr), cingulate cortex (Cg), prefrontal cortex (PL), thalamus (THL), and midbrain (Raphe)^{52, 53}.”

“These strengthened connections suggest improved motor pathway integration, consistent with the broader functional reorganization observed in treated animals^{54, 55}.”

Added references:

40. Florence G, Sameshima K, Fonoff ET, Hamani C. Deep Brain Stimulation: More Complex than the Inhibition of Cells and Excitation of Fibers. *The Neuroscientist* **22**, 332–345 (2015).
41. Herrington TM, Cheng JJ, Eskandar EN. Mechanisms of deep brain stimulation. *Journal of Neurophysiology* **115**, 19–38 (2015).
42. Prasad A, *et al.* Comprehensive characterization and failure modes of tungsten microwire arrays in chronic neural implants. *Journal of Neural Engineering* **9**, 056015 (2012).
43. Yuk H, Lu B, Zhao X. Hydrogel bioelectronics. *Chemical Society Reviews* **48**, 1642–1667 (2019).
46. Palasz E, *et al.* BDNF as a Promising Therapeutic Agent in Parkinson's Disease. *Int. J. Mol. Sci.* **21**, 1170 (2020).
47. Wu N, Sun X, Zhou C, Yan J, Cheng C. Neuroblasts migration under control of reactive astrocyte-derived BDNF: a promising therapy in late neurogenesis after traumatic brain injury. *Stem Cell Res. Ther.* **14**, 2 (2023).
51. Driscoll N, *et al.* MXene-infused bioelectronic interfaces for multiscale electrophysiology and stimulation. *Sci. Transl. Med.* **13**, eabf8629 (2021).
52. Boutet A, *et al.* Predicting optimal deep brain stimulation parameters for Parkinson's disease using functional MRI and machine learning. *Nat. Commun.* **12**, 3043 (2021).
53. Subramanian L, *et al.* Real-time functional magnetic resonance imaging neurofeedback for treatment of Parkinson's disease. *J. Neurosci.* **31**, 16309–16317 (2011).
54. Filippi M, Sarasso E, Agosta F. Resting-state Functional MRI in Parkinsonian Syndromes. *Mov. Disord. Clin. Pract.* **6**, 104–117 (2019).

55. Pelled G, Bergman H, Ben-Hur T, Goelman G. Manganese-enhanced MRI in a rat model of Parkinson's disease. *J. Magn. Reson. Imaging* **26**, 863–870 (2007).

Comment #12: Page 20, line 407: The authors state that “a simple wearable” can control their ICH-BES system. Elaborate on the translatability of this technique. How would it look in human patients? Does the control electrode need to be on the brain, or would the scalp suffice?

Response: In our system, the “simple wearable” refers to a noninvasive scalp-mounted stimulator that delivers high-frequency signals via thin gold (Au) electrodes used as capacitive coupling transmitters. In practice, there is insulated soft Au foil placed on the scalp surface; no electrode is placed on the brain. The implanted ICH functions as a local charge-accumulating interface, enabling focal neuromodulation through volume conduction under transcranial capacitive coupling, where scalp electrodes suffice to drive localized polarization at the ICH-tissue boundary. This configuration, already illustrated in the schematic and photographic images provided in the manuscript (Fig. 3a and Supplementary Fig. 10).

Fig. 3a. Experimental setup for wireless DBS, with the ICH applied to the STN region, while monitoring neural activity in the M1 and GPi regions.

Supplementary Fig. 20b. Photograph of the experimental setup for WES, showing the Au electrode insulated by PU and grounding configuration during stimulation.

Regarding human translation, the same Au transmitter electrodes can be integrated into a conformal scalp patch with medical-grade adhesive and insulation. Control is achieved by adjusting output amplitude and frequency at the wearable, while the ICH focuses on the field locally through interfacial polarization. Thus, the control electrode does not need to be on the brain; scalp placement is sufficient.

Comment #13: *The measurement setup described Figure 2a and in Supplementary Note 2, page 8, around lines 148-152, really needs a better illustration than the tiny blue balls and wires in Figure 2a. Perhaps a zoom in Figure 2a? Or a standalone additional supplementary figure? Furthermore, regarding Figure 2a, where the working (input) electrode placed? This is essential for easily understanding Figure 2b and c.*

Response: The *ex vivo* setup in Fig. 2a was designed to replicate the high-frequency volume conduction scenario established under *in vivo* capacitive coupling. To clarify, the scalp, skull, and brain tissues were harvested together from freshly euthanized SD rats, gently rinsed with PBS, and reassembled in their anatomical order. A polyurethane-encapsulated gold foil electrode (10 mm diameter) was placed on the scalp surface to deliver the high-frequency alternating field, thereby reproducing the wearable stimulation mode used in animal experiments (as shown in Fig. 3a).

Fig. 3a. Experimental setup for wireless DBS, with the ICH applied to the STN region, while monitoring neural activity in the M1 and GPi regions.

For electrical measurements, fine copper leads were inserted into either the implanted ICH or the adjacent brain tissue to record voltage and current responses under controlled input excitation.

This configuration enabled quantitative evaluation of potential distribution and current focusing within the conductive tissue domain. The corresponding setup and electrode placement are now clearly described in the Methods section (page 31). We also include photographs of the actual experimental arrangement for better visualization here (Fig. R21). As shown in Fig. 18a, the scalp, skull, and brain were reassembled in their physiological order, and the tissues were partially sectioned to expose the internal electrode connections for visualization. The fine copper wires inserted into the brain and hydrogel regions were used to record local voltage and current responses. Fig. 18b displays the intact reconstructed tissue complex, showing the polyurethane-encapsulated gold foil electrode placed on the scalp surface for capacitive coupling stimulation, faithfully reproducing the *in vivo* configuration used in animal experiments.

Fig. R21. Ex vivo setup for electrical characterization of ICH. **a**, Partially dissected brain-skull-scalp tissue showing the polyurethane-insulated gold (Au) electrode positioned on the scalp surface and copper wires inserted into the implanted ICH and adjacent brain tissue for voltage and current recording. The basal copper plate served as the grounding path. **b**, Reassembled intact brain-skull-scalp complex demonstrating the placement of the Au electrode, replicating the *in vivo* stimulation geometry.

Modifications to the manuscript: We have expanded the Methods section (page 30-31, marked in red) to detail tissue preparation, electrode placement, and voltage/current measurement procedures, reads as follows:

“Measurement of the electrical outputs

Brain tissue, skull, and scalp harvested from freshly euthanized adult SD rats were rinsed in phosphate-buffered saline (PBS, pH 7.4) to remove residual blood and connective tissue. The dissected skull and scalp were anatomically reassembled over the cerebral surface, followed by placement of a polyurethane-encapsulated gold foil electrode (10 mm diameter) on the reconstructed tissue complex. Electrical grounding was established at the basal cerebral surface. Voltage and current transmission characteristics were assessed using a sinusoidal signal (± 2.5 V

output) applied to the gold foil (wearable electrode) using a function generator (Tektronix, AFG3021C) and a power amplifier (ATA-1200B, Aigtek). Voltage was recorded via a parallel circuit using an oscilloscope (DHO1204, RIGOL) across the brain, while current was measured in series using a current amplifier (OE4102, Sine Scientific Instruments) between the brain base and ground. Frequency-dependent responses were evaluated from 100 kHz to 7 MHz.

For *ex vivo* electrical output measurements of ICH, the cerebral hemispheres were bisected along the central sulcus, and 2.5 μL of ICH were injected into the region at the same depth as the STN in the right hemisphere. Two insulated copper wires with exposed tips were implanted: one within the ICH and the other laterally in the adjacent brain tissue. U_1 were recorded using an oscilloscope under 5 MHz excitation at incremental input voltages (0.5-5 V, 0.5 V steps). At the same time, U_2 was assessed by inserting electrodes below the ICH maintaining identical inter-electrode spacing parameters to those employed in the U_1 measurement protocol. To evaluate current focusing, two ICHs (1.25 μL each) were injected into the right STN region, separated by a 5 μm polyurethane insulation barrier. One insulated copper wire was implanted into each ICH, and I_1 was quantified using the OE4102 amplifier under 5 MHz excitation with stepwise voltage increments (0.5-5 V, 0.5 V steps). I_2 was measured in series between the cerebral base and ground. I_3 was measured between the center of the ICHs and the adjacent brain tissue. I_4 was measured as a control for I_1 , with electrodes placed in brain tissue but separated by an insulating PU membrane, mimicking an interface without ICH-mediated conduction. Current densities (J , mA cm^{-2}) were calculated using Equation 13.”.

Comment #14: *In Supplementary Note 1’s sections on electrical and electrochemical characterization of ICHs (SI page 5), they describe determination of conductivity and CSC. Were these measurements done using the 2D “compressed” ICHs? Would one expect that conductivity assessment and CSC would translated to the fully hydrated 3D in vivo ICHs?*

Response: For conductivity testing, the ICH initially forms as a microgel dispersion, which cannot maintain a stable geometry required for four-point probe measurement. To obtain reproducible contact and defined dimensions, the ICH samples were lyophilized and gently compressed into uniform thin films. Although this procedure does not reproduce the hydrated 3D state, it enables standardized comparison of intrinsic bulk conductivity among different formulations. The

measured values therefore serve as relative indicators of conductive network efficiency rather than absolute *in vivo* conductivity.

Regarding the electrochemical characterization (EIS, CV, CSC), these measurements were conducted on fully hydrated ICHs formed *in situ* on gold electrodes, which reflect the interfacial charge delivery behavior under physiological conditions and better represent the *in vivo* state. As described in the Methods section, this procedure was already specified to ensure that the electrochemical testing conditions correspond to the hydrated state of the ICHs.

Thus, the four-point probe test provides a comparative metric of bulk conductivity across formulations, whereas the electrochemical measurements capture hydrated charge transport and storage relevant to the *in vivo* performance of the ICH.

Modifications to the manuscript: We have added one sentence in the Methods section (page 26), which reads “For specimen preparation, 5 μ L of ICH precursor solution was drop-cast onto a gold electrode (3 mm diameter), evenly dispersed, and cured to form stable films. Prior to testing, the sample was equilibrated in PBS.”.

Comment #15: *The finite element modeling model description (Supplementary Note 2, pages 9-10) could use a diagram showing the model. The text description is difficult to follow. Perhaps a new Supp Fig, or slightly more detail in, e.g., Figure 1e.*

Response: We agree that a visual representation would improve clarity. Accordingly, we have added a schematic diagram illustrating the finite element model geometry and boundary setup (Fig. R22). This schematic depicts the multilayer structure of the scalp, skull, and brain, along with the placement of the insulated transmitter and the implanted ICH, helping readers to better understand the model configuration described in the text.

Fig. R21. Finite element model of the scalp-skull-brain system for WES. Schematic of the 3D model constructed for simulating electric-field and thermal distributions during WES. The model consists of a multilayer structure representing scalp, skull, and brain tissues, with an insulated metal transmitter and an implanted ICH located in the brain tissue.

Modifications to the manuscript:

(1) Figures:

Fig. R21 has been provided as Supplementary Fig. 12 in the revised Supporting Information, which reads as follows.

Supplementary Fig. 12. Finite element model of the scalp-skull-brain system for WES. Schematic of the 3D model constructed for simulating electric-field and thermal distributions during WES. The model consists of a multilayer structure representing scalp, skull, and brain tissues, with an insulated metal transmitter and an implanted ICH located in the brain tissue.

(2) Methods section:

We have added the following statement in Methods section (page 30), marked in red:

“Finite element modeling

A three-dimensional finite element model was constructed using COMSOL Multiphysics® to simulate electric field distributions within a multilayer cylindrical system (Supplementary Fig. 12). The geometry consisted of six vertically stacked components: (1) a base rectangular chamber (20 mm × 20 mm × 20 mm) filled with a conductive medium representing brain tissue; (2) a 1 mm-thick rectangular shell simulating the skull; (3) a 1 mm-thick outermost rectangular shell representing scalp tissue; (4) a 1.7 mm-diameter spherical inclusion, mimicking the ICHs, positioned 7.5 mm above the grounded base along the central vertical axis; (5) a 12.5 mm-diameter insulating barrier (0.1 mm thick) coating on the tissue scalp layer; and (6) a 10 mm-diameter coaxial gold foil electrode placed atop the insulating layer. Boundary conditions were defined by applying a 5 MHz sinusoidal voltage (± 2.5 amplitude, 0° phase) to the gold electrode, with the chamber’s bottom surface grounded (0 V). All other exterior boundaries were electrically insulated. The AC/DC module with the electric currents interface was employed for transient solver analysis. In addition, the Solid Heat Transfer physics interface was added and the model was extended to a coupled multiphysics study by enabling the Electromagnetic Heating coupling so that electromagnetic losses produced by the applied AC fields are passed directly into the heat-transfer simulation. A transient solver was used to compute the electromagnetic and thermal response simultaneously: snapshots of changes in electric field, current, and charge are shown at 0.06 μs , while the temperature evolution is presented at 30 min. A tetrahedral mesh with local refinement was applied to the ICHs, insulating layer, and gold electrode regions to ensure numerical convergence and solution accuracy. Surface charge density simulation results indicate that the maximum surface charge density on the ICHs approached 20 nC cm^{-2} , larger than the 15 nC cm^{-2} threshold required to initiate neural activation.”.

(3) Results section:

We have added the following statement in revised manuscript (page 11), marked in red: “Finite element modeling was then performed to simulate interfacial polarization and field localization in a scalp-skull-brain model with/without ICH (2.5 μL , ~ 1.7 mm diameter) (Supplementary Fig. 12).”.

Comment #16: Likewise, the experimental setup with ICH-BES in brain connected to the *in vitro* setup (Supplementary Note 3, page 11, lines 202-204) could use a schematic drawing.

Response: The *in vitro* stimulation setup connecting the ICH in the brain to the cell culture chamber was designed to replicate the same volume-conduction pathway as in *in vivo* experiments (as shown in Fig. R21 in comment #13). The copper wires extending from the ICH were connected to patterned ITO electrodes (through the electrical contact points) (Supplementary Fig. 17) fabricated on glass substrates, which served as electrodes for *in vitro* stimulation. It shows the fabrication and assembly of the ITO-based stimulation platform. The process involves sequential HCl etching, PI-tape insulation, and bonding with a PDMS-PLA culture chamber. This setup ensures that the ICH-ITO coupling accurately models the electrical configuration used in animal experiments, while maintaining compatibility with live-cell culture conditions.

Supplementary Fig. 17. Fabrication of the cell culture chamber with patterned ITO electrodes. The process begins with ITO glass, which is etched using HCl to create patterned ITO electrodes (3 mm wide). The electrodes are then insulated with PI tape, leaving exposed tips (3 mm × 3 mm) for electrical contact. The insulated ITO pattern is subsequently combined with a PLA mold and PDMS to form the final cell culture chamber.

Modifications to the manuscript: We have annotated Supplementary Fig. 17 to explicitly indicate the electrical contact points where the copper wires from the ICH were connected to the patterned ITO electrodes.

Reviewer's Comments:

Reviewer #1

Summary Comments

The authors have made a significant effort to address the comments raised. Although some of the main issues were indeed addressed, mainly the proper controls in most cases, there are still a few issues that should be addressed.

Response: We sincerely thank you for acknowledging our substantial revisions and for the constructive suggestions that have further improved the manuscript. All remaining concerns have now been fully addressed, with specific changes made in response to each of the comments. These revisions have also been reflected in the revised manuscript.

Comment #1: *Figure 1e still shows that although the ICH can generate some local enhancement of the electrical stimulation around it, the top part of the brain (cortex most likely) is still being stimulated. So the claims of targeting specific regions in the brain seem invalid in the current setting. See also in comment 3 below.*

Response: The stimulation mechanism is based on interfacial polarization under volume conduction, where neuronal activation occurs only when localized electric field concentration is generated at a strong conductivity/permittivity discontinuity. The hydrogel interface, which has high conductivity and low permittivity, causes a significant contrast when compared to the surrounding brain tissue, which has low conductivity and high permittivity. This mismatch in material properties leads to interfacial charge accumulation and localized electric field concentration at the ICH-tissue boundary. Such field localization is essential for achieving the focused neuronal depolarization required for deep brain stimulation in volume conduction scenario.

To clarify the observation in Fig. 1e, the bright coloration near the cortical surface primarily reflects charge accumulation at the insulation-air interface of the external transmitter electrode rather than strong stimulation within cortical tissue. To experimentally validate the spatial distribution of the applied field, we performed additional voltage (U_3) and current density (J_5) measurements in the cortex without the ICH, while maintaining the same relative electrode spacing as for the corresponding deep-brain measurements (U_2 and J_4) (Fig. R1a). The cortical and deep measurements exhibited comparable voltage and current density (Fig. R1b, c), indicating that the volume-conducted field distributes broadly across tissue layers. Therefore, even though the cortex

is closer to the transmitter and sits at a higher absolute potential, the field there remains spatially diffuse, resulting in large-area but weak stimulation that is insufficient to depolarize neurons.

In response to your suggestion, we have also included additional data (Fig. R1d, e) showing c-Fos expression in the cortex. These data reveal that, despite the enhanced electric field in the cortex, there is no significant neuronal activation in this region. This confirms that the electric field in the cortex is not sufficiently localized to directly trigger neuronal activation.

Fig. R1. Quantitative validation of cortical responses under WES. **a**, Schematic of the measurement configuration for cortical voltage (U_3) and current density (J_5), using the same relative electrode spacing as the deep-brain measurements (U_2 and J_4). **b**, Voltage measured in cortex (U_3) and deep brain (U_2) under different input voltages ($n = 5$ independent experiments). **c**, Corresponding current density in cortex (J_5) and deep brain (J_4) under different input voltages ($n = 5$ independent experiments). **d**, Representative images of c-Fos expression in the cortical region of rats. Scale bar, 100 μm . The dark brown nuclei represent c-Fos signals, indicating activated neuronal nuclei, while the bluish-purple nuclei correspond to hematoxylin counterstaining, denoting all neuronal nuclei. **e**, Quantification of c-Fos intensity in the cortical region ($n = 5$ independent animals). Data are presented as the mean \pm standard deviation in (**b**, **c** and **e**) and were analyzed by one-way ANOVA first, followed by the Tukey's post hoc test in (**e**). NS, not significant.

Modifications to the manuscript:

(1) Figures:

The electrical measurement data previously presented as Fig. R1a-c have now been incorporated into Supplementary Fig. 16 (as c, e, f) in the revised Supporting Information, which reads as follows:

Supplementary Fig. 16. Current output of capacitive coupling system and ICH with varying input voltage.

a, Schematic illustration of current measurement points (I_1 and I_2) in the rat brain. I_1 represents the current through the ICHs, I_2 the total current across the entire brain. **b**, Schematic illustration of current measurement points (I_3 and I_4) in the rat brain. I_3 the current measured between the center of the ICHs and the adjacent brain tissue. I_4 serves as a control for I_1 , with electrodes placed in brain tissue but separated by an insulating PU membrane, mimicking an interface without ICH-mediated conduction. The electrode depth relative to the external electrode were strictly matched between the I_1 and I_4 conditions, and the only difference between the them was the presence or absence of the ICH. **c**, Schematic of the measurement configuration for cortical voltage (U_3) and current density (J_5), using the same relative electrode spacing as the deep-brain measurements (U_2 and J_4). **d**, Current densities (J_1 - J_4) under different input voltages (n = 5 independent experiments). (Note: to account for the contact impedance at the Cu wire-brain tissue interface, the measured values of I_2 and I_4 were multiplied by a correction factor $K = 1.5$. At 5 MHz, the contact impedance of the hemispherical electrode was modeled using a parallel RC circuit: The correction factor K , reflecting the ratio of ideal resistive current to actual current, was derived from the impedance magnitude, thus defined as $K = \sqrt{1 + (\omega RC)^2}$). **e**, Voltage measured in cortex (U_3) and deep brain (U_2) under different input voltages (n = 5 independent experiments). **f**, Corresponding current density in cortex (J_5) and deep brain (J_4) under different input voltages (n = 5 independent experiments). Data are presented as the mean \pm standard deviation in (**d**, **e**, **f**).

We have also added the cortical c-Fos data (Fig. R1d, e) as Supplementary Fig. 23 in the revised Supporting Information, which reads as follows:

Supplementary Fig. 23. c-Fos expression in the cortical region of rats. a, Representative images of c-Fos expression in the cortical region of rats. Scale bar, 100 μ m. The dark brown nuclei represent c-Fos signals, indicating activated neuronal nuclei, while the bluish-purple nuclei correspond to hematoxylin counterstaining, denoting all neuronal nuclei. **b,** Quantification of c-Fos intensity in the cortical region ($n = 5$ independent animals). Data are presented as the mean \pm standard deviation in (b) and were analyzed by one-way ANOVA first, followed by the Tukey's post hoc test. NS, not significant. p_1 (ICH (+) WES (+) vs ICH (+) WES (-)), p_2 (ICH (-) WES (-) vs ICH (+) WES (-)), p_3 (ICH (-) WES (-) vs ICH (+) WES (+)), p_4 (ICH (-) WES (+) vs ICH (+) WES (-)), p_5 (ICH (-) WES (+) vs ICH (+) WES (+)), p_6 (ICH (-) WES (+) vs ICH (-) WES (-)). **b,** $p_1 = 0.8821$, $p_2 = 0.9960$, $p_3 = 0.7757$, $p_4 = 0.9518$, $p_5 = 0.9966$, $p_6 = 0.8767$.

(2) Results section:

We have updated the Results section to include the cortical electrical response data, and the revised text has been marked in red in the manuscript (page 12-13), which reads as follows:

“We also evaluated whether cortical regions, which are closest to the external transmitter electrode, might be indirectly stimulated under volume conduction by measuring cortical voltage (U_3) and current density (J_5) in the absence of ICH, using the same relative electrode spacing as in the deep-brain measurements (U_2 and J_4) (Supplementary Fig. 16c). Both J_1 and J_3 involve ICH in the current path, and they exhibited similarly higher current density across with increasing input voltages (Supplementary Fig. 16d). Whereas J_4 , which lacked ICH, showed minimal current density. Furthermore, cortical (U_3/J_5) and deep-brain (U_2/J_4) measurements without ICH showed similar electrical responses (Supplementary Fig. 16e, f), indicating that the electrical field distributes broadly across brain layers in the absence of ICH, resulting in diffuse and insufficient stimulation for neuronal activation.”

We have revised the Results section to incorporate the cortical c-Fos findings (on page 16), with the updated text marked in red in the manuscript, which reads as follows:

“In the M1 region, to assess neuronal activation, we first quantified c-Fos expression in M1 (Supplementary Fig. 23). The results showed no significant difference in c-Fos expression across experimental groups, indicating that the cortical region did not receive a strong direct stimulation in a volume conduction scenario.”.

(3) Methods section:

We have clarified the cortical electrical measurements in Methods in the revised manuscript (page 33). The revised text has been marked in red in the manuscript and reads as follows:

“For comparison, cortical voltage (U_3) and current density (J_5) were recorded using the same relative electrode depth difference as the deep-brain measurements (U_2 and J_4).”.

Comment #2: *In the new revised figure 2a-c, the authors should measure the current between the edge of the ICH and the adjacent tissue. The center of the ICH is irrelevant. Also, to show the effect of the ICH, the exact same location should be measured with and without the ICH, as the J_4 is measured farther away from the capacitor, which might be the cause of the lower currents there. The dF/F results in fig 2d-f do not look anything like neuronal spikes. They appear only as moderate elevations in the baseline Ca level of the cells, with no indication of electrical activity, which should manifest as action potentials. It is not clear that these are neurons, as their morphology does not resemble that of differentiated SH-SY5Y cells and, as mentioned above, they do not exhibit neuron-like electrical activity. Calcium baseline levels can change when experiencing stress, which might be the case here. Fig. 2g-h still do not present the appropriate controls.*

Response: We would like to clarify your concern regarding the current measurement location. Measuring the current “between the edge of the ICH and the adjacent tissue” is not technically feasible with sufficient spatial accuracy because the hydrogel conforms to the surrounding tissue and the geometric boundary cannot be precisely defined during electrode insertion. Therefore, inserting the electrode into the center of the ICH while placing another electrode in the adjacent tissue represents the most robust and reproducible configuration for assessing the interface-related current response. Regarding your comment on J_4 , we acknowledge that the original schematic may have been misleading. Although J_4 appeared farther from the transmitter, the electrode depth relative to the external electrode were strictly matched between the J_1 and J_4 conditions, and the only difference between the them was the presence or absence of the ICH. The same depth-

matched configuration was applied for voltage measurements, where U_2 and U_1 differed only in whether the ICH was included in the current path. We have revised the schematic to accurately reflect the experimental setup, as shown in Fig. R2a-b.

Fig. R2. Updated current-measurement schematics, cell morphology and added flow-cytometry controls.

a, Schematic illustration of current measurement points (I_1 and I_2) in the rat brain. I_1 represents the current through the ICHs, I_2 the total current across the entire brain. **b**, Schematic illustration of current measurement points (I_3 and I_4) in the rat brain. I_3 the current measured between the center of the ICHs and the adjacent brain tissue. I_4 serves as a control for I_1 , with electrodes placed in brain tissue but separated by an insulating PU membrane, mimicking an interface without ICH-mediated conduction. The electrode depth relative to the external electrode were strictly matched between the I_1 and I_4 conditions, and the only difference between the them was the presence or absence of the ICH. **c**, Bright-field image showing morphology of SH-SY5Y cells. Scale bar, 50 μm . **d**, Phalloidin staining showing intact cytoskeleton. Scale bar, 50 μm . **e**, Representative flow cytometry plots of PC12 cells (ICH (-) WES (-), and ICH (-) WES (+)). **f**, Apoptosis rate of PC12 cells with and without WES ($n = 3$ independent experiments).

Regarding the calcium imaging concerns, we would like to emphasize that Ca^{2+} imaging reflects intracellular calcium accumulation secondary to membrane depolarization, and therefore its temporal profile is inherently slower than electrical action potentials. As widely recognized for Ca^{2+} indicators, the $\Delta\text{F}/\text{F}$ signals typically manifest as a rapid rise followed by a gradual decay due

to fast Ca²⁺ influx and slower Ca²⁺ clearance dynamics. Thus, Ca²⁺ traces generally appear as graded and sustained transients rather than spike-like waveforms, even in primary neurons (*Nat. Biomed. Eng.*, 2018, 2(7): 508-521.; *Nat. Biomed. Eng.*, 2023, 7(4): 486-498.), as exemplified by a canonical neuronal Ca²⁺ transient shown in Fig. R3a.

Furthermore, both differentiated SH-SY5Y cells (*Sci. Adv.*, 2021, 7(3): eabc4189.; *Nat. Biomed. Eng.*, 2023, 7(2): 149-163.) and undifferentiated SH-SY5Y cells (*Nat. Commun.*, 2023, 14(1): 8386.; *J. Am. Chem. Soc.*, 2025, 147(10): 8406-8421.; *ACS nano*, 2024, 18(26): 16853-16866.) have been extensively validated in stimulation-evoked Ca²⁺ imaging studies, as undifferentiated SH-SY5Y retain functional voltage-gated calcium influx and depolarization capability (representative examples shown in Fig. R3b and Fig. R3c). Prior comparative reports show that differentiation primarily modifies cellular morphology and synaptic phenotype but does not eliminate depolarization-induced Ca²⁺ responses; the main differences lie in kinetics, not responsiveness (*Sci. Rep.*, 2025, 15(1): 34196.). Therefore, neuronal differentiation is not a prerequisite for Ca²⁺-based functional assessment in SH-SY5Y models.

In our experiments, SH-SY5Y cells were not intentionally induced to undergo neuronal differentiation, and this is consistent with the cellular morphology observed in the bright-field images and actin cytoskeleton staining (phalloidin), which show only partial neurite outgrowth rather than fully differentiated neuronal phenotypes (Fig. R2c, d). Importantly, robust and stimulus-related $\Delta F/F$ increases were observed only under the ICH (+) WES (+) condition, while all control groups showed minimal activity, indicating that the Ca²⁺ elevations originate from targeted electrical depolarization rather than nonspecific stress-induced baseline fluctuations.

Finally, in response to your comment that appropriate controls were missing in the original Fig. 2g-h, we have now added the corresponding additional data. The additional data incorporates two other experimental conditions, specifically ICH (-) WES (-) and ICH (-) WES (+), as shown in Fig. R2e-f. These added control results confirm that in the absence of ICH, there is no evident difference in apoptosis between the WES (-) and WES (+) conditions. In other words, when ICH is not present, applying WES does not alter cell viability.

[REDACTED]

Fig. R3. Published examples demonstrating stimulation-evoked Ca²⁺ responses in neural cell models. a, Primary DRG neurons (reproduced from *Nat. Biomed. Eng.*, 2023, 7(4): 486-498.). **b,** Differentiated SH-SY5Y cells (adapted from *Sci. Adv.*, 2021, 7(3): eabc4189.). **c,** Undifferentiated SH-SY5Y cells (adapted from *ACS nano*, 2024, 18(26): 16853-16866.).

Modifications to the manuscript:

(1) Figures:

We have added the flow cytometry analysis data (Fig. R2e) as Supplementary Fig. 20 in the revised Supporting Information, and replaced the apoptosis quantification data with the updated statistical data (Fig. R2f), now shown as the revised Fig. 2h in the main manuscript. We have

updated the legend of Fig. 2b to clarify the electrode configuration used for U_1 and U_2 recordings, which has been marked in red in the revised manuscript (page 54)

Supplementary Fig. 20. Flow cytometry analysis of PC12 cell viability under ICH (-) conditions. Representative flow cytometry plots of PC12 cells with and without WES treatment.

Fig. 2. ICH-mediated wireless neural stimulation *in vitro*. **a**, Schematic diagram of the measurement of localized electric potential and concentrated current density in ICH. **b**, Output voltages (U_1 , U_2) exhibit a linear dependence on input voltage ($n = 5$ independent experiments). The electrode depth relative to the external electrode were strictly matched between the U_1 and U_2 conditions, and the only difference between the them was the presence or absence of the ICH. **c**, Current density (J_1 , J_2) versus input voltage reveals a proportional relationship, showing enhanced current focusing at J_1 ($n = 5$ independent experiments). **d**, Time-lapse calcium imaging of SH-SY5Y cells under different conditions. Red asterisks indicate activated cells. Scale bar, 50 μm . **e**, Representative $\Delta F/F_0$ traces showing stimulation-dependent fluorescence enhancement in the ICH (+) WES (+) group. The inset highlights $\Delta F/F_0$ across all groups ($n = 5$ independent experiments). **f**, Quantitative comparison of excited cells across all conditions, confirming that cellular activation occurs exclusively in the ICH (+) WES (+) group ($n = 5$ independent experiments). **g**, Flow cytometry analysis of PC12 cells viability with/without WES. **h**, Apoptosis rate of PC12 cells across all groups ($n = 3$ independent experiments). Data are presented as the mean \pm standard deviation in (**c**, **e**, **f**, and **h**) and were analyzed by one-way ANOVA first, followed by the Tukey's post hoc test in (**e**, **f**, and **h**). *** $P \leq 0.001$. NS, not significant. p_1 (ICH (+) WES (+) vs ICH (+) WES (-)), p_2 (ICH (-) WES (-) vs ICH (+) WES (-)), p_3 (ICH (-) WES (-) vs ICH (+) WES (+)), p_4 (ICH (-) WES (+) vs ICH (+) WES (-)), p_5 (ICH (-) WES (+) vs ICH (+) WES (+)), p_6 (ICH (-) WES (+) vs ICH (-) WES (-)). **e**, $p_1 = 0.0001$, $p_2 = 0.9991$, $p_3 = 0.0001$, $p_4 = 0.9914$, $p_5 = 0.0001$, $p_6 = 0.9986$. **f**, $p_1 = 0.0001$, $p_2 = 0.9976$, $p_3 = 0.0001$, $p_4 = 0.9878$, $p_5 = 0.0001$, $p_6 = 0.999$. **h**, $p_1 = 0.6760$, $p_2 = 0.9807$, $p_3 = 0.4710$, $p_4 = 0.5972$, $p_5 = 0.9989$, $p_6 = 0.4021$.

We have updated the current-measurement schematics by replacing the previous one in Supplementary Fig. 16a with the revised versions (corresponding to Fig. R2a, b), which now appear as Supplementary Fig. 16a, b:

Supplementary Fig. 1. Current output of capacitive coupling system and ICH with varying input voltage.

a, Schematic illustration of current measurement points (I_1 and I_2) in the rat brain. I_1 represents the current

through the ICHs, I_2 the total current across the entire brain. **b**, Schematic illustration of current measurement points (I_3 and I_4) in the rat brain. I_3 the current measured between the center of the ICHs and the adjacent brain tissue. I_4 serves as a control for I_1 , with electrodes placed in brain tissue but separated by an insulating PU membrane, mimicking an interface without ICH-mediated conduction. The electrode depth relative to the external electrode were strictly matched between the I_1 and I_4 conditions, and the only difference between the them was the presence or absence of the ICH. **c**, Schematic of the measurement configuration for cortical voltage (U_3) and current density (J_5), using the same relative electrode spacing as the deep-brain measurements (U_2 and J_4). **d**, Current densities (J_1 - J_4) under different input voltages ($n = 5$ independent experiments). (Note: to account for the contact impedance at the Cu wire-brain tissue interface, the measured values of I_2 and I_4 were multiplied by a correction factor $K = 1.5$. At 5 MHz, the contact impedance of the hemispherical electrode was modeled using a parallel RC circuit: The correction factor K , reflecting the ratio of ideal resistive current to actual current, was derived from the impedance magnitude, thus defined as $K = \sqrt{1 + (\omega RC)^2}$). **e**, Voltage measured in cortex (U_3) and deep brain (U_2) under different input voltages ($n = 5$ independent experiments). **f**, Corresponding current density in cortex (J_5) and deep brain (J_4) under different input voltages ($n = 5$ independent experiments). Data are presented as the mean \pm standard deviation in (**d**, **e**, **f**).

(2) Results section:

The Results section has also been updated to clarify the interpretation of the current-path comparison on page 12-13, with the revised text marked in red in the manuscript, which reads as follows:

“To further study the influence of ICH on the current path, we compared J_1 with additional configurations including J_3 (from the center of the ICH to adjacent-tissue) and J_4 (tissue-tissue with PU insulation) (Supplementary Fig. 16a, b). We also evaluated whether cortical regions, which are closest to the external transmitter electrode, might be indirectly stimulated under volume conduction by measuring cortical voltage (U_3) and current density (J_5) in the absence of ICH, using the same relative electrode spacing as in the deep-brain measurements (U_2 and J_4) (Supplementary Fig. 16c). Both J_1 and J_3 involve ICH in the current path, and they exhibited similarly higher current density across with increasing input voltages (Supplementary Fig. 16d). Whereas J_4 , which lacked ICH, showed minimal current density. Furthermore, cortical (U_3/J_5) and deep-brain (U_2/J_4) measurements without ICH showed similar electrical responses (Supplementary Fig. 16e, f), indicating that the electrical field distributes broadly across brain layers in the absence of ICH, resulting in diffuse and insufficient stimulation for neuronal activation.”

The text in the Results has been updated to clarify the flow cytometry analysis on page 14-15, with the revised sentences marked in red, which read as follows:

“When ICH was not present, the WES (-) and WES (+) groups exhibited comparable distributions of live, early apoptotic, and late apoptotic or necrotic cells, indicating that WES alone does not induce cellular stress or cytotoxicity (Supplementary Fig. 20). When ICH was present, apoptosis rates in the ICH (+) WES (-) and ICH (+) WES (+) groups remained comparable, demonstrating that the stimulation delivered through ICH does not compromise cell viability (Fig. 2g, h).”.

(3) Methods section:

We have clarified the flow cytometry control measurements in Methods in the revised manuscript (page 35). The revised text has been marked in red in the manuscript and reads as follows:

“For flow cytometry analysis, the ICH (-) WES (-) and ICH (-) WES (+) control groups were tested using the same procedure, except that the Cu wires were directly implanted into normal brain tissue without ICH while all stimulation parameters remained identical to those used in the ICH (+) groups.”.

Comment #3: *The authors have addressed my concerns regarding the proper controls. However, the authors here look only at the cells around the ICH, while according to fig. 1e, other regions might be activated as well. Thus, to show local activity, it is critical to check these regions (the cortex near the capacitor) for the same activated neurons.*

Response: As detailed in our response to Comment #1, we have now performed additional cortical voltage/current measurements and c-Fos analysis to experimentally verify whether the cortex is activated during WES. These results (Fig. R1) show that, although the cortex experiences a higher absolute potential due to its proximity to the external transmitter, the electric field there is spatially diffuse and thus insufficient for neuronal activation. Consistently, cortical c-Fos expression remained unchanged across all conditions (Fig. R1c, d), confirming the absence of neuronal activation in this region. Together, these results demonstrate that effective stimulation occurs selectively near the ICH interface, where the conductivity/permittivity discontinuity focuses the electric field and enables localized neural excitation in deep brain tissue.

Fig. R1. Quantitative validation of cortical responses under WES. **a**, Schematic of the measurement configuration for cortical voltage (U_3) and current density (J_5), using the same relative electrode spacing as the deep-brain measurements (U_2 and J_4). **b**, Voltage measured in cortex (U_3) and deep brain (U_2) under different input voltages ($n = 5$ independent experiments). **c**, Corresponding current density in cortex (J_5) and deep brain (J_4) under different input voltages ($n = 5$ independent experiments). **d**, Representative images of c-Fos expression in the cortical region of rats. Scale bar, 100 μm . The dark brown nuclei represent c-Fos signals, indicating activated neuronal nuclei, while the bluish-purple nuclei correspond to hematoxylin counterstaining, denoting all neuronal nuclei. **e**, Quantification of c-Fos intensity in the cortical region ($n = 5$ independent animals). Data are presented as the mean \pm standard deviation in (**b**, **c** and **e**) and were analyzed by one-way ANOVA first, followed by the Tukey's post hoc test in (**e**). NS, not significant.

Modifications to the manuscript:

(2) Figures:

The electrical measurement data previously presented as Fig. R1a-c have now been incorporated into Supplementary Fig. 16 (as c, e, f) in the revised Supporting Information, which reads as follows:

Supplementary Fig. 16. Current output of capacitive coupling system and ICH with varying input voltage.

a, Schematic illustration of current measurement points (I_1 and I_2) in the rat brain. I_1 represents the current through the ICHs, I_2 the total current across the entire brain. **b**, Schematic illustration of current measurement points (I_3 and I_4) in the rat brain. I_3 the current measured between the center of the ICHs and the adjacent brain tissue. I_4 serves as a control for I_1 , with electrodes placed in brain tissue but separated by an insulating PU membrane, mimicking an interface without ICH-mediated conduction. The electrode depth relative to the external electrode were strictly matched between the I_1 and I_4 conditions, and the only difference between the them was the presence or absence of the ICH. **c**, Schematic of the measurement configuration for cortical voltage (U_3) and current density (J_5), using the same relative electrode spacing as the deep-brain measurements (U_2 and J_4). **d**, Current densities (J_1 - J_4) under different input voltages ($n = 5$ independent experiments). (Note: to account for the contact impedance at the Cu wire-brain tissue interface, the measured values of I_2 and I_4 were multiplied by a correction factor $K = 1.5$. At 5 MHz, the contact impedance of the hemispherical electrode was modeled using a parallel RC circuit: The correction factor K , reflecting the ratio of ideal resistive current to actual current, was derived from the impedance magnitude, thus defined as $K = \sqrt{1 + (\omega RC)^2}$). **e**, Voltage measured in cortex (U_3) and deep brain (U_2) under different input voltages ($n = 5$ independent experiments). **f**, Corresponding current density in cortex (J_5) and deep brain (J_4) under different input voltages ($n = 5$ independent experiments). Data are presented as the mean \pm standard deviation in (**d**, **e**, **f**).

We have also added the cortical c-Fos data (Fig. R1d, e) as Supplementary Fig. 23 in the revised Supporting Information, which reads as follows:

Supplementary Fig. 23. c-Fos expression in the cortical region of rats. **a**, Representative images of c-Fos expression in the cortical region of rats. Scale bar, 100 μ m. The dark brown nuclei represent c-Fos signals, indicating activated neuronal nuclei, while the bluish-purple nuclei correspond to hematoxylin counterstaining, denoting all neuronal nuclei. **b**, Quantification of c-Fos intensity in the cortical region ($n = 5$ independent animals). Data are presented as the mean \pm standard deviation in **(b)** and were analyzed by one-way ANOVA first, followed by the Tukey's post hoc test. NS, not significant. p_1 (ICH (+) WES (+) vs ICH (+) WES (-)), p_2 (ICH (-) WES (-) vs ICH (+) WES (-)), p_3 (ICH (-) WES (-) vs ICH (+) WES (+)), p_4 (ICH (-) WES (+) vs ICH (+) WES (-)), p_5 (ICH (-) WES (+) vs ICH (+) WES (+)), p_6 (ICH (-) WES (+) vs ICH (-) WES (-)). **b**, $p_1 = 0.8821$, $p_2 = 0.9960$, $p_3 = 0.7757$, $p_4 = 0.9518$, $p_5 = 0.9966$, $p_6 = 0.8767$.

(2) Results section:

We have updated the Results section to include the cortical electrical response data, and the revised text has been marked in red in the manuscript (page 12-13), which reads as follows:

“We also evaluated whether cortical regions, which are closest to the external transmitter electrode, might be indirectly stimulated under volume conduction by measuring cortical voltage (U_3) and current density (J_5) in the absence of ICH, using the same relative electrode spacing as in the deep-brain measurements (U_2 and J_4) (Supplementary Fig. 16c). Both J_1 and J_3 involve ICH in the current path, and they exhibited similarly higher current density across with increasing input voltages (Supplementary Fig. 16d). Whereas J_4 , which lacked ICH, showed minimal current density. Furthermore, cortical (U_3/J_5) and deep-brain (U_2/J_4) measurements without ICH showed similar electrical responses (Supplementary Fig. 16e, f), indicating that the electrical field distributes broadly across brain layers in the absence of ICH, resulting in diffuse and insufficient stimulation for neuronal activation.”

We have revised the Results section to incorporate the cortical c-Fos findings (on page 16), with the updated text marked in red in the manuscript, which reads as follows:

“In the M1 region, to assess neuronal activation, we first quantified c-Fos expression in M1 (Supplementary Fig. 23). The results showed no significant difference in c-Fos expression across experimental groups, indicating that the cortical region did not receive a strong direct stimulation in a volume conduction scenario.”.

(3) Methods section:

We have clarified the cortical electrical measurements in Methods in the revised manuscript (page 33). The revised text has been marked in red in the manuscript and reads as follows:

“For comparison, cortical voltage (U_3) and current density (J_5) were recorded using the same relative electrode depth difference as the deep-brain measurements (U_2 and J_4).”.

Comment #4: *Minor- Authors should provide the behavioral tests at the beginning, i.e., week 0. It will help appreciate the effect when the score starts at the same level.*

Response: We agree that that adding the baseline behavioral performance at week 0 would provide clearer insight into group comparability prior to treatment. Accordingly, we have now provided the week-0 behavioral data, as shown in Fig. R4 below. These results demonstrate that all groups exhibited similar baseline scores prior to treatment, confirming equivalent initial status across groups.

Fig. R4. Locomotor performance assessment in PD rats after ICH-mediated wireless DBS. a, Trajectory images of OFT after the ICH-mediated wireless DBS (n = 5 independent animals). **b,** Total travelled distance of rats in different groups (n = 5 independent animals). **c,** Max speed of rats in different groups (n = 5 independent animals). **d,** Active time of rats in different groups (n = 5 independent animals).

Modifications to the manuscript:

(1) Figures:

The week-0 data have now been incorporated into the main figures and Supplementary Information. Specifically, the ICH (+) WES (+) trajectory images originally shown in Fig. R3a, together with the updated quantitative results from Fig. R3b-d, have replaced the previous baseline data in Fig. 4. The remaining week-0 data from Fig. R3a for the other experimental groups have been moved to the Supplementary Information and are now presented as Supplementary Fig. 28.

Fig. 4. ICH-mediated wireless DBS alleviates parkinsonian symptoms in PD rats. b, Trajectory images of OFT (n = 5 independent animals). c, Total travelled distance of rats (n = 5 independent animals). d, Max speed of rats (n = 5 independent animals). e, Active time of rats (n = 5 independent animals).

Supplementary Fig. 2. Open field test of PD rats during 4-week treatment. Trajectory images of open field test for evaluation of the locomotor behaviour of PD rats in different groups. The open field test was conducted in a chamber with an automated video tracking system for 5 min.

(2) Results section:

The text in the Results has been updated to explicitly state that week-0 testing was performed and that no significant differences were observed among groups before treatment (on page 18), which was marked in red and reads as follows:

“Behavioral assessments at week 0 showed no significant differences among groups, confirming comparable pre-operative functional status (Fig. 4b and Supplementary Fig. 28). After stimulation, ICH-mediated DBS improved locomotor activity, as indicated by increased movement distance, maximum speed, and active time. In contrast, the ICH (+) WES (-) and ICH (-) WES (+) groups showed no significant improvement in locomotor activity (Fig. 4c-e).”.

Reviewer's Comments:

Reviewer #1

Summary Comments

Authors response to my comments are satisfactory, and I think the manuscript is suitable for publication. I have only one minor suggestion: Can the authors add to fig 1e the Finite element simulation results of the local electric field and not only the charge distribution? This can help the readers to avoid misinterpretation of the results described here.

Response: We thank the reviewer for the positive assessment and the constructive suggestion. Accordingly, we have now added the finite element simulation results of the local electric field to Fig. 1e, alongside the charge distribution (which now as Fig. 1f).

Modifications to the manuscript:

(1) Figures:

The finite element simulation of the local electric field distribution has been added as the new Fig. 1e. The original charge distribution simulation previously shown in Fig. 1e has been moved to Fig. 1f. In addition, the frequency-adaptive circuit model, originally presented as Fig. 1f, has been relocated to the Supporting Information and is now shown as Supplementary Fig. 14a. The revised figure organization reads as follows:

Fig. 1. Wireless DBS with injectable hydrogel bioelectronics. **a**, The preparation of *in vivo* injectable conductive hydrogel (ICH) with glucose-initiated gelation within brain tissue. **b**, Bio-catalytic polymerization and electrostatic assembly mechanism of ICHs. **GOx**, glucose oxidase; **HRP**, horseradish peroxidase; **PEDOT:PSS**, poly(3,4-ethylenedioxythiophene): polystyrene sulfonate. **c**, Representative images show *in situ* gelation of ICHs in tissue-mimicking agarose gel over time. Scale bar, 5 mm. **d**, A wearable capacitive-coupling system induces weak volume conduction in brain tissue via capacitive wireless power transfer, and the impedance mismatch between ICH and native tissue drives interfacial charge redistribution at the hydrogel-tissue interface. **STN**, subthalamic nucleus. **e**, Finite element simulation of electric field intensity distribution across the scalp-skull-brain multilayered structure, incorporating tissue-specific dielectric properties, demonstrates electrical field localization at the ICH-tissue interface due to interfacial polarization. **f**, Finite element simulation of surface charge distribution in the multilayer head model, demonstrates charge accumulation at the ICH-tissue interface due to interfacial polarization.

Supplementary Fig. 1. Frequency-dependent specific absorption rate (SAR). **a**, Frequency-adaptive circuit model integrating capacitive coupling, multi-tissue impedance, and brain admittance with interfacial polarization mechanisms. **b**, SAR values as a function of frequency, indicating SAR remains below the biosafety threshold at frequency of 5 MHz.

(2) Results section:

We have updated the Results section and the revised text has been marked in red in the manuscript (page 11), which reads as follows:

“At 5 MHz, the introduction of the ICH markedly enhanced local electric field intensity around the implantation site (Fig. 1e and Supplementary Fig. 13a). The degree of field localization increased with the conductivity of the ICH, indicating stronger polarization and field confinement near highly conductive hydrogels. Correspondingly, the surface charge density at the ICH-tissue interface also increased with conductivity (Fig. 1f and Supplementary Fig. 13b), consistent with enhanced interfacial polarization caused by permittivity and conductivity mismatch between the two media.”.

“To assess biosafety of such a wireless power transfer process, we calculated the specific absorption rate (SAR) using a frequency-adaptive circuit model (Supplementary Fig. 14a) that incorporates capacitive transmitter dynamics, multilayer tissue impedance, and interfacial polarization elements.”

“Across all tested frequencies (≤ 5 MHz), the calculated SAR remained below 2 W kg^{-1} , well within internationally accepted safety limits for local tissue exposure³⁷, indicating safe operation under the capacitive coupling-based volume conduction scenario (Supplementary Fig. 14b).”